# Content-Style Learning from Unaligned Domains: Identifiability under Unknown Latent Dimensions

**Sagar Shrestha and Xiao Fu**
School of Electrical Engineering and Computer Science
Oregon State University
Corvallis, OR 97331, USA
`{shressag,xiao.fu}@oregonstate.edu`

## Abstract

Understanding identifiability of latent content and style variables from unaligned multi-domain data is essential for tasks such as domain translation and data generation. Existing works on content-style identification were often developed under somewhat stringent conditions, e.g., that all latent components are mutually independent and that the dimensions of the content and style variables are known. We introduce a new analytical framework via cross-domain *latent distribution matching* (LDM), which establishes content-style identifiability under substantially more relaxed conditions. Specifically, we show that restrictive assumptions such as component-wise independence of the latent variables can be removed. Most notably, we prove that prior knowledge of the content and style dimensions is not necessary for ensuring identifiability, if sparsity constraints are properly imposed onto the learned latent representations. Bypassing the knowledge of the exact latent dimension has been a longstanding aspiration in unsupervised representation learning—our analysis is the first to underpin its theoretical and practical viability. On the implementation side, we recast the LDM formulation into a regularized multi-domain GAN loss with coupled latent variables. We show that the reformulation is equivalent to LDM under mild conditions—yet requiring considerably less computational resource. Experiments corroborate with our theoretical claims.

## 1 Introduction

In multi-domain learning, "domains" are typically characterized by a distinct "style" that sets their data apart from others (Choi et al., 2020). Take handwritten digits as an example: writing styles of different persons can define different domains. Shared information across all domains, such as the identities of the digits in this case, is termed as "content". Learning content and style representations from multi-domain data facilitates many important applications, e.g., domain translation (Huang et al., 2018), image synthesis (Choi et al., 2020), and self-supervised representation learning (Von Kügelgen et al., 2021; Lyu et al., 2022; Daunhawer et al., 2023); see more in Huang et al. (2018); Lee et al. (2020); Choi et al. (2020); Wang et al. (2016); Yang et al. (2020); Wu et al. (2019).

Recent advances showed that understanding the *identfiability* of the latent content and style components from multi-domain data allows to design more reliable, predicable, and trustworthy learning systems (Hyvarinen et al., 2019; Lyu et al., 2022; Xie et al., 2023; Kong et al., 2022; Shrestha & Fu, 2024; Gresele et al., 2020; Gulrajani & Hashimoto, 2022). A number of works studied content/style identifiability when the multi-domain data have *sample-to-sample cross-domain alignment* according to shared contents. Specifically, identifiability was established for sample-aligned multi-domain settings under the assumption that multi-domain data are linear and nonlinear mixtures of latent content and style components, in the context of canonical correlation analysis (CCA), multiview analysis and self-supervised learning (SSL); see Ibrahim et al. (2021); Sørensen et al. (2021); Wang & Isola (2020); Von Kügelgen et al. (2021); Lyu et al. (2022); Karakasis & Sidiropoulos (2023); Daunhawer et al. (2023)

When cross-domain samples are *unaligned*, it becomes significantly more challenging to establish identifiability of the content and style components. The recent works in Xie et al. (2023); Sturma et al. (2023); Kong et al. (2022); Timilsina et al. (2024) made meaningful progresses towards this goal. These works considered mixture models of content and style for each domain, similar to those in Lyu et al. (2022); Von Kügelgen et al. (2021); Ibrahim et al. (2021); Sørensen et al. (2021); Karakasis & Sidiropoulos (2023); Daunhawer et al. (2023), but without cross-domain alignment. The new results in (Xie et al., 2023; Sturma et al., 2023; Kong et al., 2022; Timilsina et al., 2024) provide theory-backed solutions to a suite of timely and important applications, e.g., cross-language retrieval, multimodal single cell data alignment, causal representation learning, and image data translation and generation.

**Challenges.** The content-style identifiability results in existing unaligned multi-domain learning works are intriguing and insightful, but some challenges remain. First, the conditions used in their proofs have a number of restrictions, which limits the proof's applicability in many cases. For example, Sturma et al. (2023); Timilsina et al. (2024) assume that the all data reside in a linear subspace, which is over-simplification of reality; Xie et al. (2023); Kong et al. (2022) assume that the content and style variables are component-wise independent and that a large number of domains exist—both can be hard to fulfil. Second, the existing identifiability analyses in unaligned multi-domain learning (Xie et al., 2023; Kong et al., 2022; Sturma et al., 2023; Timilsina et al., 2024) (as well as those in aligned multi-domain learning) all need to know the dimensions of the content and style variables, which are not available in practice. Selecting these dimensions often involves extensive trial and error.

**Contributions.** In this work, we advance the analytical and computational aspects of content-style learning from unaligned multi-domain data. Our detailed contributions are as follows:

*(i) Enhanced Identifiability of Content and Style*: We propose a content-style identification criterion via constrained *latent distribution matching* (LDM). We show that the identifiability conditions under LDM are much more relaxed relative to those in existing works. Specifically, our results hold for nonlinear mixture models, as opposed to the linear ones used in Sturma et al. (2023); Timilsina et al. (2024). Unlike Xie et al. (2023); Kong et al. (2022); Sturma et al. (2023), no elementwise mutual independence assumption is needed in our proof. More importantly, our result holds for as few as *two* domains (whereas Xie et al. (2023); Kong et al. (2022) needs the existence of a large number of domains). The new results widens the applicability of content-style identifiable models in a substantial way.

*(ii) Content-Style Identifiability under Unknown Latent Dimensions*: We consider the scenario where the latent content and style dimensions are unknown—which is the case in practical settings. Note that existing works determine the content and style dimensions often by heuristics, e.g., trial-and-error. However, wrongly selected latent dimensions can largely degrade the performance of some tasks; e.g., an over-estimated style dimension hinders the diversity of data in generation tasks (see Sec. 6). We show that, by imposing proper sparsity constraints onto the LDM formulation, the content-style identifiability is retained even without knowing the exact latent dimensions. To our knowledge, this result is the first of the kind in the context of nonlinear mixture identification.

*(iii) Efficient Implementation*: We prove that the LDM formulation is equivalent to a sparsity-constrained, latent variable-coupled muti-domain GAN loss, under reasonable conditions. Directly realizing the LDM formulation would impose multiple complex modules, including the DM and content-style separation modules, in the learned latent domain. Simultaneously learning the latent space and optimizing these modules can be computationally involved. The GAN-based formulation circumvents such complicated operations and thus substantially simplifies the implementation.

For theory validation, we perform experiments over a series of image translation and generation tasks.

**Notation.** Please see Appendix A.1 for detailed notation designation. A particular remark is that $\mathbb{P}_{\boldsymbol{x}}$ and $p_{\boldsymbol{x}}(\cdot)$ represent the probability measure of $\boldsymbol{x}$ and the probability density function (PDF) of $\boldsymbol{x}$, respectively. The "push forward" notation $[\boldsymbol{f}]_{\#\mathbb{P}_{\boldsymbol{x}}}$ means the distribution of $\boldsymbol{f}(\boldsymbol{x})$.

## 2    BACKGROUND

**Content-Style Modeling in Multi-Domain Analysis.** Consider the case where the data are acquired over $N$ domains $\mathcal{X}^{(n)} \subseteq \mathbb{R}^d$, where $n = 1, \ldots, N$. We assume that any sample from domain $n$ can be represented as a function (or, a nonlinear mixture) of content and style components, i.e.,

$$\boldsymbol{c} \sim \mathbb{P}_{\boldsymbol{c}}, \; \boldsymbol{s}^{(n)} \sim \mathbb{P}_{\boldsymbol{s}^{(n)}}, \; \boldsymbol{x}^{(n)} = \boldsymbol{g}(\boldsymbol{c}, \boldsymbol{s}^{(n)}), \tag{1}$$

where $\mathbb{P}_{\boldsymbol{s}^{(n)}}$ and $\mathbb{P}_{\boldsymbol{c}}$ are distributions of the style components in $n$th domain and the content components, respectively. Let $\mathcal{C} \subseteq \mathbb{R}^{d_C}$ and $\mathcal{S}^{(n)} \subseteq \mathbb{R}^{d_S}$ be the open set supports of $\mathbb{P}_{\boldsymbol{c}}$ and $\mathbb{P}_{\boldsymbol{s}^{(n)}}$. Then, we define $\mathcal{X}^{(n)} = \{\boldsymbol{g}(\boldsymbol{c}, \boldsymbol{s}^{(n)}) | \boldsymbol{c}, \boldsymbol{s}^{(n)} \in \mathcal{C} \times \mathcal{S}^{(n)}\} \subseteq \mathbb{R}^d$ as the support of $\boldsymbol{x}^{(n)} \sim \mathbb{P}_{\boldsymbol{x}^{(n)}}$. Let $\mathcal{X} = \cup_{n=1}^{N} \mathcal{X}^{(n)} \subseteq \mathbb{R}^d$ and $\mathcal{S} = \cup_{n=1}^{N} \mathcal{S}^{(n)} \subseteq \mathbb{R}^{d_S}$ represent the whole data space and the whole style space, respectively. We assume that the nonlinear function $\boldsymbol{g} : \mathcal{C} \times \mathcal{S} \to \mathcal{X}$ is a differentiable *bijective* function. This is a common assumption in latent component identification works, e.g., Von Kügelgen et al. (2021); Hyvarinen et al. (2019); Khemakhem et al. (2020), which basically says that every data sample has an associated unique representation in a latent domain. A remark is that although $\mathcal{X} \subseteq \mathbb{R}^d$ and $d$ might be greater than $d_S + d_C$, the bijective property can hold as $\mathcal{X}$ resides within a low dimensional manifold (Von Kügelgen et al., 2021).

The model in (1) is widely adopted (explicitly or implicitly) in multi-domain analysis; see examples from Huang et al. (2018); Lee et al. (2020); Choi et al. (2020); Wang et al. (2016); Yang et al. (2020); Wu et al. (2019). This model makes sense when the "domains" are participated using distinguishable semantic meaning; e.g., in Fig. 1, "style" includes the writing manners (handwritten/printed) and display background colors (black/gray). Under the model in (1), learning $\boldsymbol{g}$ (and its inverse $\boldsymbol{f}$) as well as the latent components $\boldsymbol{c}$ and $\boldsymbol{s}^{(n)}$ is the key to facilitate a number of important applications.

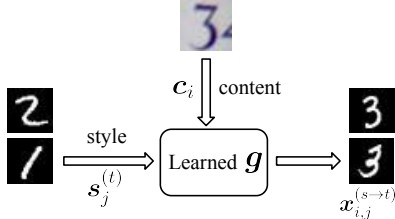

Figure 1: Cross-domain translation from source domain $s$ to target domain $t$.

*Application: Cross-Domain Translation.* Learning content and style components from a sample in the source domain $(\boldsymbol{c}_i, \boldsymbol{s}_i^{(s)}) = \boldsymbol{f}(\boldsymbol{x}_i^{(s)})$ and a sample from the target domain $(\boldsymbol{c}_j, \boldsymbol{s}_j^{(t)}) = \boldsymbol{f}(\boldsymbol{x}_j^{(t)})$ can assist translate $\boldsymbol{x}_i^{(s)}$ to its corresponding representation in the target domain. This can be realized by generating a new sample $\boldsymbol{x}_{i,j}^{(s \to t)} = \boldsymbol{g}(\boldsymbol{c}_i, \boldsymbol{s}_j^{(t)})$; see Fig. 1 for illustration and Lyu et al. (2022); Huang et al. (2018); Wang et al. (2016).

*Application: Data Generation.* If $\boldsymbol{c}$ and $\boldsymbol{s}^{(n)}$ can be learned from the samples, then one can also learn the distributions $\mathbb{P}_{\boldsymbol{c}}$ and $\mathbb{P}_{\boldsymbol{s}^{(n)}}$ using off-the-shelf distribution learning tools, e.g., GAN (Goodfellow et al., 2014). This way, one can draw samples from the distributions, i.e., $\boldsymbol{c}_{\text{new}} \sim \mathbb{P}_{\boldsymbol{c}}, \; \boldsymbol{s}_{\text{new}}^{(n)} \sim \mathbb{P}_{\boldsymbol{s}^{(n)}}$ and generate new samples $\boldsymbol{x}_{\text{new}}^{(n)} = \boldsymbol{g}(\boldsymbol{c}_{\text{new}}, \boldsymbol{s}_{\text{new}}^{(n)})$ with intended styles.

*Other Applications.* We should mention that the content-style modeling is also a critical perspective for understanding representation learning paradigms, e.g., the SSL frameworks (Von Kügelgen et al., 2021; Lyu et al., 2022; Daunhawer et al., 2023; Wang & Isola, 2020).

**Content-Style Identifiability.** In recent years, the identifiability of $\boldsymbol{f}$, $\boldsymbol{c}$ and $\boldsymbol{s}^{(n)}$ started drawing attention, due to its usefulness in building more reliable/predictable systems.

*Aligned Domains: Results from Self-Supervised Learning (SSL).* The works (Von Kügelgen et al., 2021; Daunhawer et al., 2023; Lyu et al., 2022; Karakasis & Sidiropoulos, 2023) studied content identifiability in the context of representation learning, in particular, SSL and multiview learning. It was shown that when $N = 2$, if content-shared pairs $\{\boldsymbol{x}^{(1)}, \boldsymbol{x}^{(2)}\}$ are available, then enforcing $\boldsymbol{f}(\boldsymbol{x}^{(1)}) = \boldsymbol{f}(\boldsymbol{x}^{(2)})$, $\forall$ content-shared pairs $(\boldsymbol{x}^{(1)}, \boldsymbol{x}^{(2)})$ can provably learn $\boldsymbol{c}$, under reasonable conditions. The learning criterion can be realized by various loss functions, e.g., Euclidean fitting-based (Lyu et al., 2022; Karakasis & Sidiropoulos, 2023) and contrastive loss-based (Von Kügelgen et al., 2021; Daunhawer et al., 2023) criteria. The identifiability of the style components was also considered under similar aligned domain settings; see (Lyu et al., 2022; Eastwood et al., 2023).

*Unaligned Domains: Progresses and Challenges.* Aligned samples are readily available in applications such as data-augmented SSL (Von Kügelgen et al., 2021; Daunhawer et al., 2023; Lyu et al.,

2022). However, in other applications such as image style translation and image generation, aligned samples are hard to acquire (Zhu et al., 2017). For unaligned multi-domain data, the identfiiability issue of content and style has also been recently addressed. For example, the work of Sturma et al. (2023) extended the linear ICA model to unaligned multi-domain settings, in the context of causal learning. The work of Timilsina et al. (2024) took a similar linear mixture model but showed content-style identifiability under more relaxed conditions. The work of Xie et al. (2023); Kong et al. (2022) proved content-style identifiability under a more realistic nonlinear mixture model similar to that in (1). However, the main result there relies on a number of somewhat stringent conditions. That is, two notable assumptions in Xie et al. (2023); Kong et al. (2022) boil down to (i) that all components in $z = (c, s^{(n)})$ are elementwise statistically independent given the domain index $n$; and (ii) that there exist at least $2d_S + 1$ domains. These conditions can be hard to fulfil. See more detailed discussions on existing results in Appendix B.

**The Dimension Knowledge Challenge.** Notably, all the existing works in this domain (under both aligned and unaligned settings) assume that the dimensions of $c$ and $s^{(n)}$ are known. However, in mixture model learning, such knowledge is hard to acquire (especially in the nonlinear mixture case). As we will show, using wrongly selected $d_C$ and $d_S$ can be rather detrimental to content-style learning tasks—e.g., an over-estimated style dimension could lead to a serious lack of diversity in generated new samples. Consequently, the dimensions are often selected by extensive trial and error in practice.

## 3 MAIN RESULT

In this work, we revisit content-style learning from a latent distribution matching (LDM) viewpoint. Recall that $c$ and $s^{(n)}$ represent the content and the style of the $n$th domain, respectively. We assume:

**Assumption 3.1** (Block Independence). The block variables $c \in \mathbb{R}^{d_C}$ and $\{s^{(n)} \in \mathbb{R}^{d_S}\}_{n=1}^N$ are statistically independent, i.e., $p(c, s^{(1)}, \ldots, s^{(N)}) = p_c(c) \prod_{n=1}^N p_{s^{(n)}}(s^{(n)})$.

The assumption was used in various multi-domain models (Lyu et al., 2022; Eastwood et al., 2023; Wang et al., 2016; Choi et al., 2020; Timilsina et al., 2024). It makes sense when the styles can be combined with contents in an "arbitrary" way without affecting the contents (e.g., the writing style of digits can change freely without affecting the identity of the digits). Next, we will use this assumption to build our learning criterion. We propose the following learning criterion:

$$\text{find } f : \mathcal{X} \to \mathbb{R}^{d_C + d_S} \text{ injective}$$
$$\text{s.t. } [f_C]_{\#\mathbb{P}_{x^{(i)}}} = [f_C]_{\#\mathbb{P}_{x^{(j)}}}, \; i \neq j, \forall i, j \in [N], \text{ (distribution matching)} \tag{2a}$$
$$[f_S]_{\#\mathbb{P}_{x^{(n)}}} \perp\!\!\!\perp [f_C]_{\#\mathbb{P}_{x^{(n)}}}, \forall n \in [N], \qquad \text{(block-indep. enforcing)} \tag{2b}$$

where $f_C(x^{(n)}) \in \mathbb{R}^{d_C}$ represents the first $d_C$ outputs of $f$ that are designated to represent the content components, $f_S(x^{(n)}) \in \mathbb{R}^{d_S}$ represents the learned style from domain $n$, Eq. (2a) matches the distributions of $f_C(x^{(i)})$ and $f_C(x^{(j)})$—i.e., the learned contents from domains $i$ and $j$, respectively—and Eq. (2b) imposes a block independence constraint on the learned content $f_C(x^{(n)})$ and style $f_S(x^{(n)})$ from each domain following Assumption (3.1).

### 3.1 WARM UP: ENHANCED IDENTIFIABILITY WITH KNOWN LATENT DIMENSIONS

We first show that the content-style identifiability under (1) and known $d_C$ and $d_S$ can be substantially enhanced relative to existing works. We will remove the need for the dimension knowledge in the next subsection. To establish identifiability via solving Problem (2), we make the following assumption:

**Assumption 3.2** (Domain Variability). Let $\mathcal{A} \subseteq \mathcal{Z} := \mathcal{C} \times \mathcal{S}$ be any measurable set that satisfies (i) $\mathbb{P}_{z^{(n)}}[\mathcal{A}] > 0$ for any $n \in [N]$ and (ii) $\mathcal{A}$ cannot be expressed as $\mathcal{B} \times \mathcal{S}$ for any set $\mathcal{B} \subset \mathcal{C}$. Then, there exists a pair of $i_{\mathcal{A}}, j_{\mathcal{A}} \in [N]$ such that the following holds:

$$\mathbb{P}_{z^{(i_{\mathcal{A}})}}[\mathcal{A}] \neq \mathbb{P}_{z^{(j_{\mathcal{A}})}}[\mathcal{A}], . \tag{3}$$

Note that for any $\mathcal{A}$, we only need one pair of $(i_{\mathcal{A}}, j_{\mathcal{A}})$ to satisfy the condition, and the pair can change over different $\mathcal{A}$'s. Essentially, Eq. (3) requires that the styles have sufficiently diverse distributions. This assumption is a standard characterization for the distributional diversity of the domains in the literature; see Xie et al. (2023); Kong et al. (2022) and its variant Timilsina et al. (2024).

Under Assumptions 3.1 and (3.2), denote $\widehat{\boldsymbol{f}}$ as a solution to Problem (2). Then, we have:

**Theorem 3.3** (Identifiability under Known Latent Dimensions). *Under Eq. (1), suppose that Assumptions 3.1 and 3.2 hold, and that the $\widehat{\boldsymbol{f}}$ is differentiable. Then, we have $\widehat{\boldsymbol{f}}_{\mathrm{C}}(\boldsymbol{x}^{(n)}) = \boldsymbol{\gamma}(\boldsymbol{c})$ and $\widehat{\boldsymbol{f}}_{\mathrm{S}}(\boldsymbol{x}^{(n)}) = \boldsymbol{\delta}(\boldsymbol{s}^{(n)}), \forall n \in [N]$, where $\boldsymbol{\gamma} : \mathcal{C} \to \mathbb{R}^{d_C}$ and $\boldsymbol{\delta} : \mathcal{S} \to \mathbb{R}^{d_S}$ are injective functions.*

The proof of Theorem 3.3 is in Appendix C. Theorem 3.3 purports that the solution of Problem (2) identifies the model (1)—including the content/style components and the inverse mapping of the generative function $\boldsymbol{g}$ (up to $\boldsymbol{\gamma}$ and $\boldsymbol{\delta}$). Theorem 3.3 uses conditions that are significantly more relaxed relative to those in existing works Xie et al. (2023); Sturma et al. (2023); Kong et al. (2022); Timilsina et al. (2024). First, instead of assuming the elements of $\boldsymbol{z}^{(n)} = (\boldsymbol{c}, \boldsymbol{s}^{(n)})$ are statistically independent as in Xie et al. (2023); Sturma et al. (2023); Kong et al. (2022), our proof is based on the assumption that the content and styles are block independent (cf. Assumption 3.1). This block-independence assumption, which is the key for style identifiability, is similar to those in Lyu et al. (2022) and Timilsina et al. (2024)—but the former assumes aligned domains and the latter can only work under linear mixture models (see Theorem B.2 in Appendix B.2). Second, Theorem 3.3 does not need the existence of $N = 2d_{\mathrm{S}} + 1$ domains as in Xie et al. (2023); Kong et al. (2022) (see Theorem B.3 in Appendix B.3)—our result can hold over as few as $N = 2$ domains. As a result, our Theorem 3.3 applies to a considerably wider range of cases relative to those in existing works.

## 3.2 IDENTIFIABILITY WITHOUT DIMENSION KNOWLEDGE

Theorem 3.3 still uses the knowledge of $d_{\mathrm{C}}$ and $d_{\mathrm{S}}$. In this subsection, we propose a modifed learning criterion that does not use the exact dimension information. To proceed, let $\widehat{d}_{\mathrm{C}}$ and $\widehat{d}_{\mathrm{S}}$ denote the user-specified latent dimensions for $\boldsymbol{f}$, i.e., $\boldsymbol{f} : \mathcal{X} \to \mathbb{R}^{\widehat{d}_{\mathrm{C}} + \widehat{d}_{\mathrm{S}}}$, $\boldsymbol{f}_{\mathrm{C}} : \mathcal{X} \to \mathbb{R}^{\widehat{d}_{\mathrm{C}}}$ and $\boldsymbol{f}_{\mathrm{S}} : \mathcal{X} \to \mathbb{R}^{\widehat{d}_{\mathrm{S}}}$. Note that these dimensions need not to be exact. We consider the following learning criterion:

$$\underset{\boldsymbol{f}:\text{ injective}}{\text{minimize}} \quad \sum_{n=1}^{N} \mathbb{E}\left[\left\|\boldsymbol{f}_{\mathrm{S}}\left(\boldsymbol{x}^{(n)}\right)\right\|_0\right] \tag{4a}$$

$$\text{subject to } [\boldsymbol{f}_{\mathrm{C}}]_{\#\mathbb{P}_{\boldsymbol{x}^{(i)}}} = [\boldsymbol{f}_{\mathrm{C}}]_{\#\mathbb{P}_{\boldsymbol{x}^{(j)}}}, \forall i, j \in [N], \tag{4b}$$

$$[\boldsymbol{f}_{\mathrm{S}}]_{\#\mathbb{P}_{\boldsymbol{x}^{(n)}}} \perp\!\!\!\perp [\boldsymbol{f}_{\mathrm{C}}]_{\#\mathbb{P}_{\boldsymbol{x}^{(n)}}}, \forall n \in [N], \tag{4c}$$

Problem (4) minimizes the "effective dimension" of the extracted style component, while satisfying the distribution matching and independence constraints. The idea is to use excessive $\widehat{d}_{\mathrm{C}}$ and $\widehat{d}_{\mathrm{S}}$ so that one has enough dimensions to represent the content and style information. Note that trivial solutions could occur when using over-estimated $\widehat{d}_{\mathrm{C}}$ and $\widehat{d}_{\mathrm{S}}$. For instance, when $\boldsymbol{f}_{\mathrm{C}}$ is a constant function, $\boldsymbol{f}_{\mathrm{S}}$ can still be an injective function of $\boldsymbol{x}^{(n)}$ given large enough $\widehat{d}_{\mathrm{S}}$. This pathological solution satisfies both constraints (4b) and (4c). We use the sparsity objective in (4a) to "squeeze out" the redundant dimensions in $\boldsymbol{f}_{\mathrm{S}}$. This prevents the content information from "leaking" into the learned $\boldsymbol{f}_{\mathrm{S}}$. We formalize this intuition in the following theorem:

**Theorem 3.4** (Identifiability without Dimension Knowledge). *Assume that the conditions in Theorem 3.3 hold. Let $\widehat{\boldsymbol{f}}$ represent a solution of Problem (4) and $\widehat{\boldsymbol{f}}$ is differentiable. Assume the following conditions hold: (a) $\widehat{d}_{\mathrm{C}} \geq d_{\mathrm{C}}$ and $\widehat{d}_{\mathrm{S}} \geq d_{\mathrm{S}}$. (b) $0 < p_{\boldsymbol{z}^{(n)}}(\boldsymbol{z}) < \infty, \forall \boldsymbol{z} \in \mathcal{Z} = \mathcal{C} \times \mathcal{S}, \forall n \in [N]$. Then, there exists injective functions $\boldsymbol{\gamma} : \mathcal{C} \to \mathbb{R}^{\widehat{d}_{\mathrm{C}}}$ and $\boldsymbol{\delta} : \mathcal{S} \to \mathbb{R}^{\widehat{d}_{\mathrm{S}}}, \forall n \in [N]$ such that $\widehat{\boldsymbol{f}}_{\mathrm{C}}(\boldsymbol{x}^{(n)}) = \boldsymbol{\gamma}(\boldsymbol{c})$ and $\widehat{\boldsymbol{f}}_{\mathrm{S}}(\boldsymbol{x}^{(n)}) = \boldsymbol{\delta}(\boldsymbol{s}^{(n)}), \forall n \in [N]$.*

The proof of Theorem 3.4 is in Appendix D. Theorem 3.4 means that using Problem (4), there is no need to know $d_{\mathrm{S}}$ or $d_{\mathrm{C}}$ in advance. Also, note that no extra assumptions on $\boldsymbol{c}$ and $\boldsymbol{s}^{(n)}$ are needed on top of those in Theorem 3.3. Hence, the identifiability result has significant practical implications for content-style identification, where the latent dimension in practice is always hard to acquire.

## 4 IMPLEMENTATION: SPARSITY-REGULARIZED MULTI-DOMAIN GAN

At first glance, a conceptually straightforward realization of the learning criterion in Problem (2) could take the following form:

$$\underset{\boldsymbol{f}:\text{ injective}}{\text{minimize}} \sum_{i=1}^{N}\sum_{j>i}^{N}\mathcal{L}_{\text{DM}}(\boldsymbol{f}_{\text{C}}(\boldsymbol{x}^{(i)}),\boldsymbol{f}_{\text{C}}(\boldsymbol{x}^{(j)})) + \lambda\sum_{i=1}^{N}\mathcal{L}_{\text{indep}}(\boldsymbol{f}_{\text{C}}(\boldsymbol{x}^{(i)}),\boldsymbol{f}_{\text{S}}(\boldsymbol{x}^{(i)})), \quad (5)$$

where the first term and the second term promotes the distribution matching (DM) constraint (2a) and the independence constraint (2b), respectively. Similarly, Problem (4) can be implemented in a straightforward manner by adding a sparsity regularization term to Problem (5).

*Remark* 4.1. Problem (5) is potentially viable but can be costly: Both the LDM modules and the block independence regularization on the learned components often needs rather nontrivial optimization (see (Lyu et al., 2022)). Enforcing $\boldsymbol{f}$ to be injective also needs extra regularization, e.g., autoencoder type regularization (Lyu et al., 2022; Zhu et al., 2017) and entropy-type regularization (Von Kügelgen et al., 2021; Daunhawer et al., 2023).

In light of Remark 4.1, instead of using Problem (5), we reformulate Problems (2) and (4) as follows:

$$\min_{\boldsymbol{q},\boldsymbol{e}_{\text{C}},\boldsymbol{e}_{\text{S}}} \max_{\boldsymbol{d}^{(n)}} \sum_{n=1}^{N}\mathbb{E}\left[\log\left(\boldsymbol{d}^{(n)}\left(\boldsymbol{x}^{(n)}\right)\right) + \log\left(1 - \boldsymbol{d}^{(n)}\left(\boldsymbol{q}\left(\boldsymbol{e}_{\text{C}}(\boldsymbol{r}_{\text{C}}),\boldsymbol{e}_{\text{S}}^{(n)}(\boldsymbol{r}_{\text{S}}^{(n)})\right)\right)\right)\right] \quad (6a)$$

$$\text{subject to } \boldsymbol{e}_{\text{S}}^{(n)}(\boldsymbol{r}_{\text{S}}^{(n)}) \text{ has minimal } \|\boldsymbol{e}_{\text{S}}^{(n)}(\boldsymbol{r}_{\text{S}}^{(n)})\|_0, \ \forall \boldsymbol{r}_{\text{S}}^{(n)}. \quad (6b)$$

The above approximates Problems (2) and (4) when the constraint (6b) is absent and active, respectively. In practice, the sparsity constraint can be approximated using sparsity regularization terms (e.g., $\ell_1$ norm) easily. Denote $\widehat{d}_{\text{C}}$ and $\widehat{d}_{\text{S}}$ are the estimates of $d_{\text{C}}$ and $d_{\text{S}}$, respectively. The idea is to learn invertible nonlinear mappings $\boldsymbol{e}_{\text{C}}$ and $\boldsymbol{e}_{\text{S}}^{(n)}$ that transform independent Gaussian variables (i.e., $\boldsymbol{r}_{\text{C}}$ and $\boldsymbol{r}_{\text{S}}^{(n)}$) to represent content $\boldsymbol{c}$ and style $\boldsymbol{s}^{(n)}$, respectively. Generate $\boldsymbol{r}_{\text{C}} \sim \mathcal{N}(\boldsymbol{0},\boldsymbol{I}_{\widehat{d}_{\text{C}}})$ and construct an invertible $\boldsymbol{e}_{\text{C}}$ such that $\boldsymbol{e}_{\text{C}}(\boldsymbol{r}_{\text{C}}) \in \mathbb{R}^{\widehat{d}_{\text{C}}}$. Similarly, construct invertible $\boldsymbol{e}_{\text{S}}^{(n)}$ such that $\boldsymbol{e}_{\text{S}}^{(n)}(\boldsymbol{r}_{\text{S}}^{(n)}) \in \mathbb{R}^{\widehat{d}_{\text{S}}}$ with $\boldsymbol{r}_{\text{S}}^{(n)} \sim \mathcal{N}(\boldsymbol{0},\boldsymbol{I}_{\widehat{d}_{\text{S}}})$. Then, the content and style are mixed by $\boldsymbol{q}$ to match the distribution of $\boldsymbol{x}^{(n)}$ using a logistic loss (i.e., GAN-type DM). In other words, the formulation looks for $\boldsymbol{e}_{\text{C}}$, $\boldsymbol{e}_{\text{S}}^{(n)}$ and $\boldsymbol{q}$ such that $\mathbb{P}_{\boldsymbol{x}^{(n)}} = \mathbb{P}_{\boldsymbol{q}^{(n)}}$, $\boldsymbol{q}^{(n)} = \boldsymbol{q}(\boldsymbol{e}_{\text{C}}(\boldsymbol{r}_{\text{C}}),\boldsymbol{e}_{\text{S}}^{(n)}(\boldsymbol{r}_{\text{S}}^{(n)}))$, $\forall n \in [N]$. This way, instead of directly learning $\boldsymbol{f}$, we learn the generative process $\boldsymbol{g}$ using $\boldsymbol{q}$. Our next theorem shows that $\boldsymbol{q}$ is indeed the inverse of $\boldsymbol{f}$ (up to some ambiguities).

To proceed, denote $\widehat{\mathcal{C}}$ and $\widehat{\mathcal{S}}^{(n)}$ as the sets representing the range of $\widehat{e}_{\text{C}}$ and $\widehat{e}_{\text{S}}^{(n)}$, respectively. Then, the effective domain of $\widehat{q}$ is $\widehat{\mathcal{C}} \times \widehat{\mathcal{S}}$ where $\widehat{\mathcal{S}} = \cup_n \widehat{\mathcal{S}}^{(n)}$. We show that:

**Theorem 4.2.** *Let $(\widehat{q},\widehat{e}_{\text{C}},\widehat{e}_{\text{S}}^{(n)},\widehat{d})$ be any differentiable optimal solution of Problem* (6). *Let $\mathcal{C}$ and $\mathcal{S}$ be simply connected open sets. Let $0 < p_{\boldsymbol{z}^{(n)}}(\boldsymbol{z}) < \infty, \forall \boldsymbol{z} \in \mathcal{Z} = \mathcal{C} \times \mathcal{S}$. Under the assumptions in Theorem 3.3, we have the following:*

*(a) If $\widehat{d}_{\text{C}} = d_{\text{C}}$ and $\widehat{d}_{\text{S}} = d_{\text{S}}$ and* (6b) *is absent, then $\widehat{q} : \widehat{\mathcal{C}} \times \widehat{\mathcal{S}} \to \mathcal{X}$ is bijective and $\widehat{f} = \widehat{q}^{-1}$ is also a solution of Problem* (2).

*(b) If $\widehat{d}_{\text{C}} > d_{\text{C}}$, $\widehat{d}_{\text{S}} > d_{\text{S}}$ and $\widehat{q} : \widehat{\mathcal{C}} \times \widehat{\mathcal{S}} \to \mathcal{X}$ is bijective, $\widehat{f} = \widehat{q}^{-1}$ is also a solution of Problem* (4)[1].

Problem (6) has a number of practical advantages over the direct implementation in Problem (5). Particularly, it avoids complex operations in the latent domain. In LDM, performing DM on $\boldsymbol{f}_{\text{C}}(\boldsymbol{x}^{(i)})$ and $\boldsymbol{f}_{\text{C}}(\boldsymbol{x}^{(j)})$ poses quite a nontrivial optimization process. This is because both of the inputs to the DM modules (i.e., $\boldsymbol{f}_{\text{C}}(\boldsymbol{x}^{(i)})$ for all $i \in [N]$) change from iteration to iteration—yet the DM module (e.g., GAN and Wasserstein distance-based DM (Goodfellow et al., 2014; Arjovsky et al., 2017)) itself often involves complex optimization with its own parameters updated on the fly. The new formulation performs GAN-based DM in the *data domain* and keeps one input (the real data) to every GAN module fixed. This reduces a lot of agony in optimization parameter tuning. Problem (6) also does not need any explicit constraint/regularization to enforce the block independence of $\boldsymbol{f}_{\text{C}}$ and

---

[1]Problem (4) requires $\widehat{f}$ to be injective. Here, although the $\widehat{f}$ learned by Problem (6) seems to be bijective (due to $\widehat{f} = \widehat{q}^{-1}$) instead of only injective, the bijectivity is w.r.t. the domains $\mathcal{X} \to \widehat{\mathcal{C}} \times \widehat{\mathcal{S}}$. The function is indeed only injective when considered w.r.t. $\mathcal{X} \to \mathbb{R}^{\widehat{d}_{\text{C}}+\widehat{d}_{\text{S}}}$; see Sec. A.2 "Injection, bijection, and surjection".

Table 1: Evaluation of the data generation task. Standard deviation reported using $\pm$ for style diversity

| Method | FID ($\downarrow$) | | | Style Diversity ($\uparrow$) | | | Training time, hours ($\downarrow$) | | |
|---|---|---|---|---|---|---|---|---|---|
| | AFHQ | CelebA-HQ | CelebA-7 | AFHQ | CelebA-HQ | CelebA-7 | AFHQ | CelebA-HQ | CelebA-7 |
| Transitional-cGAN | 38.00 | 8.12 | 70.45 | – | – | – | 29.65 | 32.56 | 12.53 |
| StyleGAN-ADA | 8.17 | 5.89 | 72.10 | – | – | – | 28.46 | 32.26 | 11.55 |
| I-GAN | 6.28 | 5.91 | **5.18** | $0.16 \pm 0.02$ | $0.07 \pm 0.03$ | $0.07 \pm 0.03$ | 29.53 | 28.76 | 12.51 |
| Proposed | **6.19** | **5.70** | 5.27 | $\mathbf{0.50 \pm 0.03}$ | $\mathbf{0.36 \pm 0.04}$ | $\mathbf{0.26 \pm 0.06}$ | 27.36 | 27.78 | 12.28 |

$\boldsymbol{f}_{\mathrm{S}}$ (which could be resource consuming (Lyu et al., 2022; Gretton et al., 2007)), as $\boldsymbol{e}_{\mathrm{C}}$ and $\boldsymbol{e}_{\mathrm{S}}^{(n)}$ are constructed to be block independent.

Another quite interesting observation is that, the proof of Theorem 4.2 (a) shows that the bijectivity constraint on $\boldsymbol{q}$ is automatically fulfilled when an additional condition (i.e., that $\mathcal{C}$ and $\mathcal{S}$ are simply connected) is met. This means that the LDM formulation would need extra modules, e.g., $\mathbb{E}\|\boldsymbol{r} \circ \boldsymbol{f}(\boldsymbol{x}) - \boldsymbol{x}\|^2$, to impose injectivity constraints, even when $d_{\mathrm{S}}$ and $d_{\mathrm{C}}$ are known. When $d_{\mathrm{C}}$ and $d_{\mathrm{S}}$ are unknown, solving Problem (6) *per se* does not ensure $\boldsymbol{q}$ to be bijective. Nonetheless, we observed that not explicitly enforcing bijectivity in implementations does not affect the performance in practice. Similar phenomenon was observed in nICA implementations; see, e.g., (Hyvarinen & Morioka, 2017; Hyvarinen et al., 2019).

## 5 RELATED WORKS

**Nonlinear ICA.** Learning content and style components from a nonlinear mixture model is reminiscent of *nonlinear independent analysis* (nICA) (Hyvärinen & Pajunen, 1999; Hyvarinen & Morioka, 2017; Hyvarinen et al., 2019). Most nICA works were developed under single domain settings, with some recent generalizations to multiple views/domains (Gresele et al., 2020; Hyvarinen et al., 2019). Nonetheless, nICA requires that all the latent variables are (conditionally) independent. This is considered a somewhat restrictive assumption in content-style learning.

**Content-Style Models in Aligned Multi-Domain Learning.** Aligned multi-domain content-style learning is a key technique in data-augmented SSL and representation learning. There, it was shown that elementwise (conditional) independence is not needed, if the goal is to isolate content from style (Von Kügelgen et al., 2021; Lyu et al., 2022; Karakasis & Sidiropoulos, 2023). It was further shown that block independence (similar to Assumption 3.1) is the key to identify the style (Lyu et al., 2022; Daunhawer et al., 2023). However, all these works require cross-domain data alignment.

**Content-Style Identification in Unaligned Multi-Domain Learning.** Identifiability of unaligned multi-domain learning was studied in the context of various applications, e.g., image translation (Shrestha & Fu, 2024), data synthesis (Xie et al., 2023), cross-domain information retrieval Timilsina et al. (2024), and domain adaptation (Kong et al., 2022; Gulrajani & Hashimoto, 2022; Timilsina et al., 2024). In applications, content-style disentanglement has been applied in various tasks, such as (Hong et al., 2024; Huang et al., 2022; Dai et al., 2023). However, only a handful of works (Kong et al., 2022; Xie et al., 2023; Timilsina et al., 2024) have investigated the identifiability aspects. The work (Kong et al., 2022) postulated a similar content-style model as in (Xie et al., 2023) and came up with identifiability conditions similar to those in (Xie et al., 2023). The mostly related work to ours is (Xie et al., 2023), as both works are interested in content-style identification under (1). Our implementation in Problem (6) partially recovers the marginal distribution matching criterion in (Xie et al., 2023), despite the fact that our learning criteria started with an LDM perspective. Nonetheless, our method enjoys much less restrictive model assumptions for content-style identifiability. Our multi-domain GAN also admits more relaxed neural architecture (see Appendix G).

**Content-Style Learning without Knowing Latent Dimensions.** The SSL work (Von Kügelgen et al., 2021) presented a proof that essentially established that the content can be learned without knowing the exact dimension $d_{\mathrm{C}}$. However, their result was under the assumption that the domains are *aligned*. In addition, the proof could not hold when style learning is also involved. Our proof solved these challenges. The work in (Xie et al., 2023) used a mask-based formulation to remove the requirement of knowing $d_{\mathrm{C}}$ and $d_{\mathrm{S}}$. The mask-based formulation has the flavor of sparsity promoting as in our proposed method. However, they still need to know $d_{\mathrm{C}} + d_{\mathrm{S}}$, which is unlikely available in practice. In addition, the mask-based method in (Xie et al., 2023) did not have theoretical supports.

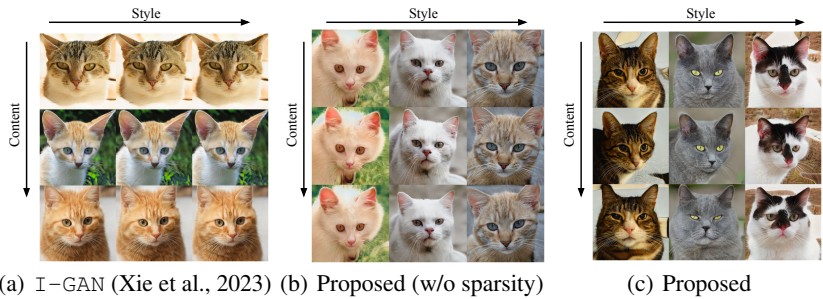

(a) `I-GAN` (Xie et al., 2023)    (b) Proposed (w/o sparsity)    (c) Proposed

Figure 2: Samples generated by learning content (pose of cat) and style (type of cat) from AFHQ.

# 6   NUMERICAL EXAMPLES

**Multi-Domain Data Generation.** For the data generation task, we validate our theoretical claims using three real world datasets: animal faces (AFHQ) (Choi et al., 2020), CelebA-HQ (Karras et al., 2018), and CelebA (Liu et al., 2015) with 3, 2, and 7 domains, respectively (see Appendix G.6).

The baselines here are `I-GAN` (Xie et al., 2023), `StyleGAN-ADA` (Karras et al., 2020) and `Transitional-cGAN` (Shahbazi et al., 2021).

Following Xie et al. (2023), we use StyleGAN2-ADA (Karras et al., 2020) to represent our generative function $q$ in (6a). We set $\widehat{d}_C = 384$ and $\widehat{d}_S = 128$ in all the experiments. We use an $\ell_1$ regularization term $\lambda\|e_S^{(n)}(r_S^{(n)})\|_1$ to approximate the sparsity constraint in (6b).

Note that other sparsity-promoting regularization (such as the $\ell_p$ function with $p < 1$) can also be easily used under our framework, which shows similar effectiveness (see Appendix H.4). We find that the algorithm is not very sensitive to the choice $\lambda$ as any positive $\lambda$ encourages sparsity of $e_S^{(n)}(r_S^{(n)})$. We use $\lambda = 0.3$ for all the experiments. More detailed experimental settings are in Appendix G.

Fig. 2 shows the qualitative results for content-style identification using various methods for the cat domain ($n = 1$) of AFHQ. For each row, we fix the content part $c = e_C(r_C)$ (i.e., pose of the cat) and randomly sample different styles $s^{(1)} = e_S^{(1)}(r_S^{(1)})$ where $r_S^{(1)} \sim \mathcal{N}(0, I_{\widehat{d}_S})$ to generate the images $x^{(1)} = q(c, s^{(1)})$. This way, the samples $s_i^{(1)}$ for $i = 1, 2, \ldots$

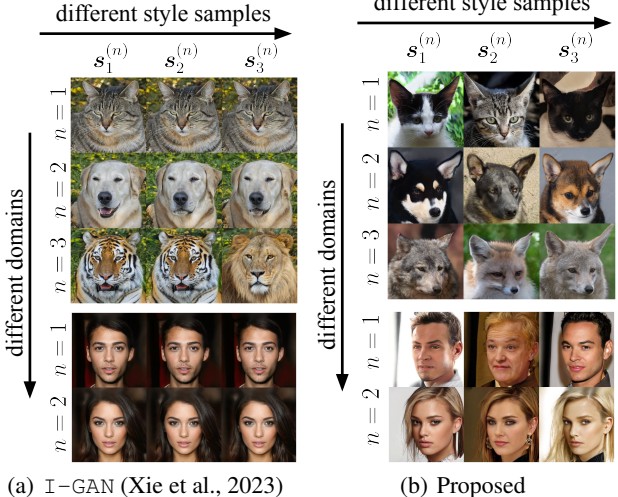

(a) `I-GAN` (Xie et al., 2023)     (b) Proposed

Figure 3: Samples generated by combining the same content $\overline{c}$ with $s^{(n)}$ for various $n$'s in AFHQ and CelebA-HQ.

correspond to various types of cats. Fig.2 (a) shows that the `I-GAN` appears to generate the same type of cat even when repeatedly sampled from their learned distribution of $s^{(1)}$. This suggests that the style components are not extracted properly. Fig. 2 (b) shows the result of using the proposed method without any sparsity regularization. As explained earlier, this can lead to learning constant content part with all information captured by the style part. Fig. 2(b) corroborates with the intuition since we see little to no pose variation in the sampled contents (i.e., the three rows). Fig. 2 (c) shows the result of proposed method, i.e., Problem (6). One can see that both content and style parts demonstrate sufficient diversity, indicating well learned content and style distributions. Appendix H shows similar results for other domains and datasets.

Fig. 3 shows the generated samples of $x^{(n)}$ for different $n$'s using models learned from the AFHQ and CelebA-HQ datasets, which correspond to different species of animals (cat, dog, and tiger) and different genders of people, respectively. The top three rows in each figure correspond to the three

$n$'s (i.e., three domains) of the AFHQ dataset, whereas the bottom two rows correspond to two $n$'s of the CelebA-HQ dataset.

For the $j$th row associated with each dataset, we sample three different styles $\boldsymbol{s}_i^{(n_j)}, i \in \{1, 2, 3\}$ and combine it with a fixed content $\overline{\boldsymbol{c}}$ to generate the image $\boldsymbol{x}_i^{(n_j)} = \boldsymbol{q}(\overline{\boldsymbol{c}}, \boldsymbol{s}_i^{(n_j)})$ in the $j$th row and $i$th column.

Both the baseline I-GAN and our method can combine a fixed $\bar{\boldsymbol{c}}$ with $\boldsymbol{s}_i^{(n_j)}$ for different $i$ to create content (pose)-consistent new data (see all the rows). However, one can see that the baseline I-GAN was not able to sample different styles in each domain. It seems that every domain $n$ always repeatedly samples the same style components $\bar{\boldsymbol{s}}^{(n)}$ as the same images always appear in the same row. The proposed method can generate quite diverse style samples in all the domains. Additional results are in the Appendix H.

Table 1 shows the FID (Heusel et al., 2017), style diversity scores, and training time of the different methods. We use LPIPS distance (Zhang et al., 2018) between pairs of images with the same $\bar{\boldsymbol{c}}$ and different style samples from $\boldsymbol{s}^{(n)}$ to measure the style diversity. The diversity scores are averaged over 6,000 images across all domains, where every 10 images contain the same content with different styles. Note that the baselines StyleGAN-ADA and Transitional-cGAN do not learn content-style models, and thus the style diversity scores of theirs are not reported. One can see that the FID scores of the methods are similar, meaning that all methods generate realistic looking images. However, the style diversity of the proposed method is 3 to 5 times higher than the baseline over all datasets. The conditional generative models (Transitional-cGAN and

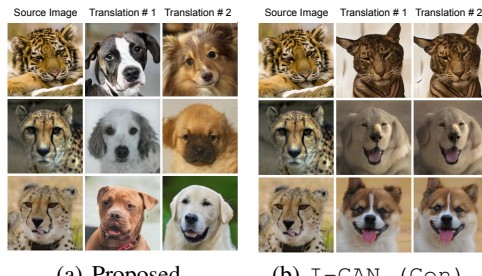

(a) Proposed          (b) I-GAN (Gen)

Figure 4: Translation by combining content (pose) randomly sampled styles from the dog domain.

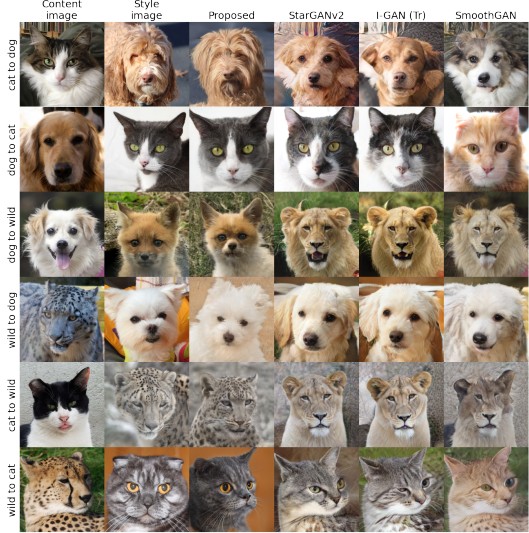

Figure 5: Guided translation by combining content (first column) with style (second column) of the images.

StyleGAN-ADA) sometimes encountered convergence issues on specific datasets as reflected by their FID scores. Finally, the training time of all the methods are in the similar range, the proposed method being slightly faster for AFHQ and CelebA-HQ datasets.

**Multi-Domain Translation.**

Existing methods use a dedicated system for multi-domain translation (Choi et al., 2018; 2020; Yang et al., 2023). However, since a multi-domain generative model can already disentangle content and style (cf. Theorems-4.2 of this work), one can simply use the generative model for domain translation.

Given an image $\boldsymbol{x}^{(i)}$ in the source domain $i$, in order to extract the corresponding content $\boldsymbol{c}$ or style $\boldsymbol{s}^{(i)}$, one can simply solve $(\widehat{\boldsymbol{c}}, \widehat{\boldsymbol{s}}) = \arg\min_{\boldsymbol{c}, \boldsymbol{s}} \text{div}(\boldsymbol{q}(\boldsymbol{c}, \boldsymbol{s}), \boldsymbol{x}^{(i)})$, where div is some distance metric/divergence measure. There exists many approaches to solving the problem, often referred to as GAN inversion (Xia et al., 2022). In our case, we simply use the Adam optimizer for this inversion step. For div, we use a pre-trained VGG16 (Simonyan & Zisserman, 2014) neural network. More details are in Appendix G. To generate the desired translation, the GAN inversion-extracted content can be combined with a randomly sampled style $\boldsymbol{s}^{(t)} = \boldsymbol{e}_{\mathrm{S}}^{(t)}(\boldsymbol{r}_{\mathrm{S}}^{(t)}), \boldsymbol{r}_{\mathrm{S}}^{(t)} \sim \mathcal{N}(\boldsymbol{0}, \boldsymbol{I}_{\widehat{d}_{\mathrm{S}}})$ from the target domain $t$. Additionally, one can also extract style from an image in target domain and combine it with extracted content from the source domain for guided translation.

Table 2: Quantitative evaluation of all methods for the translated images.

| Method | FID ($\downarrow$) | | Style Diversity ($\uparrow$) | | Training time ( hours) | | |
|---|---|---|---|---|---|---|---|
| | AFHQ | CelebA-HQ | AFHQ | CelebA-HQ | Generation | Translation | Total |
| StarGANv2 | 16.83 | **13.67** | $0.45 \pm 0.03$ | **$0.45 \pm 0.03$** | – | 50.83 | 50.83 |
| SmoothGAN | 53.68 | 29.69 | $0.14 \pm 0.04$ | $0.09 \pm 0.03$ | – | 30.90 | 30.90 |
| I-GAN (Tr) | 19.57 | 15.26 | $0.46 \pm 0.03$ | $0.29 \pm 0.05$ | 29.53 | 67.46 | 96.99 |
| Proposed | **13.74** | 16.61 | **$0.53 \pm 0.03$** | $0.41 \pm 0.03$ | 27.36 | – | **27.36** |

The baselines used are the method in (Xie et al., 2023), StarGANv2 (Choi et al., 2020), and SmoothGAN (Liu et al., 2021). Note that (Xie et al., 2023) proposed a separate system for the domain translation that uses its pre-trained multi-domain generative model to train a separate translation model (see Appendix F). However, since the aforementioned GAN inversion procedure is also applicable to their generative model as it extracts content and style, we use two versions of their system, namely, I-GAN (Gen) for the method based on GAN inversion and I-GAN (Tr) for the separate translation system proposed in (Xie et al., 2023).

Fig. 4 (a) and (b) show the result of translation from wild domain ($n = 3$) to dog domain ($n = 2$) using randomly sampled style components. The content $\widehat{c}_i^{(3)}, i \in [3]$ extracted for samples in the wild domain is combined with randomly sampled styles $s_j^{(2)}, j \in [2]$ in the dog domain to synthesize the translated images. Our translations in each row contain the same content (i.e., pose of wild) as the input source image, but different styles (i.e., dog species). However, I-GAN (Gen) seems to produce unrealistic samples in some cases (first row). Their style diversity also appears to be limited.

Fig. 5 shows results of guided-translation for all methods for all pairs of domains in the AFHQ domain. Content extracted from the images in the first column is combined with the style from the second column. One can see that the proposed method preserves the style information better than the baselines.

Further experiments on multi-domain translation are presented in Appendix H.2.

Table 2 shows that the image quality (see FID) and diversity (see style diversity) of the translated images are competitive or better than the baselines (see qualitative results in Fig. 9 and 10 of Appendix H.). One can also see that the training time (on a single Tesla V100 GPU) of proposed method is at least 22 and 69 hours shorter than the competitive baselines StarGANv2 and I-GAN (Tr), respectively.

## 7 CONCLUSION

We revisited the problem of content-style identification from unaligned multi-domain data, which is a key step for provable domain translation and data generation. We offered a LDM perspective. This new viewpoint enabled us to prove identifiability results that enjoy considerably more relaxed conditions compared to those in previous research. Most importantly, we proved that content and style can be identified without knowing the exact dimension of the latent components. To our knowledge, this stands as the first dimension-agnostic identifiability result for content-style learning. We showed that the LDM formulation is equivalent to a latent domain-coupled multi-domain GAN loss, and the latter features a simpler implementation in practice. We validated our theorems using image translation and generation tasks.

**Limitations.** Our work focused on sufficient conditions for content-style identifiability, yet the necessary conditions were not fully understood—which is also of great interest. Additionally, our model considers that the domains are in the range of the same generating function. The applicability is limited to homogeneous multi-domain data, e.g., images with the same resolution. An interesting extension is to consider heterogeneous multi-domain models that can deal with very different types of data (e.g., text and audio). Additionally, our work is also limited to continuous data modalities like images, audio, etc. Discrete data modalities like text will require extension of both theory and implementation. This challenge presents another important future work. Finally, another limitation of our work is that the proposed method is based on the GAN framework which is known to be unstable during training. Therefore, novel implementation methods based on more stable distribution matching modules such as flow matching are also of interest as a future work.

ACKNOWLEDGMENT

This work was supported in part by the National Science Foundation (NSF) CAREER Award ECCS-2144889, and in part by Army Research Office (ARO) under Project ARO W911NF-21-1-0227.

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

# Supplementary Material of "Content-Style Learning from Unaligned Domains: Identifiability under Unknown Latent Dimensions"

## A   PRELIMINARIES

### A.1   NOTATION

- A set $\mathcal{A}$ is said to have strictly positive measure under $\mathbb{P}_{\boldsymbol{x}}$ if and only if $\mathbb{P}_{\boldsymbol{x}}[\mathcal{A}] > 0$.
- For a (random) vector $\boldsymbol{x}$, $x(i)$ and $[\boldsymbol{x}]_i$ denote the $i$th element of $\boldsymbol{x}$, and $\boldsymbol{x}(i:j)$ and $[\boldsymbol{x}]_{i:j}$ denotes $[x(i), x(i+1), \ldots, x(j)]$.
- The notation $[N] = \{1, 2, \ldots, N\}$.
- $\boldsymbol{I}_{\mathrm{D}} \in \mathbb{R}^{D \times D}$ denotes identity matrix of size $D \times D$.
- For a differentiable function $\boldsymbol{f} : \mathbb{R}^M \to \mathbb{R}^N$, the Jacobian of $\boldsymbol{f}$ with respect to its input $\boldsymbol{x}$, denoted by $\boldsymbol{J}_{\boldsymbol{f}}(\boldsymbol{x})$, is the matrix of partial derivatives as follows:

$$\boldsymbol{J}_{\boldsymbol{f}}(\boldsymbol{x}) = \begin{bmatrix} \frac{\partial [\boldsymbol{f}(\boldsymbol{x})]_1}{\partial x_1} & \cdots & \frac{\partial [\boldsymbol{f}(\boldsymbol{x})]_1}{\partial x_M} \\ \vdots & \ddots & \vdots \\ \frac{\partial [\boldsymbol{f}(\boldsymbol{x})]_N}{\partial x_1} & \cdots & \frac{\partial [\boldsymbol{f}(\boldsymbol{x})]_N}{\partial x_M} \end{bmatrix}$$

- $\det(\boldsymbol{X})$ denotes the determinant of matrix $\boldsymbol{X}$.
- $\boldsymbol{x} \perp\!\!\!\perp \boldsymbol{y}$ denotes that $\boldsymbol{x}$ is statistically independent of $\boldsymbol{y}$.
- $|x|$ denotes the absolute value of $x$
- $\|\boldsymbol{x}\|_0$ denotes the number of non-zero elements in $\boldsymbol{x}$.
- $\epsilon$-**ball**: Given a metric space $\mathcal{X} \subseteq \mathbb{R}^D$ for some $D \in \mathbb{N}$, $\mathcal{N}_\epsilon(\boldsymbol{z})$ denotes the $\epsilon$-neighborhood of $\boldsymbol{z} \in \mathcal{X}$ defined as
$$\mathcal{N}_\epsilon(\boldsymbol{z}) = \{\widehat{\boldsymbol{z}} \in \mathcal{X} \mid \|\boldsymbol{z} - \widehat{\boldsymbol{z}}\|_2 < \epsilon\}.$$
- **Image set**: For a function $\boldsymbol{m} : \mathcal{W} \to \mathcal{Z}$, and a set $\mathcal{A} \subseteq \mathcal{W}$, $\boldsymbol{m}(\mathcal{A}) = \{\boldsymbol{m}(\boldsymbol{w}) \in \mathcal{Z} \mid \boldsymbol{w} \in \mathcal{A}\}$
- For an injective function $\boldsymbol{f} : \mathcal{X} \to \mathcal{Y}$, we denote $\boldsymbol{f}^{-1} : \boldsymbol{f}(\mathcal{X}) \to \mathcal{Y}$ as the left inverse of $\boldsymbol{f}$, i.e., $\boldsymbol{f}^{-1} \circ \boldsymbol{f}(\boldsymbol{x}) = \boldsymbol{x}, \forall \boldsymbol{x} \in \mathcal{X}$.
- **Pre-Image set**: For a general function $\boldsymbol{m} : \mathcal{W} \to \mathcal{Z}$, we overload the notation $\boldsymbol{m}^{-1}$ to represent the pre-image; i.e., for a set $\mathcal{A} \subseteq \mathcal{Z}$, $\boldsymbol{m}^{-1}(\mathcal{A}) = \{\boldsymbol{w} \in \mathcal{W} \mid \boldsymbol{m}(\boldsymbol{w}) \in \mathcal{A}\}$. If $\boldsymbol{m}$ is injective, then $\boldsymbol{m}^{-1}$ is simply the left inverse of $\boldsymbol{m}$; i.e., $\boldsymbol{m}^{-1} \circ \boldsymbol{m}$ is identity.
- For a function $\boldsymbol{f} : \mathcal{X} \to \mathcal{Y}$ and a probability measure $\mathbb{P}_{\boldsymbol{x}}$ defined on $\mathcal{X}$, $[\boldsymbol{f}]_{\#\mathbb{P}_{\boldsymbol{x}}}$ denotes the **push forward** measure defined by $[\boldsymbol{f}]_{\#\mathbb{P}_{\boldsymbol{x}}}[\mathcal{A}] = \mathbb{P}_{\boldsymbol{x}}[\boldsymbol{f}^{-1}(\mathcal{A})]$ for any measurable $\mathcal{A} \subseteq \mathcal{Y}$.
- $\dim(\mathcal{X})$ denotes the covering dimensio of the set $\mathcal{X}$ (see Appendix A.2).

### A.2   DEFINITIONS AND USEFUL FACTS

We employ standard definitions and facts from real analysis. We refer the readers to (Carothers, 2000; Rudin, 1976) for precise definition and more details.

**Injection, bijection, and surjection.**

Consider a function $\boldsymbol{f} : \mathcal{X} \to \mathcal{Y}$. Then

- $\boldsymbol{f}$ is *injective*, if for all $\boldsymbol{x}_1, \boldsymbol{x}_2 \in \mathcal{X}$, $\boldsymbol{x}_1 \neq \boldsymbol{x}_2 \implies \boldsymbol{f}(\boldsymbol{x}_1) \neq \boldsymbol{f}(\boldsymbol{x}_2)$.
- $\boldsymbol{f}$ is *surjective*, if for all $\boldsymbol{y} \in \mathcal{Y}$, $\exists \boldsymbol{x} \in \mathcal{X}$ such that $\boldsymbol{f}(\boldsymbol{x}) = \boldsymbol{y}$.
- $\boldsymbol{f}$ is *bijective* or *invertible*, if $\boldsymbol{f}$ is both injective and surjective.

Also consider the following useful facts:

1. $\boldsymbol{f}$ is injective if and only if there exist a function $\boldsymbol{g} : \boldsymbol{f}(\mathcal{X}) \to \mathcal{X}$ such that $\boldsymbol{g} \circ \boldsymbol{f}(\boldsymbol{x}) = \boldsymbol{x}, \forall \boldsymbol{x} \in \mathcal{X}$, i.e., $\boldsymbol{g} \circ \boldsymbol{f}$ is an identity function.

2. If $\boldsymbol{f} : \mathcal{X} \to \mathcal{Y}$ is injective, then $\boldsymbol{f} : \mathcal{X} \to \boldsymbol{f}(\mathcal{X})$ is bijective.

3. If $\boldsymbol{f} \circ \boldsymbol{g}$ is injective, and $\boldsymbol{g}$ is surjective, then $\boldsymbol{f}$ is injective.

4. If $\boldsymbol{f} \circ \boldsymbol{g}$ is bijective, then $\boldsymbol{g}$ is surjective and $\boldsymbol{f}$ is injective.

5. If $\boldsymbol{f} : \mathcal{X} \to \mathcal{Y}$ is bijective, then $\boldsymbol{f} : \mathcal{X} \to \mathcal{Z}$ is injective, where $\mathcal{Y} \subset \mathcal{Z}, \mathcal{Y} \neq \mathcal{Z}$.

An important remark is that since injective functions can be "inverted" (fact 1 above), there is no loss of information of the input, i.e., the input information can "reconstructed" from the output. Hence, for the purpose of identifiability, it is sufficient to identify the latent variables upto injective transformations.

**Simply Connected Sets.**

First, a set $\mathcal{C}$ (in $\mathcal{X}$) is *connected* if and only if there does not exist any disjoint non-empty open sets $\mathcal{A}, \mathcal{B} \subseteq \mathcal{X}$ such that $\mathcal{A} \cap \mathcal{C} \neq \phi, \mathcal{B} \cap \mathcal{C} \neq \phi$, and $\mathcal{C} \subset \mathcal{A} \cup \mathcal{B}$.

A connected set is called *simply connected* if any simple closed curve on the set can be shrunk to a point continuously while staying inside the set.

**Homeomorphism.**

A function $\boldsymbol{f} : \mathcal{X} \to \mathcal{Y}$ is a *homeomorphism* if it is a continuous bijection and has continuous inverse. If there exists a homeomorphism between $\mathcal{X}$ and $\mathcal{Y}$, then $\mathcal{X}$ and $\mathcal{Y}$ are called homeomorphic.

Some useful properties are as follows:

1. $\mathbb{R}^M$ is not homeomorphic to $\mathbb{R}^D$ for $D \neq M$.

2. Open balls in $\mathbb{R}^D$ are homeomorphic to $\mathbb{R}^D$.

3. *Invariance of Domain*: If $\mathcal{X}$ is an open subset of $\mathbb{R}^D$ and $\boldsymbol{f} : \mathcal{X} \to \mathbb{R}^D$ is an injective continuous map, then $\mathcal{Y} = \boldsymbol{f}(\mathcal{X})$ is open in $\mathbb{R}^D$, and $\boldsymbol{f}$ is a homeomorphism between $\mathcal{X}$ and $\mathcal{Y}$.

**Covering dimension** (Engelking, 1978).

An *open cover* of a set $\mathcal{X}$ is a collection of open sets such that $\mathcal{X}$ lies inside their union.

A *refinement* of a cover $\mathcal{Y}$ of a set $\mathcal{X}$ is a new cover $\mathcal{Z}$ of $\mathcal{X}$ such that every set in $\mathcal{Z}$ is contained in some set in $\mathcal{Y}$.

The *(Lebesgue) covering dimension*, denoted by $\dim(\mathcal{X})$, of a set $\mathcal{X}$ is the smallest number $n$ such that for every open cover, there is a refinement in which every point in $\mathcal{X}$ lies in the intersection of no more than $n + 1$ covering sets. For our purposes (subsets of Euclidean spaces), the covering dimension is the ordinary Euclidean deimension, e.g., one for lines and curves, two for planes, and so on.

Some propoerties of the covering dimension are as follows:

1. Homeomorphic spaces have the same covering dimension.

2. If $\mathcal{X}$ is an open subset in $\mathbb{R}^D$, then $\dim(\mathcal{X}) = D$.

3. For two open sets $\mathcal{X} \subseteq \mathbb{R}^D, \mathcal{Y} \subseteq \mathbb{R}^M, \dim(\mathcal{X} \times \mathcal{Y}) = \dim(\mathcal{X}) + \dim(\mathcal{Y})$.

4. For a continuous funtion $\boldsymbol{f}$ and open set $\mathcal{X}, \dim(\boldsymbol{f}(\mathcal{X})) \leq \dim(\mathcal{X})$.

5. The set $\mathcal{A}_q \subset \mathbb{R}^D$ of $q$-sparse vectors, i.e., $\mathcal{A}_q = \{\boldsymbol{x} \in \mathbb{R}^D | \|\boldsymbol{x}\|_0 \leq q\}$ has dimension $\dim(\mathcal{A}_q) = q$. This is because it is the union of a finite number $\binom{D}{q}$ of $q$-dimensional hyperplanes (i.e., hyperplanes with all elements 0, except q elements at fixed positions).

## B    DETAILED DISCUSSIONS ON EXISTING IDENTIFIABILITY RESULTS

In this section, we review the existing content-style identifiability results from Sturma et al. (2023); Timilsina et al. (2024); Kong et al. (2022); Xie et al. (2023).

## B.1 IDENTIFIABILITY RESULT IN STURMA ET AL. (2023)

The works of (Timilsina et al., 2024; Sturma et al., 2023) considered a linear mixture-based generative model as follows:

$$\boldsymbol{x}^{(n)} = \boldsymbol{G}^{(n)} \boldsymbol{z}^{(n)}, \quad \boldsymbol{z}^{(n)} = [\boldsymbol{c}^\top, (\boldsymbol{s}^{(n)})^\top]^\top, \ n \in \{1, 2\}, \tag{7}$$

where $\boldsymbol{c} \in \mathbb{R}^{d_C}$, $\boldsymbol{s}^{(n)} \in \mathbb{R}^{d_S^{(n)}}$, and $\boldsymbol{G}^{(n)} \in \mathbb{R}^{(d_C + d_S^{(n)}) \times (d_C + d_S^{(n)})}$ is an invertible mixing matrix for the $n$th domain.

> **Theorem B.1** (Identifiability from Sturma et al. (2023)). *Under (7), assume that the following are met: (i) The conditions for ICA identifiability (Comon, 1994) is met by each domain, including that the components of $\boldsymbol{z}^{(n)} = [\boldsymbol{c}^\top, (\boldsymbol{s}^{(n)})^\top]^\top$ are mutually statistically independent and contain at most one Gaussian variable. In addition, each $z_i^{(n)}$ has unit variance; (ii) $\mathbb{P}_{z_i^{(n)}} \neq \mathbb{P}_{z_j^{(n)}}, \mathbb{P}_{z_i^{(n)}} \neq \mathbb{P}_{-z_j^{(n)}} \ \forall i, j \in [d_C + d_S^{(n)}], \ i \neq j$. Then, assume that $(i_m, j_m)$ are obtained by ICA followed by cross domain matching distribution matching, i.e., enforcing $\widehat{c}_m^{(1)} \overset{(d)}{=\!=} \widehat{c}_m^{(2)}$ for $m = 1, \ldots, d_C$. Denote $\widehat{c}_m^{(1)} = \boldsymbol{e}_{i_m}^\top \widehat{\boldsymbol{z}}^{(1)}$ and $\widehat{c}_m^{(2)} = \boldsymbol{e}_{j_m}^\top \widehat{\boldsymbol{z}}^{(2)}$. We have the following:*
>
> $$\widehat{c}_m^{(n)} = k c_{\boldsymbol{\pi}(m)}^{(n)}, \ m \in [d_C], \tag{8}$$
>
> *where $k \in \{+1, -1\}$ and $\boldsymbol{\pi}$ is a permutation of $\{1, \ldots, d_C\}$.*

Obviously, the result here relies on the element-wise independence among the entries of $\boldsymbol{z}^{(n)}$.

## B.2 IDENTIFIABILITY RESULT IN TIMILSINA ET AL. (2024)

Timilsina et al. (2024) proposed to solve the following problem in order to extract the content and style for model in (7):

$$\text{find} \quad \boldsymbol{Q}_C^{(n)} \in \mathbb{R}^{d_C \times (d_C + d_S^{(n)})}, \boldsymbol{Q}_S^{(n)} \in \mathbb{R}^{d_S^{(n)} \times (d_C + d_S^{(n)})} \ n = 1, 2, \tag{9a}$$

$$\text{subject to} \quad \boldsymbol{Q}_C^{(1)} \boldsymbol{x}^{(1)} \overset{(d)}{=\!=} \boldsymbol{Q}_C^{(2)} \boldsymbol{x}^{(2)}, \tag{9b}$$

$$\boldsymbol{Q}_C^{(n)} \boldsymbol{x}^{(n)} \perp\!\!\!\perp \boldsymbol{Q}_S^{(n)} \boldsymbol{x}^{(n)} \quad n = 1, 2, \tag{9c}$$

$$\boldsymbol{Q}_C^{(n)} \mathbb{E}\left[\boldsymbol{x}^{(n)} (\boldsymbol{x}^{(n)})^\top\right] (\boldsymbol{Q}_C^{(n)})^\top = \boldsymbol{I} \quad n = 1, 2, \tag{9d}$$

$$\boldsymbol{Q}_S^{(n)} \mathbb{E}\left[\boldsymbol{x}^{(n)} (\boldsymbol{x}^{(n)})^\top\right] (\boldsymbol{Q}_S^{(n)})^\top = \boldsymbol{I} \quad n = 1, 2, \tag{9e}$$

where $a \overset{(d)}{=\!=} b$ denotes that the distribution of $a$ and $b$ are the same. The following identifiability result was established of the solution of (9):

> **Theorem B.2** (Identifiability from Timilsina et al. (2024)). *Let $\widehat{\boldsymbol{Q}}_C^{(n)}$ and $\widehat{\boldsymbol{Q}}_S^{(n)}$ denote the solution of (9). Under (7), assume that the following are met:*
>
> 1. *For any two linear subspaces $\mathcal{P}^{(n)} \subset \mathbb{R}^{d_C + d_S^{(n)}}$, $n = 1, 2$, with $\dim(\mathcal{P}^{(n)}) = d_S^{(n)}$, $\mathcal{P}^{(n)} \neq \boldsymbol{0} \times \mathbb{R}^{d_S^{(n)}}$ and linearly independent vectors $\{\boldsymbol{y}_i^{(n)} \in \mathbb{R}^{d_C + d_S^{(n)}}\}_{i=1}^{d_C}$, $n = 1, 2$, the sets $\mathcal{A}^{(n)} = \mathrm{conv}\{\boldsymbol{0}, \boldsymbol{y}_1^{(n)}, \ldots, \boldsymbol{y}_{d_C}^{(n)}\} + \mathcal{P}^{(n)}$, $n = 1, 2$, are such that if $\mathbb{P}_{\boldsymbol{c}, \boldsymbol{s}^{(n)}}[\mathcal{A}^{(n)}] > 0$ for $n = 1$ or $n = 2$, then there exists a $k \in \mathbb{R}$ such that the joint distributions $\mathbb{P}_{\boldsymbol{c}, \boldsymbol{s}^{(1)}}[k \mathcal{A}^{(1)}] \neq \mathbb{P}_{\boldsymbol{c}, \boldsymbol{s}^{(2)}}[k \mathcal{A}^{(2)}]$, where $k \mathcal{A}^{(n)} = \{k \boldsymbol{a} \mid \boldsymbol{a} \in \mathcal{A}^{(n)}\}$.*
>
> 2. *One of the following assumptions is satisfied:*
>
>    (a) *The individual elements of the content components are statistically independent and non-Gaussian. In addition, $c_i \overset{(d)}{\neq} k c_j, \forall i \neq j, \forall k \in \mathbb{R}$ and $c_i \overset{(d)}{\neq} -c_i, \forall i$, i.e., the marginal distributions of the content elements cannot be matched with each other by mere scaling.*

> *(b) The support $\mathcal{C}$, is a hyper-rectangle, i.e., $\mathcal{C} = [-a_1, a_1] \times \cdots \times [-a_{d_\mathrm{C}}, a_{d_\mathrm{C}}]$. Further, suppose that $c_i^{(d)} \neq kc_j^{(d)}, \forall i \neq j, \forall k \in \mathbb{R}$ and $c_i^{(d)} \neq -c_i, \forall i$.*
>
> *Then, we have $\widehat{\boldsymbol{Q}}_\mathrm{C}^{(n)} \boldsymbol{x}^{(n)} = \boldsymbol{\Theta}\boldsymbol{c}$ and $\widehat{\boldsymbol{Q}}_\mathrm{S}^{(n)} \boldsymbol{x}^{(n)} = \boldsymbol{\Xi}^{(n)} \boldsymbol{s}^{(n)}$, for some invertible $\boldsymbol{\Xi}^{(n)}$ for all $n = 1, 2$.*

The first condition is similar to the domain variability condition considered in Kong et al. (2022); Xie et al. (2023) and our Assumption 3.2, but with a more specific form that is adjusted according to the linear mixture model in (7).

## B.3 IDENTIFIABILITY RESULT IN XIE ET AL. (2023); KONG ET AL. (2022)

Note that the results in Sec. B.1 and B.2 are only applicable in the linear mixing system case (see (7)). However, the linear model is hardly practical for complex data modalities like images. In the nonlinear case, Xie et al. (2023); Kong et al. (2022) showed the following:

> **Theorem B.3** (Identifiability from (Xie et al., 2023; Kong et al., 2022)). *Denote $\boldsymbol{z} = (\boldsymbol{c}, \boldsymbol{s}) \in \mathcal{C} \times \mathcal{S}$ and $\boldsymbol{u}$ as the latent variables and an auxiliary variable[a], respectively. Assume the following assumptions hold:*
>
> *(A1) (Smooth and Positive Density): $\log p(\boldsymbol{z}|\boldsymbol{u})$ is second-order differentiable and $p(\boldsymbol{z}|\boldsymbol{u}) > 0$.*
>
> *(A2) (Conditional independence): $p(\boldsymbol{z}|\boldsymbol{u}) = \prod_{i=1}^{d_\mathrm{S}+d_\mathrm{C}} p(z_i|\boldsymbol{u})$.*
>
> *(A3) (Linear independence): For any $\boldsymbol{s} \in \mathcal{S} \subseteq \mathbb{R}^{d_\mathrm{S}}$, there exists $2d_\mathrm{S} + 1$ values of $\boldsymbol{u}$, i.e., $\boldsymbol{u}_j$ with $j = 0, 1, \ldots, 2d_\mathrm{S}$ such that the $2d_\mathrm{S}$ vectors $\boldsymbol{w}(\boldsymbol{s}, \boldsymbol{u}_j) - \boldsymbol{w}(\boldsymbol{s}, \boldsymbol{u}_0)$ are linearly independent, where $\boldsymbol{w}(\boldsymbol{s}, \boldsymbol{u}) = \left( \partial q_1/\partial \boldsymbol{s}_1, \ldots, \partial q_{d_\mathrm{S}}/\partial \boldsymbol{s}_{d_\mathrm{S}}, \partial^2 q_1/\partial \boldsymbol{s}_1^2, \ldots, \partial^2 q_{d_\mathrm{S}}/\partial \boldsymbol{s}_{d_\mathrm{S}}^2 \right)$ and $q_i = \log p(s_i|\boldsymbol{u})$.*
>
> *(A4) (Domain variability): For any set $\mathcal{A} \in \mathcal{Z}$ such that (i) $\mathbb{P}_{\boldsymbol{z}|\boldsymbol{u}'}[\mathcal{A}] > 0$ for any $\boldsymbol{u}' \in \mathcal{U}$ and (ii) $\mathcal{A}$ cannot be expressed as $\mathcal{B} \times \mathcal{S}$ for any set $\mathcal{B} \subset \mathcal{C}$, there exist $\boldsymbol{u}_1, \boldsymbol{u}_2 \in \mathcal{U}$ such that $\mathbb{P}_{\boldsymbol{z}|\boldsymbol{u}_1}[\mathcal{A}] \neq \mathbb{P}_{\boldsymbol{z}|\boldsymbol{u}_2}[\mathcal{A}]$ where $\mathbb{P}_{\boldsymbol{z}|\boldsymbol{u}_i}[\mathcal{A}] = \int_{\boldsymbol{z} \in \mathcal{A}} P(\boldsymbol{z}|\boldsymbol{u}_i) d\boldsymbol{z}$.*
>
> *Then, by matching the model distribution in (1) with the marginals $\mathbb{P}_{\boldsymbol{x}^{(n)}}$ jointly, the component-wise and block-wise identifiability of $\boldsymbol{s}$ and $\boldsymbol{c}$ are guaranteed, respectively, both up to invertible nonlinear transformations.*
>
> ―――――――――
>
> [a] The auxiliary variable $\boldsymbol{u}$ is a notion from the nICA literature (Hyvarinen & Morioka, 2017; Hyvarinen et al., 2019; Khemakhem et al., 2020). In the context of multi-domain learning, $\boldsymbol{u}$ often represents the domain index.

The result is interesting and insightful—particularly, the domain variability condition (A4) is also used in our analysis. However, some major challenges still remain: First, (A2) requires that all components of $\boldsymbol{z}^{(n)} = (\boldsymbol{c}, \boldsymbol{s}^{(n)})$ are statistically independent given $n$ (corresponding to $\boldsymbol{u}$). The use of this condition is because the identifiability proof is reminiscent of nICA (Khemakhem et al., 2020; Hyvarinen et al., 2019). But elementwise independence is a nontrivial assumption. As shown in our analysis, for block-wise identifiability of $\boldsymbol{c}$ and $\boldsymbol{s}^{(n)}$, elementwise independence among individual entries of $\boldsymbol{z} = (\boldsymbol{c}, \boldsymbol{s})$ is not necessary. In addition, (A3) needs that at least $N = 2d_\mathrm{S} + 1$ domains exist (as $\boldsymbol{u}$ represents the domain index in this context), which is hardly practical—many domain translation tasks were performed over $N = 2$ domains (Choi et al., 2020).

## C PROOF OF THEOREM 3.3

We restate the theorem here:

> **Theorem** 3.3 Under Eq. (1), suppose that Assumptions 3.1 and 3.2 hold. Then, we have $\widehat{\boldsymbol{f}}_\mathrm{C}(\boldsymbol{x}^{(n)}) = \boldsymbol{\gamma}(\boldsymbol{c})$ and $\widehat{\boldsymbol{f}}_\mathrm{S}(\boldsymbol{x}^{(n)}) = \boldsymbol{\delta}(\boldsymbol{s}^{(n)}), \forall n \in [N]$, where $\boldsymbol{\gamma} : \mathcal{C} \to \mathbb{R}^{d_C}$ and $\boldsymbol{\delta} : \mathcal{S} \to \mathbb{R}^{d_\mathrm{S}}$ are injective functions.

**Proof Outline.** The proof pipeline of Theorem 3.3 can be divided into two parts: content identification and style identification. The content identification part includes two steps: (i) showing that $\widehat{\boldsymbol{f}}_\mathrm{C}$ does

not depend upon the style part, and thus is a function (denoted by $\boldsymbol{\gamma}$) of the content only and (ii) showing that $\boldsymbol{\gamma}$ is injective. For the first step, we use the domain variability assumption and matched content distribution to conclude that the preimage sets (under $\widehat{\boldsymbol{f}} \circ \boldsymbol{g}$) for any extracted content cover the entire style domain. This will imply that the extracted content is invariant of the style part. In the second step, we use the property of the determinant of the jacobian of $\widehat{\boldsymbol{f}} \circ \boldsymbol{g}$ to derive that $\boldsymbol{\gamma}$ should be injective. For style identification, we use the independence constraint and the result of content identification part to conclude that the extracted style does not depend upon the content part. And finally we use the injectivity of $\widehat{\boldsymbol{f}}$ to show that $\boldsymbol{\delta}$ should also be injective.

*Proof.* The proof is as follows:

### C.1 CONTENT IDENTIFICATION

Let $\boldsymbol{z}^{(n)} = (\boldsymbol{c}, \boldsymbol{s}^{(n)})$ and $\mathcal{Z} = \mathcal{C} \times \mathcal{S}$. Let $\widehat{\boldsymbol{c}}^{(n)} = \widehat{\boldsymbol{f}}_{\mathrm{C}}(\boldsymbol{x}^{(n)}), \forall n \in [N]$, and $\widehat{\boldsymbol{s}}^{(n)} = \widehat{\boldsymbol{f}}_{\mathrm{S}}(\boldsymbol{x}^{(n)}), \forall n \in [N]$. The proof consists of the following two steps:

Step 1. First, we show that under the assumptions in Theorem 3.3, $\widehat{\boldsymbol{c}}^{(n)}$ does not depend upon $\boldsymbol{s}^{(n)}$.

Step 2. Next, we show that $\widehat{\boldsymbol{c}}^{(n)}$ is transformed from the true shared component $\boldsymbol{c}$ via an injective function $\boldsymbol{\gamma} : \mathcal{C} \to \mathbb{R}^{d_{\mathrm{C}}}$ for all $n \in [N]$.

**Step 1:** We want to establish the following:

$$\frac{\partial \widehat{c}_i^{(n)}}{\partial s_j^{(n)}} = 0, \forall i \in [d_{\mathrm{C}}], j \in [d_{\mathrm{S}}]. \tag{10}$$

This can be shown using similar arguments in the domain adaptation work (Kong et al., 2022). To be specific, let $\boldsymbol{h} : \mathcal{Z} \to \mathbb{R}^{d_{\mathrm{C}}+d_{\mathrm{S}}}$ be defined as follows:

$$\boldsymbol{h} := \widehat{\boldsymbol{f}} \circ \boldsymbol{g}.$$

Let $\widehat{\mathcal{C}} = \boldsymbol{h}_{\mathrm{C}}(\mathcal{Z})$. Due to the distribution matching constraint (2a), the following holds for any $\mathcal{A}_{\mathrm{C}} \subseteq \widehat{\mathcal{C}}$:

$$\mathbb{P}_{\widehat{\boldsymbol{c}}^{(i)}}[\mathcal{A}_{\mathrm{C}}] = \mathbb{P}_{\widehat{\boldsymbol{c}}^{(j)}}[\mathcal{A}_{\mathrm{C}}], \; \forall i, j \in [N]$$

$$\overset{(a)}{\Longleftrightarrow} \mathbb{P}_{\boldsymbol{z}^{(i)}}\left[\boldsymbol{h}_{\mathrm{C}}^{-1}(\mathcal{A}_{\mathrm{C}})\right] = \mathbb{P}_{\boldsymbol{z}^{(j)}}\left[\boldsymbol{h}_{\mathrm{C}}^{-1}(\mathcal{A}_{\mathrm{C}})\right] \tag{11}$$

where $\boldsymbol{h}_{\mathrm{C}}^{-1}(\mathcal{A}_{\mathrm{C}}) := \{\boldsymbol{z} \mid \boldsymbol{h}_{\mathrm{C}}(\boldsymbol{z}) \in \mathcal{A}_{\mathrm{C}}\}$ is the preimage of $\boldsymbol{h}_{\mathrm{C}}(\cdot) := [\boldsymbol{h}(\cdot)]_{1:d_{\mathrm{C}}}$. The equivalence in (a) holds because $\mathbb{P}_{\widehat{\boldsymbol{c}}^{(n)}}[\mathcal{A}_{\mathrm{C}}] = \mathbb{P}_{\boldsymbol{h}_{\mathrm{C}}(\boldsymbol{z}^{(n)})}[\mathcal{A}_{\mathrm{C}}] = \mathbb{P}_{\boldsymbol{z}^{(n)}}[\boldsymbol{h}_{\mathrm{C}}^{-1}(\mathcal{A}_{\mathrm{C}})], \forall n \in [N]$.

**Sufficient Condition of** (10)**:** In order to show (10) holds almost surely, it suffices to show that

$$\forall \overline{\boldsymbol{c}} \in \widehat{\mathcal{C}}, \; \exists \, \mathcal{B}_{\mathrm{C}} \neq \phi \text{ and } \mathcal{B}_{\mathrm{C}} \subseteq \mathcal{C}$$
$$\text{s.t. } \boldsymbol{h}_{\mathrm{C}}^{-1}(\overline{\boldsymbol{c}}) = \mathcal{B}_{\mathrm{C}} \times \mathcal{S}. \tag{12}$$

Eq. (12) implies that no matter what the value of the private variable $\boldsymbol{s}^{(n)}$ is, as long as the shared variable $\boldsymbol{c} \in \mathcal{B}_{\mathrm{C}}$, the extracted $\widehat{\boldsymbol{c}} = \overline{\boldsymbol{c}}$. This implies that small changes in $s_j \forall j \in [d_{\mathrm{S}}]$, while keeping all other variables fixed, does not result in any change in $\widehat{c}_i, \forall i \in [d_{\mathrm{C}}]$—which means that (10) holds.

Further, the following condition is sufficient to ensure that (12) holds:

$$\forall \overline{\boldsymbol{c}} \in \widehat{\mathcal{C}}, \forall \epsilon > 0, \; \exists \, \mathcal{G}_{\mathrm{C}} \neq \phi \text{ and } \mathcal{G}_{\mathrm{C}} \subseteq \mathcal{C},$$
$$\boldsymbol{h}_{\mathrm{C}}^{-1}(\mathcal{N}_\epsilon(\overline{\boldsymbol{c}})) = \mathcal{G}_{\mathrm{C}} \times \mathcal{S}, \tag{13}$$

where $\mathcal{N}_\epsilon(\overline{\boldsymbol{c}}) = \{\boldsymbol{c}' \in \widehat{\mathcal{C}} \mid \|\overline{\boldsymbol{c}} - \boldsymbol{c}'\|_2 < \epsilon\}$ is an open set. To see how (13) implies (12), we use contradiction. Suppose that (13) does not imply (12). Then there exists an $\widetilde{n} \in [N]$ and $\widetilde{\boldsymbol{z}} = (\widetilde{\boldsymbol{c}}, \widetilde{\boldsymbol{s}}^{(\widetilde{n})}) \in \mathcal{Z}$ with

$$\widetilde{\boldsymbol{c}} \in \mathcal{B}_{\mathrm{C}} := \{\boldsymbol{z}_{1:d_{\mathrm{C}}} : \boldsymbol{z} \in \boldsymbol{h}_{\mathrm{C}}^{-1}(\overline{\boldsymbol{c}})\},$$

and $\widetilde{s}^{(\widetilde{n})} \in \mathcal{S}$ such that $h_{\mathrm{C}}(\widetilde{z}) \neq \overline{c}$. Since $h_{\mathrm{C}}(\cdot)$ is a continuous function there exists $\widehat{\epsilon} > 0$ such that

$$h_{\mathrm{C}}(\widetilde{z}) \notin \mathcal{N}_{\widehat{\epsilon}}(\overline{c})$$
$$\implies \widetilde{z} \notin h_{\mathrm{C}}^{-1}(\mathcal{N}_{\widehat{\epsilon}}(\overline{c}))$$

However, (13) states that

$$h_{\mathrm{C}}^{-1}(\mathcal{N}_{\widehat{\epsilon}}(\overline{c})) = \mathcal{G}_{\mathrm{C}} \times \mathcal{S},$$

for some $\mathcal{G}_{\mathrm{C}} \subseteq \mathcal{C}$. Since $\mathcal{B}_{\mathrm{C}} \subseteq \mathcal{G}_{\mathrm{C}}$ by their respective definitions, it implies that $\widetilde{c} \in \mathcal{G}_{\mathrm{C}}$. And therefore,

$$(\widetilde{c}, \widetilde{s}^{(\widetilde{n})}) \in h_{\mathrm{C}}^{-1}(\mathcal{N}_{\widehat{\epsilon}}(\overline{c})),$$

which is a contradiction. Hence, (13) implies (12). Therefore, it is sufficient to show (13) in order to prove (10).

**Proving** (10)**:** Proving (13) (equivalently (10)) can be accomplished using contradition. From (13), it is sufficient to show that $h_{\mathrm{C}}^{-1}(\mathcal{A}_{\mathrm{C}}) = \mathcal{B}_{\mathrm{C}} \times \mathcal{S}$ for a certain $\mathcal{B}_{\mathrm{C}} \subseteq \mathcal{C}$.

Suppose that $h_{\mathrm{C}}^{-1}(\mathcal{A}_{\mathrm{C}})) \neq \mathcal{B}_{\mathrm{C}} \times \mathcal{S}$ for any $\mathcal{B}_{\mathrm{C}} \subseteq \mathcal{C}$. Then, we can divide $h_{\mathrm{C}}^{-1}(\mathcal{A}_{\mathrm{C}})$ into two disjoint subsets, namely, $\mathcal{G}$ and $\mathcal{F}$. The $\mathcal{G}$ part can be represented as $\mathcal{B}_{\mathrm{C}} \times \mathcal{S}$ for a certain set $\mathcal{B}_{\mathrm{C}} \subseteq \mathcal{C}$ (it is possible that $\mathcal{G} = \phi$). The $\mathcal{F}$ part cannot be written as $\mathcal{D}_{\mathrm{C}} \times \mathcal{S}$ for any $\mathcal{D}_{\mathrm{C}} \subseteq \mathcal{C}$. Note that $\mathcal{F} \neq \phi$ due to the fact that $h_{\mathrm{C}}^{-1}(\mathcal{A}_{\mathrm{C}}) \neq \mathcal{B}_{\mathrm{C}} \times \mathcal{S}$, and has a strictly positive measure. Then, the following follows from (11):

$$\mathbb{P}_{z^{(i)}} \left[ h_{\mathrm{C}}^{-1}(\mathcal{A}_{\mathrm{C}}) \right] = \mathbb{P}_{z^{(j)}} \left[ h_{\mathrm{C}}^{-1}(\mathcal{A}_{\mathrm{C}}) \right]$$
$$\iff \mathbb{P}_{z^{(i)}}[\mathcal{G}] + \mathbb{P}_{z^{(i)}}[\mathcal{F}] = \mathbb{P}_{z^{(j)}}[\mathcal{G}] + \mathbb{P}_{z^{(j)}}[\mathcal{F}]$$
$$\iff \mathbb{P}_{c}[\mathcal{B}_{\mathrm{C}}]\mathbb{P}_{s^{(i)}|c}[\mathcal{S}|\mathcal{B}_{\mathrm{C}}] + \mathbb{P}_{z^{(i)}}[\mathcal{F}] = \mathbb{P}_{c}[\mathcal{B}_{\mathrm{C}}]\mathbb{P}_{s^{(j)}|c}[\mathcal{S}|\mathcal{B}_{\mathrm{C}}]$$
$$+ \mathbb{P}_{z^{(j)}}[\mathcal{F}]$$
$$\overset{(a)}{\iff} \mathbb{P}_{z^{(i)}}[\mathcal{F}] = \mathbb{P}_{z^{(j)}}[\mathcal{F}] \tag{14}$$

where (a) holds because we have $\mathbb{P}_{s^{(n)}|c}[\mathcal{S}|\mathcal{B}_{\mathrm{C}}] = 1, \forall n \in [N]$. However, (14) cannot hold due to Assumption 3.2. This is a contradication.

**Step 2:**

Consider the Jacobian of $h$

$$J_{h}(c, s) = \begin{bmatrix} J_{h_{\mathrm{C}}}(c) & J_{h_{\mathrm{C}}}(s) \\ J_{h_{\mathrm{S}}}(c) & J_{h_{\mathrm{S}}}(s) \end{bmatrix}. \tag{15}$$

In step 1, we have shown that $J_{h_{\mathrm{C}}}(s) = 0$. This implies that

$$\widehat{c}^{(n)} = \gamma(c), \forall n \in [N] \tag{16}$$

for some function $\gamma : \mathcal{C} \to \mathbb{R}^{d_{C}}$. Now, we want to show that $\gamma$ is injective.

Note that $h : \mathcal{Z} \to \mathbb{R}^{d_{C}+d_{S}}$ is a differentiable injective function because it is a composition of differentiable injection $\widehat{f}$ and bijection $g$.

This implies that $|\det(J_{h}(c, s))| = |\det(J_{h_{\mathrm{C}}}(c))||\det(J_{h_{\mathrm{S}}}(s))| \neq 0, \forall(c, s) \in \mathcal{C} \times \mathcal{S}$. Hence $|\det(J_{h_{\mathrm{C}}}(c))| \neq 0, \forall c \in \mathcal{C}$, which in turn implies that $\gamma$ is injective.

### C.2 STYLE IDENTIFICATION

Our goal is to show that $h_{\mathrm{S}}(c, s^{(n)}) = \delta(s^{(n)})$ for some injective function $\delta$. As before, we first show that $h_{\mathrm{S}}(c, s^{(n)})$ is only a function of $s^{(n)}$, i.e., $h_{\mathrm{S}}(c, s^{(n)}) = \delta(s^{(n)})$ for some function $\delta$. Then, we show that $\delta$ has to be injective.

To proceed, note that the statistical independence constraint $\widehat{c}^{(n)} \perp\!\!\!\perp \widehat{s}^{(n)}$ implies the following:

$$\widehat{c}^{(n)} \perp\!\!\!\perp \widehat{s}^{(n)}$$
$$\implies p(\widehat{c}^{(n)}, \widehat{s}^{(n)}) = p(\widehat{c}^{(n)})p(\widehat{s}^{(n)})$$
$$\implies I(\widehat{s}^{(n)}; \widehat{c}^{(n)}) = 0,$$

where $I(X; Y)$ denotes the mutual information between $X$ and $Y$. Since we have already established $\widehat{c}^{(n)} = \gamma(c)$ for some injective function $\gamma$, we express $c = \gamma^{-1}(\widehat{c}^{(n)})$ (where $\gamma^{-1}$ is the left inverse of $\gamma$; see Appendix A.1). Then, $\widehat{s}^{(n)} \to \widehat{c}^{(n)} \to \gamma^{-1}(\widehat{c}^{(n)}) = c$ is a Markov chain. This is because, when conditioned on $\widehat{c}^{(n)}$, $c = \gamma^{-1}(\widehat{c}^{(n)})$ is a constant, which is independent with $\widehat{s}^{(n)}$. Next, the *data processing inequality* in information theory (Cover, 1999, Theorem 2.8.1) (and corollary following the theorem) implies that

$$I(\widehat{s}^{(n)}; \widehat{c}^{(n)}) \geq I(\widehat{s}^{(n)}; \gamma^{-1}(\widehat{c}^{(n)})) = I(\widehat{s}^{(n)}; c)$$

As mutual information is always non-negative, we have $I(\widehat{s}^{(n)}; c) = 0$. This implies that $h_{\mathrm{S}}(c, s^{(n)})$ does not depend upon $c$. In other words, as pointed out in (Eastwood et al., 2023, Theorem 2, Step 3), we have $h_{\mathrm{S}}(c, s^{(n)}) = \delta(s^{(n)}), \forall n$ for some function $\delta$.

Finally, to see that $\delta$ is injective, we use contradiction. Suppose that $\delta$ is not injective. Then there exists $s_1, s_2 \in \mathcal{S}$ and $s_1 \neq s_2$ such that

$$\delta(s_1) = \delta(s_2). \tag{17}$$

However,

$$h(c, s_1) = (h_{\mathrm{C}}(c, s_1), h_{\mathrm{S}}(c, s_1)) = (\gamma(c), \delta(s_1))$$

Similarly,

$$h(c, s_2) = (h_{\mathrm{C}}(c, s_2), h_{\mathrm{S}}(c, s_2)) = (\gamma(c), \delta(s_2))$$

However, Eq. (17) implies that

$$h(c, s_1) = h(c, s_2)$$

which is a contradiction to the injectivity of $h$. Hence, $\delta$ is an injective function.

$\square$

## D    PROOF OF THEOREM 3.4

The theorem is re-stated as follows:

**Theorem** 3.4 Assume that the conditions in Theorem 3.3 hold. Let $\widehat{f}$ represent any solution of Problem (4). Assume the following conditions hold: (a) $\widehat{d}_{\mathrm{C}} \geq d_{\mathrm{C}}$ and $\widehat{d}_{\mathrm{S}} \geq d_{\mathrm{S}}$. (b) $0 < p_{z^{(n)}}(z) < \infty, \forall z \in \mathcal{Z} = \mathcal{C} \times \mathcal{S}, \forall n \in [N]$. Then, there exists injective functions $\gamma : \mathcal{C} \to \mathbb{R}^{\widehat{d}_C}$ and $\delta : \mathcal{S} \to \mathbb{R}^{\widehat{d}_S}, \forall n \in [N]$ such that $\widehat{c} = \widehat{f}_{\mathrm{C}}(x^{(n)}) = \gamma(c)$ and $\widehat{f}_{\mathrm{S}}(x^{(n)}) = \delta(s^{(n)}), \forall n \in [N]$.

**Proof Outline.** The proof of Theorem 3.4 follows the following pipeline

1. Content identification:

   Step 1  Showing that the extracted content $\widehat{c}^{(n)}$ does not depend upon the style part $s^{(n)}$.

   Step 2  Showing that the sparsity regularization forces the extracted style part to be free of any content information; and that, consequently, the extracted content has to be an injective function of the true content.

2. Style identification: showing that the extracted style is an injective function of the true style.

We will fist show that unknown dimensionality does not affect *Step 1 of content identification*; i.e., when $\widehat{d}_{\mathrm{C}} \geq d_{\mathrm{C}}$, $\widehat{c}^{(n)}$ will be independent of $s^{(n)}$ using the arguments from the proof of Theorem 3.3. The most important part of the proof is *Step 2 of content identification*, which requires completely different analytical approaches compared to that used for Theorem 3.3. For this, we first use contradiction to show that if $\gamma$ were not injective, the injectivity of $h$ will imply that $h_{\mathrm{S}}$ depends upon $c$ within a set of strictly positive measure. This will imply that the extracted style contains information from both the style as well as the content part, which will require $\mathbb{E}\|\widehat{s}_0^{(n)}\| > d_{\mathrm{S}}$. However, a feasible solution to Problem (4) can be constructed using the inverse of $g$ which has $\mathbb{E}\|\widehat{s}_0^{(n)}\| = d_{\mathrm{S}}$.

This will contradict our assumption that $\widehat{f}$ is an optimal solution to Problem (4). Hence, $\gamma$ should be injective. Finally, style identification follows the same proof as that of Theorem 3.3.

*Proof.* The proof is as follows:

**Content Identification.**

**Step 1:** We want to show that

$$\frac{\partial \widehat{c}_i^{(n)}}{\partial s_j^{(n)}} = 0, \forall i \in [\widehat{d}_{\mathrm{C}}], j \in [d_{\mathrm{S}}], \tag{18}$$

which will imply that $\boldsymbol{h}_{\mathrm{C}}(\boldsymbol{c}, \boldsymbol{s}) = \boldsymbol{\gamma}(\boldsymbol{c})$ for some function $\boldsymbol{\gamma}$.

For this, the proof of Step 1 of content identifiability of Theorem 3.3 holds without any modification since all arguments of the proof are dimension-agnostic.

**Step 2:** We want to show that $\boldsymbol{\gamma}$ is an injective function. For the sake of contradiction, assume that $\boldsymbol{\gamma}$ is not injective. Then there exists $\overline{\boldsymbol{c}} \in \mathcal{C}$ and $\widetilde{\boldsymbol{c}} \in \mathcal{C}, \overline{\boldsymbol{c}} \neq \widetilde{\boldsymbol{c}}$, such that $\boldsymbol{\gamma}(\overline{\boldsymbol{c}}) = \boldsymbol{\gamma}(\widetilde{\boldsymbol{c}})$. This is equivalent to

$$\boldsymbol{h}_{\mathrm{C}}(\overline{\boldsymbol{c}}, \boldsymbol{s}) = \boldsymbol{h}_{\mathrm{C}}(\widetilde{\boldsymbol{c}}, \boldsymbol{s}). \tag{19}$$

However, we know that $\boldsymbol{h} : \mathcal{Z} \to \mathbb{R}^{\widehat{d}_{\mathrm{C}} + \widehat{d}_{\mathrm{S}}}$ is an injective function, since it is a composition of injective and bijective function. This implies that

$$\boldsymbol{h}(\overline{\boldsymbol{c}}, \boldsymbol{s}) \neq \boldsymbol{h}(\widetilde{\boldsymbol{c}}, \boldsymbol{s}) \tag{20}$$

Eq. (19) and (20) imply that

$$\boldsymbol{h}_{\mathrm{S}}(\overline{\boldsymbol{c}}, \boldsymbol{s}) \neq \boldsymbol{h}_{\mathrm{S}}(\widetilde{\boldsymbol{c}}, \boldsymbol{s}) \tag{21}$$

Since $\boldsymbol{h}_{\mathrm{S}}$ is a differentiable function, (21) implies that there exists $\ell \in [\widehat{d}_{\mathrm{S}}], r \in [d_{\mathrm{C}}]$ and $\boldsymbol{c}^{\star} \in \mathcal{C}, \boldsymbol{s}^{\star} \in \mathcal{S}$, such that

$$\frac{\partial [h_{\mathrm{S}}]_\ell}{\partial c_r}(\boldsymbol{c}^{\star}, \boldsymbol{s}^{\star}) \neq 0. \tag{22}$$

Also, $\boldsymbol{h}_{\mathrm{S}}$ being differentiable implies that the above partial derivative is continuous. Hence, there exists $\epsilon > 0$, such that the above non-equality holds on $\mathcal{N}_\epsilon(\boldsymbol{z}^{\star}) \subseteq \mathcal{Z}$, where $\boldsymbol{z}^{\star} = (\boldsymbol{c}^{\star}, \boldsymbol{s}^{\star})$. In addition, with a small enough $\epsilon$, because of the continuity of $\frac{\partial [h_{\mathrm{S}}]_\ell}{\partial c_r}$, $[h_{\mathrm{S}}]_\ell$ is strictly monotonic (either increasing or decreasing) within the set $\mathcal{N}_\epsilon(\boldsymbol{z}^{\star})$, with respect to $c_r$. This implies that $[h_{\mathrm{S}}]_\ell$ is a locally monotonic function w.r.t. its input $c_r$ at any $\overline{\boldsymbol{z}} \in \mathcal{N}_\epsilon(\boldsymbol{z}^{\star})$.

However, this will imply $[h_{\mathrm{C}}]_i, \forall i \in [\widehat{d}_{\mathrm{C}}]$ cannot depend upon $c_r$ when restricted to $\mathcal{N}_\epsilon(\boldsymbol{z}^{\star})$, i.e., $\forall \overline{\boldsymbol{z}} \in \mathcal{N}_\epsilon(\boldsymbol{z}^{\star})$ and all $i \in [\widehat{d}_{\mathrm{C}}]$,

$$\frac{\partial [h_{\mathrm{C}}]_i}{\partial c_r}(\overline{\boldsymbol{z}}) = 0. \tag{23}$$

This is because, if $\frac{\partial [h_{\mathrm{C}}]_i}{\partial c_r}(\overline{\boldsymbol{z}}) \neq 0$ for some $\overline{\boldsymbol{z}} \in \mathcal{N}_\epsilon(\boldsymbol{z}^{\star})$ and $i \in [\widehat{d}_{\mathrm{C}}]$, then there exists a $\eta > 0$ such that $[h_{\mathrm{C}}]_i$ is strictly monotonic within $\mathcal{N}_\eta(\overline{\boldsymbol{z}})$, with respect to $c_r$.

Consider the non-empty open set $\mathcal{R} = \mathcal{N}_\epsilon(\boldsymbol{z}^{\star}) \cap \mathcal{N}_\eta(\overline{\boldsymbol{z}}) \cap \mathcal{Z}$. Due to the assumption that $0 < p_{\boldsymbol{z}^{(n)}}(\boldsymbol{z}) < \infty, \forall \boldsymbol{z} \in \mathcal{Z}, \mathcal{R}$ has strictly positive measure. However, $[h_{\mathrm{C}}]_i$ and $[h_{\mathrm{S}}]_\ell$ are simultaneously changing monotonically everywhere in $\mathcal{R}$ with respect $c_r$, which implies that $[h_{\mathrm{S}}]_\ell \perp\!\!\!\perp [h_{\mathrm{C}}]_i$ does not hold. This contradicts the independence constraint (2b). Hence (23) has to hold.

Now, Eq. (23) implies that

$$\boldsymbol{h}_{\mathrm{C}}(\overline{\boldsymbol{c}}_{-r}, \overline{c}_r, \overline{\boldsymbol{s}}) = \boldsymbol{h}_{\mathrm{C}}(\overline{\boldsymbol{c}}_{-r}, \widetilde{c}_r, \widetilde{\boldsymbol{s}}),$$

for any two points $(\overline{\boldsymbol{c}}, \overline{\boldsymbol{s}}), (\widetilde{\boldsymbol{c}}, \widetilde{\boldsymbol{s}}) \in \mathcal{N}_\epsilon(\boldsymbol{z}^{\star})$. Here, $\overline{\boldsymbol{c}}_{-r}$ represents all components of $\overline{\boldsymbol{c}}$ excluding $\overline{c}_r$. Next, the injectivity of $\boldsymbol{h}$ indicates that for any $(\overline{\boldsymbol{c}}, \overline{\boldsymbol{s}}), (\widetilde{\boldsymbol{c}}, \widetilde{\boldsymbol{s}}) \in \mathcal{N}_\epsilon(\boldsymbol{z}^{\star})$, if $\overline{c}_r \neq \widetilde{c}_r$ or $\overline{\boldsymbol{s}} \neq \widetilde{\boldsymbol{s}}$, then

$$\boldsymbol{h}_{\mathrm{S}}(\overline{\boldsymbol{c}}_{-r}, \overline{c}_r, \overline{\boldsymbol{s}}) \neq \boldsymbol{h}_{\mathrm{S}}(\overline{\boldsymbol{c}}_{-r}, \widetilde{c}_r, \widetilde{\boldsymbol{s}}). \tag{24}$$

Next, we show that Eq. (24) results in a contradiction to the fact that $\widehat{\boldsymbol{f}}$ is an optimal solution of Problem (4). This contradiction will conclude that $\boldsymbol{\gamma}$ is injective, as Eq. (24) was obtained by the assumption that $\boldsymbol{\gamma}$ is not injective.

Note that one can construct a feasible solution $\widetilde{\boldsymbol{f}}$ of Problem (4) satisfying the following:

$$\widetilde{\boldsymbol{f}}_{\mathrm{C}}(\boldsymbol{x}^{(n)}) = \begin{bmatrix} [\boldsymbol{g}^{-1}]_{\mathrm{C}}(\boldsymbol{x}^{(n)}) \\ 0 \\ \vdots \\ 0 \end{bmatrix} \in \mathbb{R}^{\widehat{d}_{\mathrm{C}}} \quad \text{and} \quad \widetilde{\boldsymbol{f}}_{\mathrm{S}}(\boldsymbol{x}^{(n)}) = \begin{bmatrix} [\boldsymbol{g}^{-1}]_{\mathrm{S}}(\boldsymbol{x}^{(n)}) \\ 0 \\ \vdots \\ 0 \end{bmatrix} \in \mathbb{R}^{\widehat{d}_{\mathrm{S}}}, \forall n \in [N], \boldsymbol{x}^{(n)} \in \mathcal{X}^{(n)},$$

which implies that $\forall n$,

$$\mathbb{E}[\|\widetilde{\boldsymbol{f}}_{\mathrm{S}}(\boldsymbol{x}^{(n)})\|_0] = d_{\mathrm{S}}. \tag{25}$$

This means that the optimal value of Problem (4b) is smaller than or equal to $d_{\mathrm{S}}$, as at least one solution can be constructed to attain this value.

Now, consider our learned $\widehat{\boldsymbol{s}}^{(n)} = [\widehat{\boldsymbol{f}} \circ \boldsymbol{g}(\boldsymbol{x}^{(n)})]_{\mathrm{S}} = \boldsymbol{h}_{\mathrm{S}}(\boldsymbol{x}^{(n)})$. We will show the following chain of inequalities $\forall n \in [N]$:

$$\begin{aligned}
\mathbb{E}_{\widehat{\boldsymbol{s}} \sim \mathbb{P}_{\widehat{\boldsymbol{s}}^{(n)}}}[\|\widehat{\boldsymbol{s}}\|_0] &= \int_{\boldsymbol{h}_{\mathrm{S}}(\mathcal{Z})} \|\widehat{\boldsymbol{s}}\|_0 \, d\mathbb{P}_{\widehat{\boldsymbol{s}}^{(n)}} \\
&= \int_{\boldsymbol{h}_{\mathrm{S}}(\mathcal{Z}) \backslash \boldsymbol{h}_{\mathrm{S}}(\mathcal{N}_\epsilon(\boldsymbol{z}^\star))} \|\widehat{\boldsymbol{s}}\|_0 \, d\mathbb{P}_{\widehat{\boldsymbol{s}}^{(n)}} + \int_{\boldsymbol{h}_{\mathrm{S}}(\mathcal{N}_\epsilon(\boldsymbol{z}^\star))} \|\widehat{\boldsymbol{s}}\|_0 \, d\mathbb{P}_{\widehat{\boldsymbol{s}}^{(n)}} \\
&\overset{(a)}{\geq} \int_{\boldsymbol{h}_{\mathrm{S}}(\mathcal{Z}) \backslash \boldsymbol{h}_{\mathrm{S}}(\mathcal{N}_\epsilon(\boldsymbol{z}^\star))} d_{\mathrm{S}} \, d\mathbb{P}_{\widehat{\boldsymbol{s}}^{(n)}} + \int_{\boldsymbol{h}_{\mathrm{S}}(\mathcal{N}_\epsilon(\boldsymbol{z}^\star))} (d_{\mathrm{S}} + 1) \, d\mathbb{P}_{\widehat{\boldsymbol{s}}^{(n)}} \\
&= d_{\mathrm{S}}(\mathbb{P}_{\widehat{\boldsymbol{s}}^{(n)}}[\boldsymbol{h}_{\mathrm{S}}(\mathcal{Z})] - \mathbb{P}_{\widehat{\boldsymbol{s}}^{(n)}}[\boldsymbol{h}_{\mathrm{S}}(\mathcal{N}_\epsilon(\boldsymbol{z}^\star))]) + (d_{\mathrm{S}} + 1)\mathbb{P}_{\widehat{\boldsymbol{s}}^{(n)}}[\boldsymbol{h}_{\mathrm{S}}(\mathcal{N}_\epsilon(\boldsymbol{z}^\star))] \\
&= d_{\mathrm{S}} + \mathbb{P}_{\widehat{\boldsymbol{s}}^{(n)}}[\boldsymbol{h}_{\mathrm{S}}(\mathcal{N}_\epsilon(\boldsymbol{z}^\star))] \\
&> d_{\mathrm{S}}. \tag{26}
\end{aligned}$$

where $(a)$ used the to-be-proven facts that $\|\widehat{\boldsymbol{s}}\|_0 \geq d_{\mathrm{S}}$ almost everywhere on $\boldsymbol{h}_{\mathrm{S}}(\mathcal{Z})$ and that $\|\widehat{\boldsymbol{s}}\|_0 \geq d_{\mathrm{S}} + 1$ almost everywhere on $\boldsymbol{h}_{\mathrm{S}}(\mathcal{N}_\epsilon(\boldsymbol{z}^\star))$. Note that if (26) holds, we reach a contradiction that $\widehat{\boldsymbol{f}}$ is an optimal solution of Problem (4), as the solution in (25) can attain a smaller objective value. We prove the facts used in (a) in the following:

First, to see that $\|\widehat{\boldsymbol{s}}\|_0 \geq d_{\mathrm{S}}, a.e.$ (on $\boldsymbol{h}_{\mathrm{S}}(\mathcal{Z})$), suppose for the sake of contradiction that $\|\widehat{\boldsymbol{s}}\|_0 < d_{\mathrm{S}}$ on a set of strictly positive measure, say $\mathcal{Q} \subseteq \boldsymbol{h}_{\mathrm{S}}(\mathcal{Z})$, i.e.,

$$[\boldsymbol{h}_{\mathrm{S}}]_{\#\mathbb{P}_{\boldsymbol{z}^{(n)}}}(\mathcal{Q}) > 0$$
$$\implies \mathbb{P}_{\boldsymbol{z}^{(n)}}[\boldsymbol{h}_{\mathrm{S}}^{-1}(\mathcal{Q})] > 0.$$

Since $\mathbb{P}_{\boldsymbol{z}^{(n)}}$ admits a PDF with $0 < p_{\boldsymbol{z}^{(n)}}(\boldsymbol{z}) < \infty, \forall \boldsymbol{z} \in \mathcal{Z}, \boldsymbol{h}_{\mathrm{S}}^{-1}(\mathcal{Q})$ should contain an open ball $\overline{\mathcal{Z}} \subseteq \mathcal{Z}$. However $\dim(\overline{\mathcal{Z}}) = d_{\mathrm{C}} + d_{\mathrm{S}}$ since $\overline{\mathcal{Z}}$ is homeomorphic to $\mathbb{R}^{d_{\mathrm{C}} + d_{\mathrm{S}}}$ (see Appendix A). Let $\mathcal{A}_q = \{\boldsymbol{a} \in \mathbb{R}^{\widehat{d}_{\mathrm{S}} + \widehat{d}_{\mathrm{C}}} \mid \|\boldsymbol{a}\|_0 \leq q\}$, i.e., the set of all vectors with at most $q$ non-zero elements. Note that $\dim(\mathcal{A}_q) = q$. Then $\mathcal{Q} \subseteq \mathcal{A}_{d_{\mathrm{S}} - 1}$. Hence $\dim(\mathcal{Q}) \leq d_{\mathrm{S}} - 1$.

Next, $\boldsymbol{h}$ is also an injective function from $\overline{\mathcal{Z}}$ to $\mathcal{Q} \times \boldsymbol{\gamma}(\mathcal{C})$. Note that $\dim(\boldsymbol{\gamma}(\mathcal{C})) \leq d_{\mathrm{C}}$ because $\boldsymbol{\gamma}$ is a continuous function and $\dim(\mathcal{C}) = d_{\mathrm{C}}$. But this implies that $\dim(\mathcal{Q} \times \boldsymbol{\gamma}(\mathcal{C})) = \dim(\mathcal{Q}) + \dim(\boldsymbol{\gamma}(\mathcal{C})) < d_{\mathrm{C}} + d_{\mathrm{S}}$.

However, this contradicts that $\boldsymbol{h}$ is an injective function from $\overline{\mathcal{Z}}$ to $\mathcal{Q} \times \boldsymbol{\gamma}(\mathcal{C})$ because the domain of $\boldsymbol{h}$ has larger dimension than the co-domain $\mathcal{Q} \times \boldsymbol{\gamma}(\mathcal{C})$. Hence $[\boldsymbol{h}_{\mathrm{S}}]_{\#\mathbb{P}_{\boldsymbol{z}^{(n)}}}(\mathcal{Q}) = 0$ and $\|\widehat{\boldsymbol{s}}\|_0 \geq d_{\mathrm{S}}, a.e.$

Second, to show that $\|\widehat{\boldsymbol{s}}\|_0 \geq d_{\mathrm{S}} + 1$ almost everywhere on $\boldsymbol{h}_{\mathrm{S}}(\mathcal{N}_\epsilon(\boldsymbol{z}^\star))$, consider the set $\mathcal{S} \times \mathcal{C}_r$, where $\mathcal{C}_r$ is the support of $c_r$. Eq. (24) implies that $\mathcal{M} = (\mathcal{S} \times \mathcal{C}_r) \cap \mathcal{N}_\epsilon(\boldsymbol{z}^\star)$ is homeomorphic to a subset of $\boldsymbol{h}_{\mathrm{S}}(\mathcal{N}_\epsilon(\boldsymbol{z}^\star))$. To see this, consider a fixed $(\overline{\boldsymbol{c}}, \overline{\boldsymbol{s}}) \in \mathcal{N}_\epsilon(\boldsymbol{z}^\star)$. Then, define a continuous function

$$\boldsymbol{t}(c_r, \boldsymbol{s}) = \boldsymbol{h}_{\mathrm{S}}(\overline{\boldsymbol{c}}_{-r}, c_r, \boldsymbol{s}),$$

i.e., $\boldsymbol{h}_{\mathrm{S}}$ with fixed partial input $\overline{\boldsymbol{c}}_{-r}$. Eq. (24) implies that $\boldsymbol{t}$ is an injective function over $\mathcal{M}$. Now, note that we have $\boldsymbol{t}(\mathcal{M}) \subseteq \boldsymbol{h}_{\mathrm{S}}(\mathcal{N}_\epsilon(\boldsymbol{z}^\star))$, and that $\boldsymbol{t}$ defines a homeomorphism between $\mathcal{M}$ and $\boldsymbol{t}(\mathcal{M})$. Hence $\mathcal{M}$ is homeomorphic to a subset of $\boldsymbol{h}_{\mathrm{S}}(\mathcal{N}_\epsilon(\boldsymbol{z}^\star))$. This implies that $\dim(\boldsymbol{h}_{\mathrm{S}}(\mathcal{N}_\epsilon(\boldsymbol{z}^\star)) \geq \dim(\mathcal{M})$. However, $\dim(\mathcal{M}) = d_{\mathrm{S}} + 1$ because $\mathcal{M}$ is an open set in $\mathbb{R}^{d_{\mathrm{S}} + 1}$ as $\mathcal{S}$ is an open

set in $\mathbb{R}^{d_S}$, $\mathcal{C}_r$ is an open set in $\mathbb{R}$, and thus $(\mathcal{S} \times \mathcal{C}_r) \cap \mathcal{N}_\epsilon(z^\star)$ is an open set in $\mathbb{R}^{d_S+1}$. Hence, $\dim(h_S(\mathcal{N}_\epsilon(z^\star))) \geq \dim(\mathcal{M}) = d_S + 1$. This concludes the proof of the inequalities in (26).

Finally $\mathbb{E}_{\widehat{s}^{(n)}}[\|\widehat{s}\|_0] > d_S, \forall n \in [N]$ contradicts that $\widehat{f}$ is an optimal solution to Problem (4). Hence, $\gamma$ is an injective function.

**Style Identification.**

Finally, $h_S$ cannot depend upon $c$ by the same reason as outlined in Sec. C.2. This implies that $h_S(c, s) = \delta(s)$, for some function $\delta : \mathcal{S} \to \mathbb{R}^{\widehat{d}_S}$. Similarly, $\delta$ is injective by the same reason outlined in Sec. C.2.

This concludes the proof. □

# E  PROOF OF THEOREM 4.2

## E.1  RELATIONSHIP BETWEEN THE DATA AND LATENT SPACES.

Before proceeding with the proof of Theorem 3.4, we clarify the relationship between the data and latent spaces to provide a clearer understanding of the theorem and its proof.

First, note that $\mathcal{C}$ and $\mathcal{S}$ are open sets in $\mathbb{R}^{d_C}$ and $\mathbb{R}^{d_S}$, respectively. Thus, $\mathcal{X} = g(\mathcal{C} \times \mathcal{S})$ forms a $d_C + d_S$-dimensional manifold within $\mathbb{R}^d$. In Theorem 4.1(b), by assuming $q : \widehat{\mathcal{C}} \times \widehat{\mathcal{S}} \to \mathcal{X}$ is bijective, we effectively assume that $\widehat{\mathcal{C}} \times \widehat{\mathcal{S}}$ is a $d_C + d_S$-dimensional manifold within $\mathbb{R}^{\widehat{d}_C+\widehat{d}_S}$. This holds because $\widehat{\mathcal{C}} \times \widehat{\mathcal{S}} = q^{-1} \circ g(\mathcal{C} \times \mathcal{S})$, and since $q^{-1} \circ g$, a composition of two bijections, is itself a bijection. Consequently, although the ambient space $\mathbb{R}^{\widehat{d}_C+\widehat{d}_S}$ is higher dimensional than $\mathcal{C} \times \mathcal{S}$, the spaces $\widehat{\mathcal{C}} \times \widehat{\mathcal{S}}, \mathcal{C} \times \mathcal{S}$, and $\mathcal{X}$ all have the same manifold dimension of $d_C + d_S$, allowing for the definition of bijective mappings between them.

## E.2  THEOREM 4.2

The theorem is restated as follows:

> **Theorem** 4.2 Let $(\widehat{q}, \widehat{e}_C, \widehat{e}_S^{(n)}, \widehat{d})$ be any differentiable optimal solution of Problem (6). Let $\mathcal{C}$ and $\mathcal{S}$ be simply connected open sets. Let $0 < p_{z^{(n)}}(z) < \infty, \forall z \in \mathcal{Z} = \mathcal{C} \times \mathcal{S}$. Under the assumptions in Theorem 3.3, we have the following:
>
> (a) If $\widehat{d}_C = d_C$ and $\widehat{d}_S = d_S$ and (6b) is absent, then $\widehat{q} : \widehat{\mathcal{C}} \times \widehat{\mathcal{S}} \to \mathcal{X}$ is bijective and $\widehat{f} = \widehat{q}^{-1}$ is also a solution of Problem (2).
>
> (b) If $\widehat{d}_C > d_C$ and $\widehat{d}_S > d_S$ and $\widehat{q} : \widehat{\mathcal{C}} \times \widehat{\mathcal{S}} \to \mathcal{X}$ is bijective, then $\widehat{f} = \widehat{q}^{-1}$ is also a solution of Problem (4).

## E.3  PROOF OF PART (A).

One can see that problem (6a) is equivalent to the following optimization problem:

$$\text{find} \quad q, e_C, \{e_S^{(n)}\}_{n=1}^N \tag{27a}$$

$$\text{subject to} \quad [q]_{\#\mathbb{P}_{e_C(r_C), e_S^{(n)}(r_S^{(n)})}} = \mathbb{P}_{x^{(n)}}, \tag{27b}$$

$$r_C \sim \mathcal{N}(\mathbf{0}, I_{d_C}), r_S^{(n)} \sim \mathcal{N}(\mathbf{0}, I_{d_S}). \tag{27c}$$

First, the solution of (6a) satisfies the distribution matching constraint (27b) because of (Goodfellow et al., 2014) [Theorem 1]. Next, define:

$$\alpha^{(n)}(r_C, r_S^{(n)}) = \widehat{q}(\widehat{e}_C(r_C), \widehat{e}_S^{(n)}(r_S^{(n)})).$$

Consider the following theorem from (Zimmermann et al., 2021).

**Proposition E.1** ( Sec. A.5. "Effects of the Uniformity Loss", Proposition 5 (Zimmermann et al., 2021)). *Let $\mathcal{M}$ and $\mathcal{N}$ be simply connected and oriented $\mathcal{C}^1$ manifolds without boundaries and $\boldsymbol{h} : \mathcal{M} \to \mathcal{N}$ be a differentiable map. Further, let the random variable $\boldsymbol{z} \in \mathcal{M}$ be distributed according to $\boldsymbol{z} \in p(\boldsymbol{z})$ for a regular function $p$, i.e.,$0 < p < \infty$. If the push forward $p_{\#\boldsymbol{h}}(\boldsymbol{z})$ of $p$ through $\boldsymbol{h}$ is also a regular density, i.e., $0 < p_{\#h} < \infty$, then $\boldsymbol{h}$ is a bijection.*

Note that Proposition E.1 applies to a range of cases, including low dimensional manifolds $\mathcal{M}$ and $\mathcal{N}$ embedded in spaces with different dimensions, as long as $\mathcal{M}$ and $\mathcal{N}$ have the same manifold dimension (see application of Proposition E.1 (Zimmermann et al., 2021) Corollary 1). The proof relies on showing that the determinant of the Jacobian of $\boldsymbol{h}$ cannot vanish. Here, the Jacobian of $\boldsymbol{h}$ at a point $p$ corresponds to the matrix representation of the differential $d\boldsymbol{h}(p) : T_p\mathcal{M} \to T_{\boldsymbol{h}(p)}\mathcal{N}$, where $T_p\mathcal{M}$ and $T_{\boldsymbol{h}(p)}\mathcal{N}$ are tangent spaces at point at point $p$ and $\boldsymbol{h}(p)$ in $\mathcal{M}$ and $\mathcal{N}$, respectively Lee & Lee (2012)[Chapter 3]. Since the manifold dimensions of $\mathcal{M}$ and $\mathcal{N}$ are the same (required for $\boldsymbol{h}$ to be bijective), the Jacobian is a square matrix.

In our case, $\mathcal{Z}$ and $\mathcal{X}$ is a simply connected and oriented $\mathcal{C}^1$ manifold without boundaries. In addition, $\boldsymbol{\alpha}^{(n)} : \mathbb{R}^{d_S+d_C} \to \mathcal{X}$ is the differentiable map that we hope to use the above proposition to characterize. Every element of the random vector $\boldsymbol{r} = (\boldsymbol{r}_C, \boldsymbol{r}_S^{(n)})$ follows the standard normal distribution. Hence, it is readily seen that $0 < p_{\boldsymbol{r}^{(n)}}(\boldsymbol{r}) < \infty, \forall \boldsymbol{r} \in \mathbb{R}^{d_C+d_S}$. In addition, the assumptions that $0 < p_{\boldsymbol{z}^{(n)}}(\boldsymbol{z}) < \infty, \forall \boldsymbol{z} \in \mathcal{Z}$ with simply-connected $\mathcal{Z}$ and that $g$ is a differentiable bijection imply that $0 < p_{\boldsymbol{x}^{(n)}}(\boldsymbol{x}) < \infty, \forall \boldsymbol{x} \in \mathcal{X}, \forall n$ and that $\mathcal{X}$ is simply connected.

Following the above arguments, we can apply Proposition E.1 to conclude that $\boldsymbol{\alpha}^{(n)}, \forall n \in [N]$ are bijections. This, in turn, implies that $\widehat{\boldsymbol{q}}$ is a surjection and $\widehat{\boldsymbol{e}}_C$ and $\widehat{\boldsymbol{e}}_S$ are injections. However, since $\widehat{\boldsymbol{q}}$ is defined on the range of $\widehat{\boldsymbol{e}}_C$ and $\widehat{\boldsymbol{e}}_S$, $\widehat{\boldsymbol{q}}$ is a bijection. Consequently, $\widehat{\boldsymbol{e}}_C$ and $\widehat{\boldsymbol{e}}_S$ are also bijections.

To proceed, notice that (27b) implies that $\widehat{\boldsymbol{f}} = \widehat{\boldsymbol{q}}^{-1}$ satisfies the following:

$$[\widehat{\boldsymbol{q}}_C^{-1}]_{\#\mathbb{P}_{\boldsymbol{x}^{(i)}}} = [\widehat{\boldsymbol{q}}_C^{-1}]_{\#\mathbb{P}_{\boldsymbol{x}^{(j)}}}, \forall i, j \in [N],$$

where $\widehat{\boldsymbol{q}}_C^{-1}$ is the output dimensions of $\widehat{\boldsymbol{q}}^{-1}$ that correspond to the content part. To see the above equality, let $\widehat{\boldsymbol{x}}^{(n)} = \widehat{\boldsymbol{q}}(\widehat{\boldsymbol{e}}_c(\boldsymbol{r}_C)), \widehat{\boldsymbol{e}}_s^{(n)}(\boldsymbol{r}_S)$. Then,

$$\widehat{\boldsymbol{q}}_C^{-1}(\widehat{\boldsymbol{x}}^{(n)}) = \widehat{\boldsymbol{q}}_C^{-1} \circ \widehat{\boldsymbol{q}}(\widehat{\boldsymbol{e}}_c(\boldsymbol{r}_C), \widehat{\boldsymbol{e}}_s^{(n)}(\boldsymbol{r}_S)) = \widehat{\boldsymbol{e}}_c(\boldsymbol{r}_C).$$

Since $\widehat{\boldsymbol{x}}^{(n)} \sim \mathbb{P}_{\boldsymbol{x}^{(n)}}$, this implies that

$$[\widehat{\boldsymbol{q}}_C^{-1}]_{\#\mathbb{P}_{\boldsymbol{x}^{(n)}}} = [\widehat{\boldsymbol{e}}_c]_{\#\mathbb{P}_{\boldsymbol{r}_C}}, \forall n \in [N]$$

Since the distribution $[\widehat{\boldsymbol{e}}_c]_{\#\mathbb{P}_{\boldsymbol{r}_C}}$ is independent of $n$, $[\widehat{\boldsymbol{q}}_C^{-1}]_{\#\mathbb{P}_{\boldsymbol{x}^{(n)}}}, \forall n \in [N]$ are matched. This means that $\widehat{\boldsymbol{q}}^{-1}$ satisfies the constraint (2a). Similarly, since $\widehat{\boldsymbol{e}}_c(\boldsymbol{r}_C) \perp\!\!\!\perp \widehat{\boldsymbol{e}}_s^{(n)}(\boldsymbol{r}_S^{(n)}), \forall n \in [N]$,

$$[\widehat{\boldsymbol{q}}_C^{-1}]_{\#\mathbb{P}_{\boldsymbol{x}^{(n)}}} \perp\!\!\!\perp [\widehat{\boldsymbol{q}}_S^{-1}]_{\#\mathbb{P}_{\boldsymbol{x}^{(n)}}}, \forall n \in [N].$$

Hence $\widehat{\boldsymbol{q}}^{-1}$ also satisfies the constraint (2b). Therefore, $\widehat{\boldsymbol{q}}^{-1}$ is an optimal solution of Problem (2)

It remains to show that a solution to Problem (27) under our data generative model in (1) exists. When $\mathcal{C} \subseteq \mathbb{R}^{d_C}$ and $\mathcal{S} \subseteq \mathbb{R}^{d_S}$ are open sets and simply connected, and $p_{\boldsymbol{c}}(\boldsymbol{c}) > 0$ and $p_{\boldsymbol{s}^{(n)}}(\boldsymbol{s}) > 0, \forall \boldsymbol{s} \in \mathcal{S}, \boldsymbol{c} \in \mathcal{C}, n \in [N]$, there exists a smooth bijective function that transforms standard normal distributions, i.e., $\mathcal{N}(\boldsymbol{0}, \boldsymbol{I}_{d_C})$ and $\mathcal{N}(\boldsymbol{0}, \boldsymbol{I}_{d_S})$, to the content and style distributions, $\mathbb{P}_{\boldsymbol{c}}$ and $\mathbb{P}_{\boldsymbol{s}^{(n)}}$, respectively. Such functions can be constructed by using Darmois construction (Darmois, 1951) to first convert the normal distribution to uniform distribution, and then using inverse Darmois construction to convert the uniform distribution to $\mathbb{P}_{\boldsymbol{c}}$ (and $\mathbb{P}_{\boldsymbol{s}^{(n)}}$) (see (Hyvärinen & Pajunen, 1999) for more details). Let $\boldsymbol{\kappa}_C : \mathcal{N}(\boldsymbol{0}, \boldsymbol{I}_{d_C}) \to \mathcal{C}$ and $\boldsymbol{\kappa}_S : \mathcal{N}(\boldsymbol{0}, \boldsymbol{I}_{d_S}) \to \mathcal{S}$ denote the aforementioned functions. Then, a solution to Problem (27) is $g$, $\boldsymbol{\kappa}_C$, and $\boldsymbol{\kappa}_S^{(n)}, \forall n \in [N]$ for the optimization functions $\boldsymbol{q}$, $\boldsymbol{e}_C$, and $\boldsymbol{e}_S^{(n)} \forall n \in [N]$, respectively. Hence, problem (27) is feasible. This concludes the proof.

### E.4 Proof of Part (b).

Problem (6) is equivalent to the following problem:

$$\min_{\boldsymbol{q}, e_{\mathrm{C}}, \{e_{\mathrm{S}}^{(n)}\}_{n=1}^{N}} \quad \frac{1}{N} \sum_{n=1}^{N} \mathbb{E}\left[\left\|\boldsymbol{e}_{\mathrm{S}}^{(n)}(\boldsymbol{r}_{\mathrm{S}}^{(n)})\right\|_{0}\right] \tag{28a}$$

$$\text{subject to} \quad [\boldsymbol{q}]_{\#\mathbb{P}_{\boldsymbol{e}_{\mathrm{C}}(\boldsymbol{r}_{\mathrm{C}}), \boldsymbol{e}_{\mathrm{S}}^{(n)}(\boldsymbol{r}_{\mathrm{S}}^{(n)})}} = \mathbb{P}_{\boldsymbol{x}^{(n)}}, \tag{28b}$$

$$\boldsymbol{r}_{\mathrm{C}} \sim \mathcal{N}(\boldsymbol{0}, \boldsymbol{I}_{d_{\mathrm{C}}}), \boldsymbol{r}_{\mathrm{S}}^{(n)} \sim \mathcal{N}(\boldsymbol{0}, \boldsymbol{I}_{d_{\mathrm{S}}}) \tag{28c}$$

Following the same arguments as in the proof of Theorem Part (a), Eq. (28) implies that

$$[\widehat{\boldsymbol{q}}_{\mathrm{C}}^{-1}]_{\#\mathbb{P}_{\boldsymbol{x}^{(i)}}} = [\widehat{\boldsymbol{q}}_{\mathrm{C}}^{-1}]_{\#\mathbb{P}_{\boldsymbol{x}^{(j)}}}, \forall i, j \in [N]$$

The above implies that

$$[\widehat{\boldsymbol{q}}_{\mathrm{C}}^{-1}]_{\#\mathbb{P}_{\boldsymbol{x}^{(n)}}} = [\widehat{\boldsymbol{e}}_{c}]_{\#\mathbb{P}_{\boldsymbol{r}_{\mathrm{C}}}}, \forall n \in [N],$$

which means that $\widehat{\boldsymbol{q}}_{\mathrm{C}}^{-1}$ satisfies the constraint (4b).

Similarly, since $\widehat{\boldsymbol{e}}_{c}(\boldsymbol{r}_{\mathrm{C}}) \perp\!\!\!\perp \widehat{\boldsymbol{e}}_{s}^{(n)}(\boldsymbol{r}_{\mathrm{S}}^{(n)}), \forall n \in [N]$,

$$[\widehat{\boldsymbol{q}}_{\mathrm{C}}^{-1}]_{\#\mathbb{P}_{\boldsymbol{x}^{(n)}}} \perp\!\!\!\perp [\widehat{\boldsymbol{q}}_{\mathrm{S}}^{-1}]_{\#\mathbb{P}_{\boldsymbol{x}^{(n)}}}, \forall n \in [N].$$

Hence $\widehat{\boldsymbol{q}}^{-1}$ also satisfies the constraint (4c). Therefore, $\widehat{\boldsymbol{q}}^{-1}$ is also a solution to Problem (4).

## F Benchmarked Multi-Domain Translation Schemes

**StarGAN and StarGAN v2.** The representative multi-domain translation frameworks, namely, StarGAN (Choi et al., 2018) and StarGAN v2 (Choi et al., 2020), learn a common translation function that matches distribution between all pairs of domains. To be precise, let $\boldsymbol{t} : \mathcal{X} \times \mathcal{S} \rightarrow \mathcal{X}$ represent the translation function that takes in a sample $\boldsymbol{x}^{(s)}$ in the source domain $s$, and a style component $\boldsymbol{s}^{(t)}$ from the target domain $t$, and outputs the translated sample $\boldsymbol{x}^{(s \rightarrow t)}$ from the target domain $t$ as

$$\boldsymbol{x}^{(s \rightarrow t)} = \boldsymbol{t}(\boldsymbol{x}^{(s)}, \boldsymbol{s}^{(t)}).$$

In (Choi et al., 2020) $\boldsymbol{t}$ is learnt by optimizing the following criteria:

$$\mathcal{L}_{\mathrm{StarGAN}} = \mathcal{L}_{\mathrm{match}} + \mathcal{L}_{\mathrm{inv}},$$

where $\mathcal{L}_{\mathrm{inv}}$ promotes invertibility of the translation function, and is composed of cycle-consistency, style diversity, and style invertibility terms, whereas $\mathcal{L}_{\mathrm{match}}$ promotes distribution matching between all pairs of domains after translation by $\boldsymbol{t}$. Specifically,

$$\mathcal{L}_{\mathrm{match}} = \tag{29}$$

$$\sum_{s,t \in [N] \times [N]} \mathbb{E}\left[\log \boldsymbol{d}_{s}(\boldsymbol{x}^{(s)}) + \log\left(1 - \boldsymbol{d}_{t}(\boldsymbol{\tau}(\boldsymbol{x}^{(s)}, \boldsymbol{s}^{(t)}))\right)\right], \tag{30}$$

where $\boldsymbol{d}_{i}$ is the discriminator for the $i$th domain. Here, all pairs of domains, $(s, t) \in [N] \times [N]$, are considered for distribution matching. This can be potentially expensive for large $N$. The StarGAN loss matches $N(N-1)$ pairs of distributions for $N$ domains, which may not be affordable for large $N$.

**Learning Pipeline of (Xie et al., 2023).** The work (Xie et al., 2023) proposed the following pipeline. First, an multi-domain generative model is trained, which is similar to our system. Second, instead of directly using the trained system, the work proposed to train a separate StarGAN with new regularization. To be specific, the work proposed to use synthetic paired samples $(\boldsymbol{x}^{(s)}, \boldsymbol{x}^{(t)})$ generated by their generative model to further regularize $\mathcal{L}_{\mathrm{StarGAN}}$, which leads to the following loss:

$$\mathcal{L}_{\mathrm{StarGAN}} + \mathcal{L}_{\mathrm{sup}}$$

where the second term is defined as

$$\mathcal{L}_{\text{sup}} = \mathbb{E}_{\boldsymbol{x}^{(s)}, \boldsymbol{x}^{(t)} \sim \mathbb{P}_{\widehat{\boldsymbol{\theta}}}} \| \boldsymbol{t}(\boldsymbol{x}^{(s)}, \boldsymbol{s}^{(t)}) - \boldsymbol{x}^{(t)} \|,$$

in which $\mathbb{P}_{\widehat{\boldsymbol{\theta}}}$ represents distribution of the learned multi-domain generative model, and $\boldsymbol{s}^{(t)}$ is obtained from $\boldsymbol{x}^{(t)}$ by using a separate style encoder.

As we mentioned, as one can show that the generators are in fact invertible in the learned generative model (see Theorem 4.2), the above procedure can be circumvented. Using an off-the-shelf GAN inversion solver assists translation without retraining a regularization StarGAN.

## G  EXPERIMENTAL DETAILS

### G.1  NEURAL NETWORK

We adopt the neural architecture of StyleGAN-ADA to represent the generator $\boldsymbol{q}$ as well as the discriminator $\boldsymbol{d}$ in Problem (6). Specifically, StyleGAN-ADA generator consists of a mapping network and a synthesis network (Karras et al., 2020). We use the synthesis network to represent $\boldsymbol{q}$, whereas the mapping network is discarded. For each of $\boldsymbol{e}_{\text{C}}$ and $\{\boldsymbol{e}_{\text{S}}^{(n)}\}_{n=1}^{N}$, we use 3 layer MLP with 512 hidden units in each hidden layer. The input and output size of the MLPs for $\boldsymbol{e}_{\text{C}}$ and $\{\boldsymbol{e}_{\text{S}}^{(n)}\}_{n=1}^{N}$ are $\widehat{d}_{\text{C}} = 384$ and $\widehat{d}_{\text{S}} = 128$, respectively. We feed the output of $\boldsymbol{e}_{\text{C}}$ to the first 5 layers of the synthesis network (i.e., $\boldsymbol{q}$), and the output of $\boldsymbol{e}_{\text{S}}^{(n)}, \forall n \in [N]$ are fed to the remaining layers of $\boldsymbol{q}$. Compared to (Xie et al., 2023), $\boldsymbol{e}_{\text{S}}^{(n)}, \forall n$ are much more simplified since they are no longer restricted to component-wise invertible functions, which require special neural network design.

### G.2  TRAINING HYPERPARAMETER SETTING

The hyperparameters used for GAN training in Problem (6) is similar to those used in StyleGAN-ADA. Mainly, we use Adam (Kingma & Ba, 2015) with an initial learning rate of 0.0025 with a batch size of 16. The hyperparameters in Adam that control the exponential decay rates of first and second order moments are set to $\beta_1 = 0$ and $\beta_2 = 0.99$, respectively. For all datasets, we train the networks for 300,000 iterations. Sparsity regularization weight in Problem (6) is set to 0.3 for all experiments. We also use the data augmentation techniques proposed in (Karras et al., 2020) for improved GAN training on limited data.

### G.3  GAN INVERSION SETTING

We follow the optimization procedure in (Karras et al., 2020)[2] for GAN Inversion. To summarize, recall the GAN inversion problem for the multi-domain translation task:

$$(\widehat{\boldsymbol{c}}, \widehat{\boldsymbol{s}}) = \arg \min_{\boldsymbol{c}, \boldsymbol{s}} \text{div}(\boldsymbol{q}(\boldsymbol{c}, \boldsymbol{s}), \boldsymbol{x}^{(i)}), \tag{31}$$

Eq. 31 is solved using gradient based optimization of $\boldsymbol{c}$ and $\boldsymbol{s}$ using Adam (Kingma & Ba, 2015) with an initial learning rate of 0.1. The hyperparameters of Adam are set to $\beta_1 = 0.9, \beta_2 = 0.999$. The optimization is carried out for 400 steps. The div function is realized by using mean squared error in the feature space of pre-trained VGG16 (Simonyan & Zisserman, 2014) neural network, i.e.,

$$\text{div}(\boldsymbol{a}, \boldsymbol{b}) = \frac{1}{M} \| \text{VGG}(\boldsymbol{a}) - \text{VGG}(\boldsymbol{b}) \|_2^2,$$

where $M$ is the output dimension of $\text{VGG}(\cdot)$.

### G.4  BASELINES FOR MULTI-DOMAIN GENERATION

We use `I-GAN`[3] (Xie et al., 2023), `StyleGAN-ADA`[2](Karras et al., 2020), and `Transitional-cGAN`[4] (Shahbazi et al., 2021) as baselines for multi-domain generation task.

---

[2]https://github.com/NVlabs/stylegan2-ada-pytorch.git
[3]https://github.com/Mid-Push/i-stylegan.git
[4]https://github.com/mshahbazi72/transitional-cGAN.git

Note that `I-GAN` is chosen because it is the most relevant to our work, whereas `StyleGAN-ADA` is a very representative work among conditional GANs. Finally, `Transitional-cGAN` is a recent and representative conditional GAN for limited data regime (note that the datasets used in our work fall under limited data regime (Karras et al., 2020)). For all the baselines, training is done with their default setting for all datasets. All the baselines are also trained for 300,000 iterations with a batch size of 16 for a fair comparison, as well as to control the training time.

### G.5 BASELINES FOR MULTI-DOMAIN TRANSLATION

We use `StarGANv2`[5] (Choi et al., 2020), `SmoothGAN`[6] (Liu et al., 2021), `I-GAN (Gen)`[3] (Xie et al., 2023) and `I-GAN (Tr)`[3] (Xie et al., 2023) as baselines for the multi-domain translation task. Note that `I-GAN (Gen)` refers to the method that uses the multidomain generative model proposed by (Xie et al., 2023) to carry out the translation procedure via GAN inversion as described in Section 6. Whereas, `I-GAN (Tr)` is the separate domain translation system proposed by (Xie et al., 2023) as described in Appendix F. Note that `I-GAN (Tr)` is trained using the paired samples generated by `I-GAN` (see Sec. G.4).

### G.6 DATASETS DETAILS

**AFHQ.** We use the AFHQ (Choi et al., 2020) dataset for both multi-domain generation and translation tasks. The AFHQ dataset contains images of animal faces in three domains: cat, dog, and wild with 5066, 4679, and 4594 training images, and 494, 492, and 484 testing images. We resize all images to $256 \times 256$ for training and testing.

**CelebA-HQ.** The CelebA-HQ dataset (Karras et al., 2018) for both multi-domain generation and translation tasks. It contains high-resolution images of celebrity faces along with attributes such as gender, hair color, etc. We split the dataset into two domains based on gender. The male domain contains 18,875 images, whereas the female domain contains 11,025 images. We hold out 1000 images from each domain for testing and use the rest for training. Similar to AFHQ, we resize all images $256 \times 256$ for training and testing.

**CelebA.** The CelebA dataset (Liu et al., 2015) is used for multi-domain generation task. It contains 202,599 images of celebrity faces along with different attributes. We split the dataset into 7 domains based on the following attributes: "Black hair", "Blonde hair", "Brown hair", "Female", "Male", "Old", and "Young". We resize all images to $64 \times 64$ for training and testing.

### G.7 REGARDING GAN TRAINING STABILITY

GAN training is known to be quite sensitive to hyperparameter setting, and even randomness in initialization. However, in our experiments, we did not observe severe optimization issues or performance degradation with minor changes in hyperparameters, e.g., $\lambda$, batch size, and random initialization. Nonetheless, we did observe that using overly deep fully connected networks (>5 layers) for content encoder $e_C$ and style encoders $e_S^{(n)}, \forall n$ could lead to convergence issues. Hence, we fixed the number of layers of the content and style encoders to be 3 for all datasets except for the MNIST digits experiment presented in Appendix H.1.

## H ADDITIONAL EXPERIMENTAL RESULTS

### H.1 ADDITIONAL DATASET: ROTATED AND COLORED MNIST

In this part, we present additional experiments on MNIST digits, where the domains are rotated MNIST and colored MNIST. Here, the style corresponds to the color and rotations for the colored and rotated MNIST, respectively. Whereas, the content is the identity of the digits.

**Dataset.** For the rotated MNIST domain, we apply rotation uniformly sampled from [-70, 70] degrees to all the MNIST digits. For the colored MNIST domain, we apply randomly sampled color to the

---

[5]https://github.com/clovaai/stargan-v2.git

[6]https://github.com/yhlleo/SmoothingLatentSpace.git

MNIST digit. Specifically, for a given MNIST digit image, we uniformly sample a vector from $[0,1] \times [0,1] \times [0,1]$, corresponding to the normalized RGB channel values. Then, we multiply each pixel of the given image—which is also in the range $[0,1]$—with the color value. We resize the images in both domains into $32 \times 32$ resolution. Both domains contain 60,000 training samples each.

**Neural Networks.** Since MNIST digits are of very low resolution compared to the AFHQ and CelebA-HQ datasets, we use much smaller and simpler neural architectures to represent the generator $q$, content encoder $e_C$, style encoder $q_S^{(n)}$, and the discriminator $d^{(n)}$. Mainly, both $e_C$ and $e_S^{(n)}$ are represented by linear layers, i.e., matrices of dimensions $\widehat{d}_C \times \widehat{d}_C$ and $\widehat{d}_S \times \widehat{d}_S$, respectively. The architecture of $q$ and $d^{(n)}$ are shown in Table 3 and 4. In the tables, Conv represents convolutional layer, BN represents batch normalization, l-ReLU represents leaky ReLU activation with slope 0.2, and Up represents nearest neighbor upsampling layer, and dropout represents dropout with probability 0.25.

| Layer | Description |
|---|---|
| Linear | Concatenates content and style, maps to $128 \times 2 \times 2$ |
| Conv, BN, l-ReLU, Up | 128 filters of $3 \times 3$, stride 1, padding 1, upsample 2 |
| Conv, BN, l-ReLU, Up | 128 filters of $3 \times 3$, stride 1, padding 1, upsample 2 |
| Conv, BN, l-ReLU, Up | 64 filters of $3 \times 3$, stride 1, padding 1, upsample 2 |
| Conv, BN, l-ReLU, Up | 64 filters of $3 \times 3$, stride 1, padding 1, upsample 2 |
| Conv, Tanh | 3 filters of $3 \times 3$, stride 1, padding 1 |

Table 3: Architecture of $q$.

| Layer | Description |
|---|---|
| Embedding | Embeds domain index into $32 \times 32$ spatial dimensions |
| Concatenate | (input image and domain embedding) $4 \times 32 \times 32$ tensor |
| Conv, l-ReLU, dropout | 64 filters of $3 \times 3$, stride 2, padding 1 |
| Conv, l-ReLU, dropout, BN | 128 filters of $3 \times 3$, stride 2, padding 1 |
| Conv, l-ReLU, dropout, BN | 256 filters of $3 \times 3$, stride 2, padding 1 |
| Linear, Sigmoid | Fully connected layer, maps to 1 output |

Table 4: Architecture of $d^{(n)}$.

**Hyperparameter Setting.** We use Adam optimizer with an initial learning rate of $0.0001$ and parameters $\beta_1 = 0.5, \beta_2 = 0.999$. We use a batch size of $128$. We use $\ell_1$ regularization with $\lambda = 0.15$. The models are trained for 500 epochs.

**Result.** Fig. 6 shows the result of generating samples across the two domains for the same content input for the case of $\lambda = 0.15$ (i.e., Fig. 6(a)) and $\lambda = 0$ (i.e., Fig. 6(b)). Image in the $n$th row and $i$th column is generated by combining content $c_i$ with style $s_i^{(n)}$, where $n = 1, 2$. It can be observed that the proposed method generates the same digit for the same content information, with varying digit styles (color and rotation) in their respective domains. However, when the sparsity regularization is not used, the digit identity is not preserved across domains due to the content-leakage issue (see, e.g., the first, third, and fifth columns).

Fig. 7 shows the result of image generation for the case of $\lambda = 0.15$ (i.e., Fig. 7(a, b)) and $\lambda = 0$ (i.e., Fig. 7(c, d)) . Image in the $i$th row and $j$th column is generated by combining content $c_i$ and $s_j^{(n)}$ for a given domain $n$. Hence each row contains the same content and each column contains the same style. Fig. 7(a,b) shows that content and style components are correctly learnt by the proposed method, i.e., changing the content corresponds to changing the digit, whereas changing the style corresponds to changing the color and rotation in colored MNIST and rotated MNIST, respectively. However, when the sparsity regularization is not used, Fig. 7(c,d) shows the issue of content leakage into the extracted style information. Specifically, chaning the style component results in not only the change of color and rotation, but also the digit identity themselves. This validates Theorem 3.4 and the discussions in Sec. 3.2.

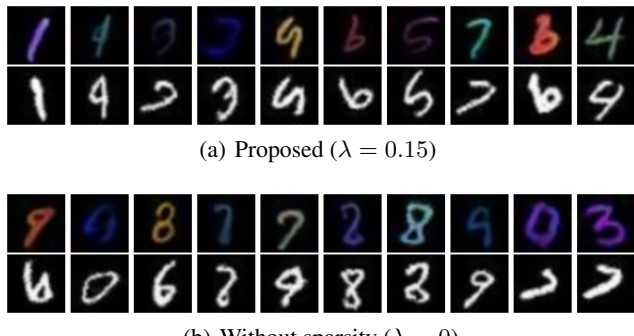

(a) Proposed ($\lambda = 0.15$)

(b) Without sparsity ($\lambda = 0$)

Figure 6: Result of sample generation across the two domains when the content is fixed. Images in each column was generated using the same content (digit identity). The first row is expected to have various colors and the second various rotations. Ideally, the digits in the two rows of the same column should be the same.

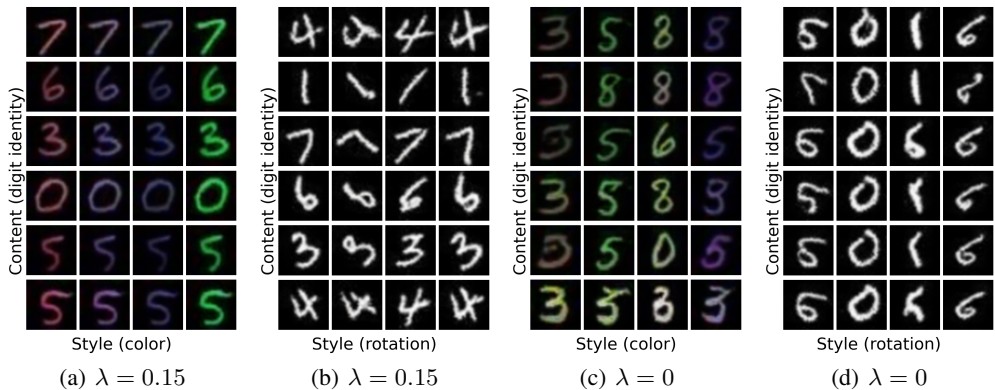

Style (color)     Style (rotation)     Style (color)     Style (rotation)

(a) $\lambda = 0.15$     (b) $\lambda = 0.15$     (c) $\lambda = 0$     (d) $\lambda = 0$

Figure 7: Result of generation by varying both the content and style latent codes. Content is the digit identity, whereas style is the color and rotation in the colored MNIST and rotated MNIST, respectively. Average FID for the proposed method: 22.08; Average FID without sparsity regularization: 24.08.

## H.2 MULTI-DOMAIN TRANSLATION.

In this section, we present qualitative results for multi-domain translation tasks by the proposed method and the baselines.

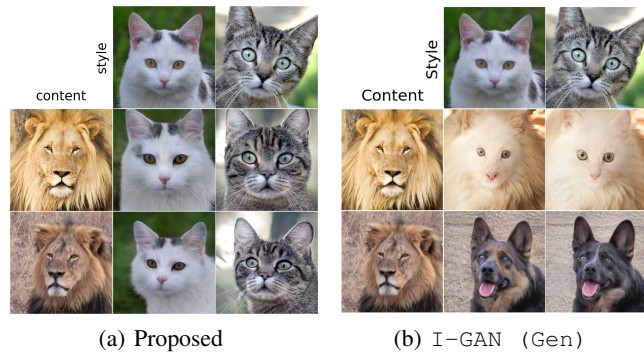

(a) Proposed     (b) I-GAN (Gen)

Figure 8: Translation Task: Combining content (pose) with styles of the image in the top row (the cat domain)

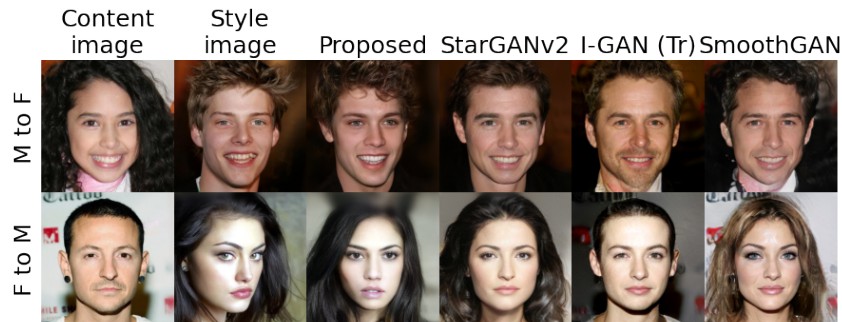

Figure 9: Target image-guided translation between different pairs of domains. Content (pose) from the image in the first column is combined with the style (appearance) of the image in the second to generate the translated images.

Fig. 8 (a) and (b) shows guided translation from wild domain ($n = 3$) to cat ($n = 1$). The first column contains two samples $x_i^{(3)}, i \in [2]$ in the wild domain, from which the content $\widehat{c}_i^{(3)}, i \in [2]$ is extracted using GAN inversion. The first row contains two samples $x_j^{(1)}, j \in [2]$ in the cat domain, from which the styles $\widehat{s}_j^{(1)}, j \in [2]$ are extracted. Finally, the translated image are synthesized using $\widehat{q}(\widehat{c}_i^{(3)}, \widehat{s}_j^{(1)})$. One can see that the translated images appear consistent with their respective input contents (i.e., pose of wild) and styles (i.e., type of dog) for the proposed method, whereas `I-GAN (Gen)` seems to ignore the style information.

Fig. 9 shows the result of target domain image-guided translation between Male and Female of the CelebA-HQ datasets. Here, the first and second columns show the images in the source domain $x^{(s)}$ and target domain $x^{(t)}$, respectively. We, first, extract the content $\widehat{c}^{(s)}$ from $x^{(s)}$ and style from $s^{(t)}$ using GAN inversion. Then, the translated image $x^{(s \to t)}$ is generated using $q(c^{(s)}, s^{(t)})$. Fig. 9 shows that the proposed method generates translations that are more consistent with the style (appearance) of the target domain images compared to the baselines (e.g., see first row).

Fig. 10 shows the result of randomly sampled translations between different pairs of domains in the AFHQ and CelebA-HQ datasets. Here, the first column shows the images in the source domain $x^{(s)}$. We extract the content $\widehat{c}^{(s)}$ from $x^{(s)}$ using GAN inversion. Then, a style vector $s^{(t)}$ is randomly sampled in the target domain $t$ to generate the translated image $x^{(s \to t)} = q(\widehat{c}^{(s)}, s^{(t)})$. One can see that the translation quality is competitive with the baselines. Moreover, in some cases, the translations produced by the proposed method appear more natural compared to the baselines (e.g., see ears of wild animals in the "dog to wild" translation task).

### H.3 MULTI-DOMAIN GENERATION.

In this section, we present additional qualitative results for multi-domain data generation by the proposed method and the baselines.

Fig. 11 shows the generated images by varying content and style components in various domains for the proposed method and the baseline. Image in the $i$th row and $j$th column is generated by combining content $c_i$ and $s_j^{(n)}$ for a given domain $n$. Hence each row contains the same content and each column contains the same style. One can see that the proposed method has successfully learnt to disentangle content (pose of the object) and style (appearance of the object) for all domains. Whereas, for the baseline `I-GAN`, it seems that the style component is not properly learnt.

Fig. 12 shows the qualitative results for the CelebA dataset. As in Fig. 3, for the $j$th column associated with domain $n_j$, we generate three different styles $s_i^{(n_j)}, i \in [3]$. Then, the image $x_{i,j}$ in the $i$th row and $j$th column is generated by $q(\overline{c}, s_i^{(n_j)})$, using fixed content $\overline{c}$. One can see that the proposed method generates content (pose) aligned samples in different domains with varying

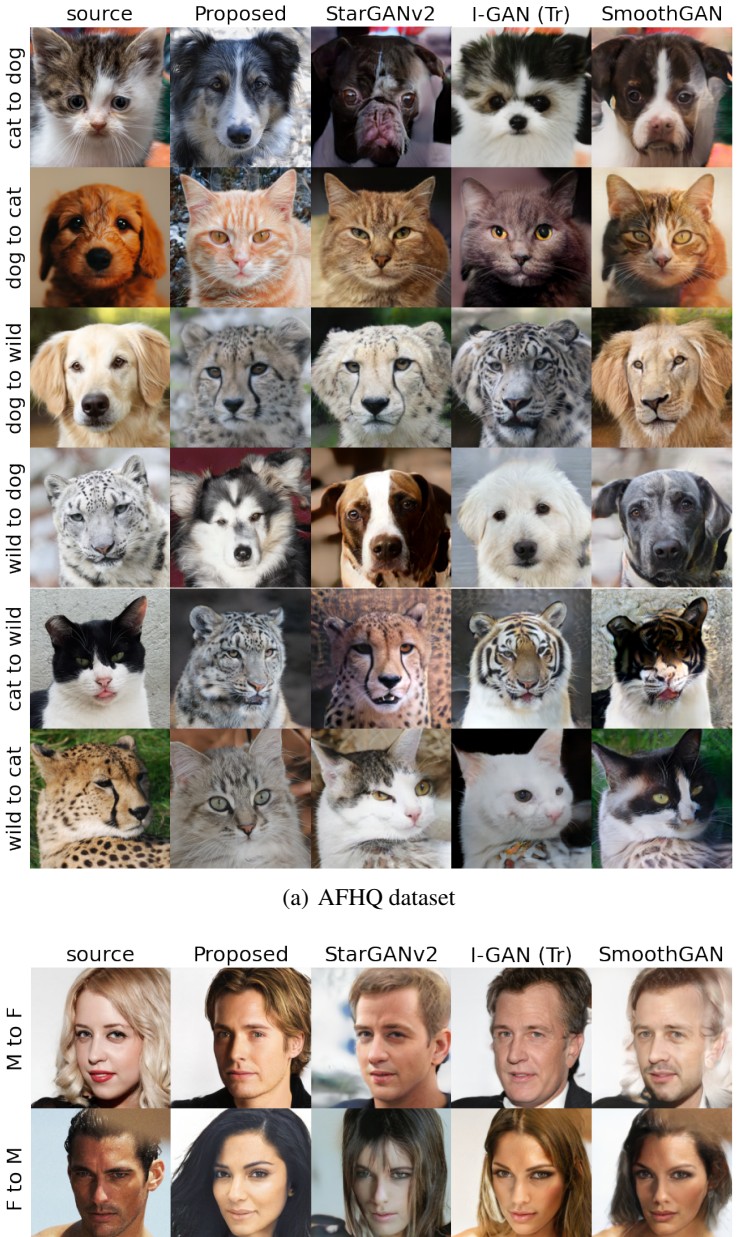

(a) AFHQ dataset

(b) CelebA-HQ dataset. M: Male, F: Female

Figure 10: Randomly sampled style based translation between different pairs of domains. Content (pose) from the image in the first column is combined with a randomly sampled style in the target domain to generate the translated images.

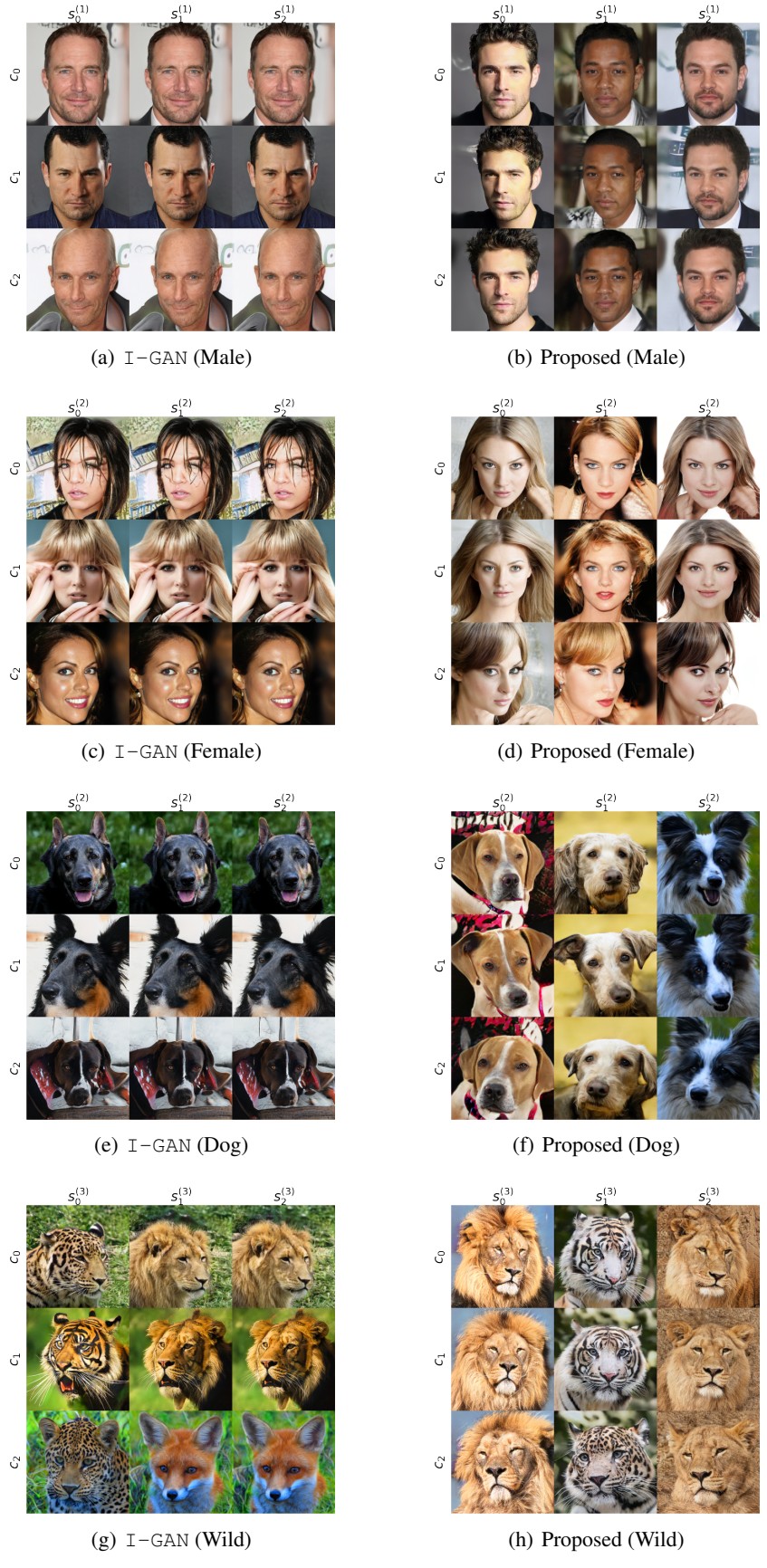

Figure 11: Different contents $c_i$ combined with different styles $s_j^{(n)}$ in different domains $n$.

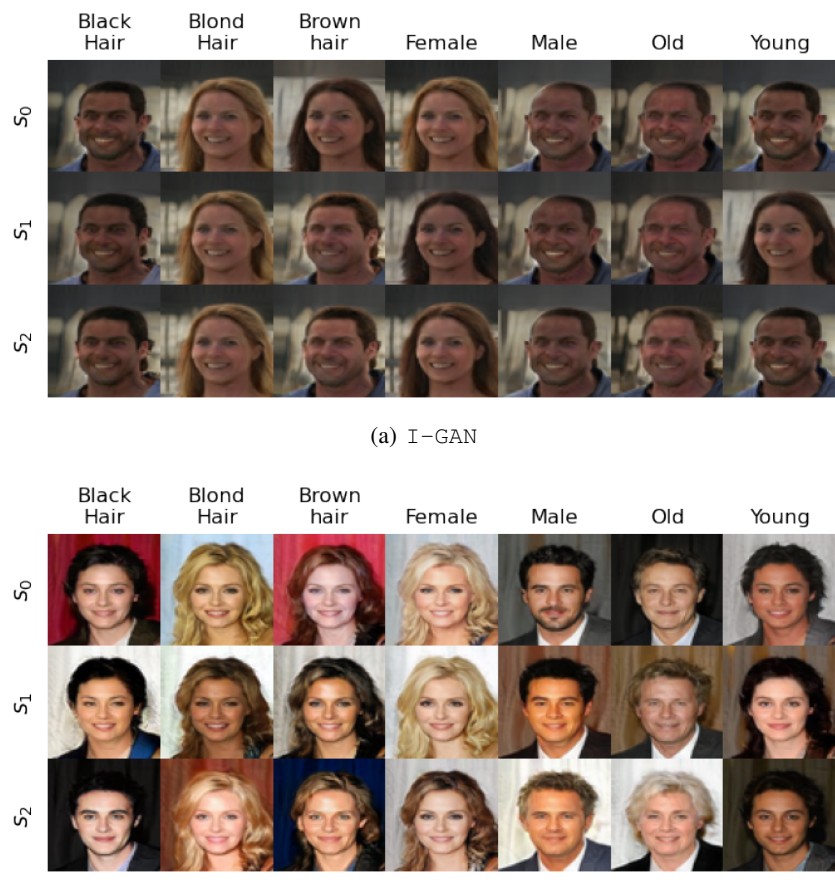

Figure 12: Qualitative results for the CelebA dataset. All images are generated using the same content $\overline{c}$ but different styles $s_i^{(n)}, i \in [3], n \in [7]$, i.e., image $x_{i,j}$ in the $(i,j)$-th location in the grid is $x_{i,j} = q(\overline{c}, s_i^{(j)}), \forall i, j \in [3] \times [7]$.

style. However, the baseline `I-GAN` seems to generate the same image for a given domain, as in other experiments.

### H.4 Other Sparsity promoting regularization

As mentioned in Section 6, any sparsity promoting regularization other than $\ell_1$ can also be used in order to avoid the content leakage problem when the latent dimensions are unknown. In this section, we show that using $\ell_p$-norm with $p < 1$ can be similarly effective. To that end we use $\lambda \|e_S^{(n)}(r_S^{(n)}\|_{\frac{1}{2}}$. We set $\lambda = 0.3$ and fix all other setting to be the same as described in Sec. G.

Fig. 13 shows the result of image generation in the three domains (i.e., cat, dog, and wild) of the AFHQ dataset by varying content and style latent variables. Similar to the results obtained by using $\ell_1$-norm based regularization, the content (pose) and the style (appearance) information is correctly learned.

Fig. 14 shows the result of image generation across different domains for the same content information. Image in the $n$th row and $i$th column is generated using content $c$ and style $s_i^{(n)}$. One can observe that samples across different domains are content-aligned.

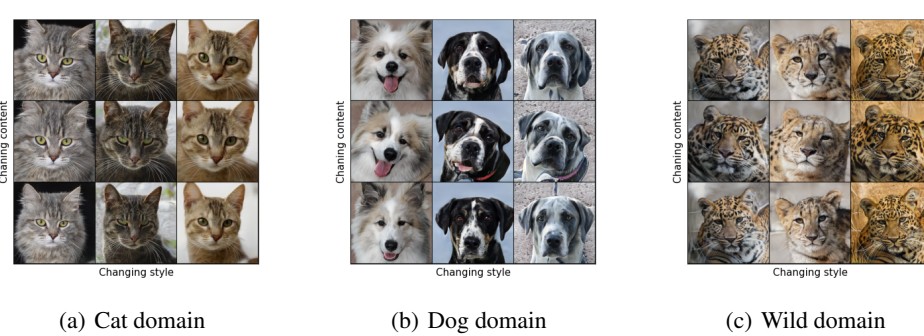

(a) Cat domain      (b) Dog domain      (c) Wild domain

Figure 13: Generation result for $\ell_{\frac{1}{2}}$-norm based sparsity regularization on AFHQ dataset. FID: 6.61

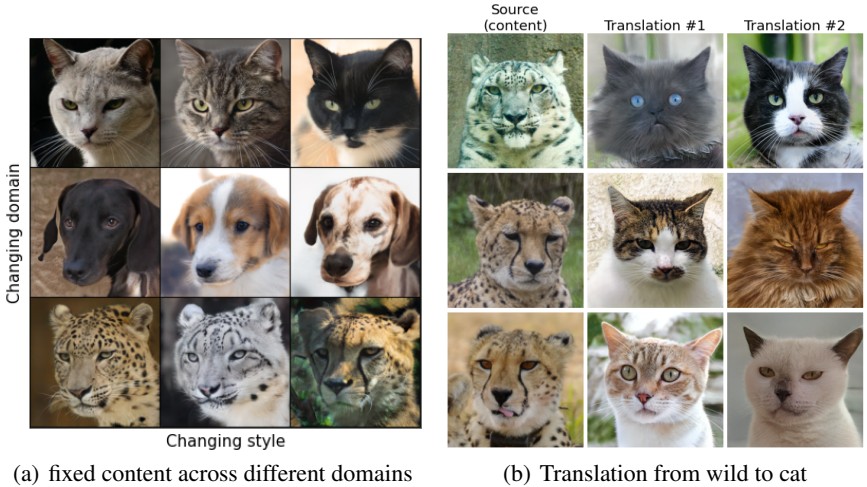

(a) fixed content across different domains      (b) Translation from wild to cat

Figure 14: (a) Generated images across different domains for the same content information in the AFHQ dataset. (b) Image translation from wild domain to cat domain of the AFHQ dataset

### H.5 Conditional Image generation.

Fig. 15 and 16 shows random samples from each domain. That is, each image is generated using randomly sampled content and styles for the proposed method and I-GAN. Whereas, for StyleGAN-ADA and Transitional-cGAN, each image in a given domain is generated using

randomly sampled latent vector. One can see that the compared to the class conditional generative models `StyleGAN-ADA` and `Transitional-cGAN`, the proposed multi-domain generative model does not incur any loss in the visual quality. However, due to disentangled latent representations, the proposed method enables content/style controlled generation and domain translation.

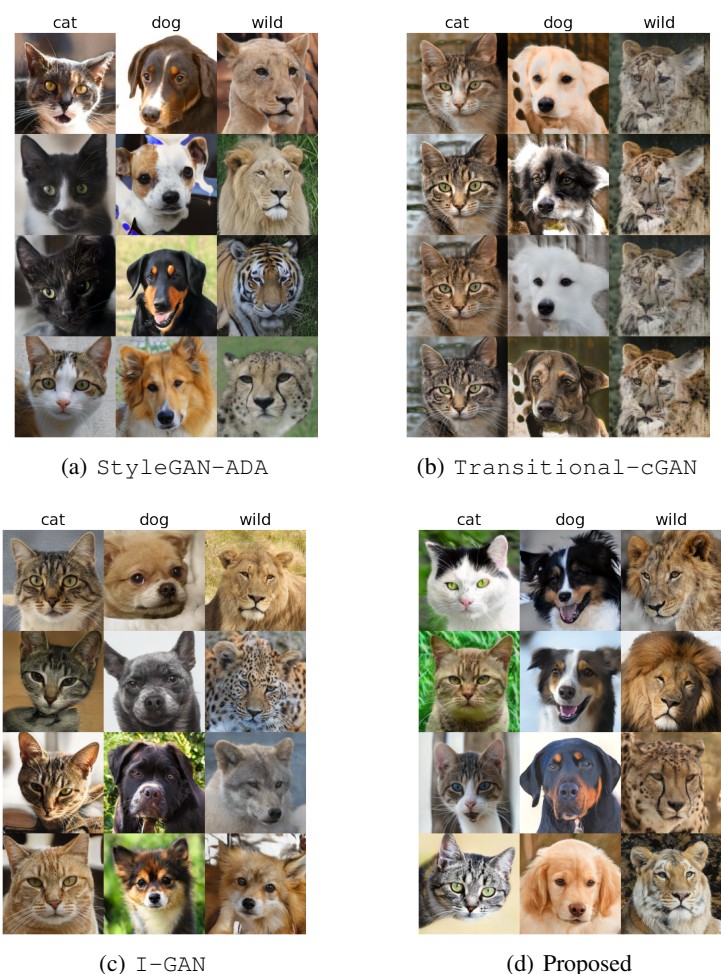

Figure 15: Class conditional image generation by all methods for AFHQ dataset.

## H.6 UNDERESTIMATED LATENT DIMENSION

In this section, we provide some discussions on the implications of underestimated latent dimensions $d_C$ and $d_S$. Generally, when $\hat{d}_C < d_C$ or $\hat{d}_S < d_S$, solving Problem (4) cannot identify the content and the style parts. In addition, a feasible solution to Problem (4) may not even exist. To clarify, we provide some intuitive explanation for the following three cases:

1. $\hat{d}_C + \hat{d}_S < d_C + d_S$ : In this case, a differentiable injective map $\boldsymbol{f}$ does not exist from $d_C + d_S$-dimensional manifold, $\mathcal{X}$, to $R^{\hat{d}_C + \hat{d}_S}$. Considering our implementation in Problem (6), the generated data may not be able to match the data distribution $\mathbb{P}_{\boldsymbol{x}^{(n)}}$ well since the generated data $\boldsymbol{q}\left(\boldsymbol{e}_C(\boldsymbol{r}_C), \boldsymbol{e}_S^{(n)}(\boldsymbol{r}_S^{(n)})\right)$ will have to live on a lower dimensional manifold than $\mathcal{X}$.

2. $\hat{d}_C < d_C$ but $\hat{d}_C + \hat{d}_S \geq d_C + d_S$ : The content dimension will be insufficient to capture all the content information. As such the style part might be a mixture of content and style.

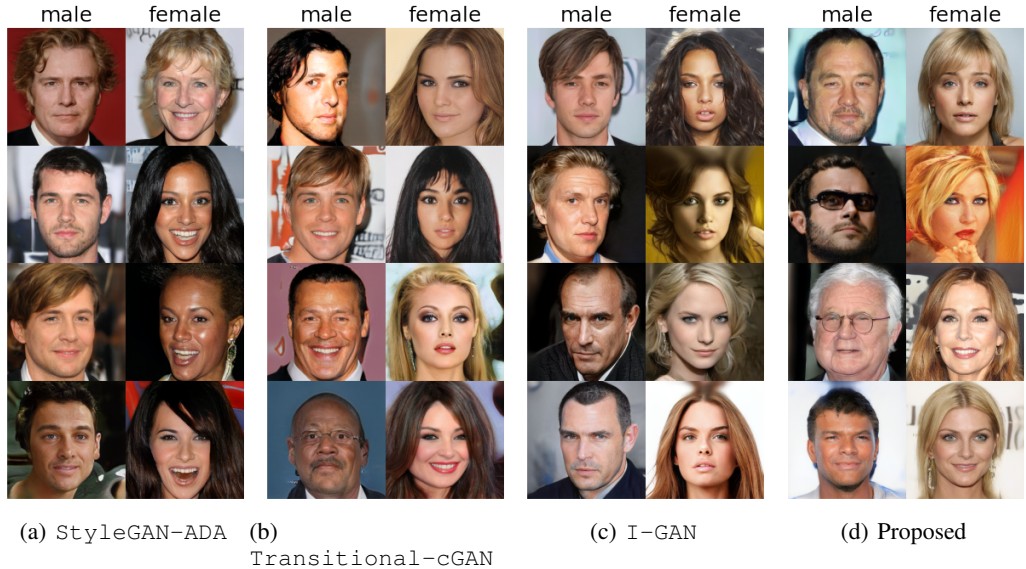

| male | female | male | female | male | female | male | female |

(a) `StyleGAN-ADA`  (b) `Transitional-cGAN`  (c) `I-GAN`  (d) Proposed

Figure 16: Class conditional image generation for CelebA-HQ dataset.

3. $\widehat{d}_\text{S} < d_\text{S}$ but $\widehat{d}_C + \widehat{d}_S \geq d_C + d_S$ : The style dimension will be insufficient to capture all the style information. But Assumption 3.2 still ensures that the style does not leak into the extracted content part. Hence, the style variability may not be captured by the generated images.

Fig. 17 shows generated samples by varying the content and style dimensions for the cat domain of the AFHQ dataset. We show results corresponding to each of the above cases. Empirical tests showed that $d_C > 8$ and $d_S > 8$ are reasonable settings for the AFHQ dataset, which we will treat as the "ground truth". For each row, we fix the content part $c = e_C(r_C)$ (i.e., pose of the cat) and randomly sample different styles $s^{(1)} = e_\text{S}^{(1)}(r_\text{S}^{(1)})$ where $r_\text{S}^{(1)} \sim \mathcal{N}(0, I_{\widehat{d}_\text{S}})$ to generate the images $x^{(1)} = q(c, s^{(1)})$. We set $d_C = 8$ and/or $\widehat{d}_\text{S} = 8$ according to the specific case that we tested.

Fig. 17(a) corresponds to Case 1. One can see that although content-style disentanglement appears satisfactory, the FID attained is slightly worse than the other two cases, which could be due to insufficient latent dimension.

Fig. 17(b) corresponds to Case 2. One can see that changing the style component seems to change both pose and appearance, which means that extracted style is a mixture of the content and style.

Fig. 17(c) corresponds to Case 3. Here, changing the style component has very little effect on the appearance of the cat which corroborates our intuition that the style information is not fully captured.

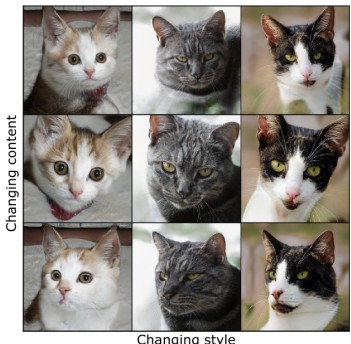
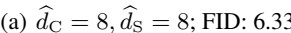
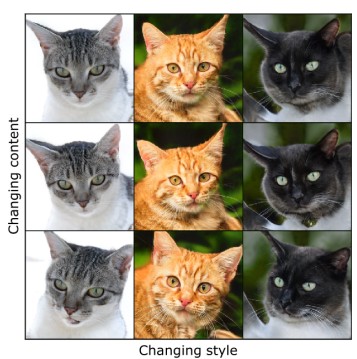

(a) $\widehat{d}_{\mathrm{C}} = 8, \widehat{d}_{\mathrm{S}} = 8$; FID: 6.33

(b) $\widehat{d}_{\mathrm{C}} = 8, \widehat{d}_{\mathrm{S}} = 504$; FID: 6.01

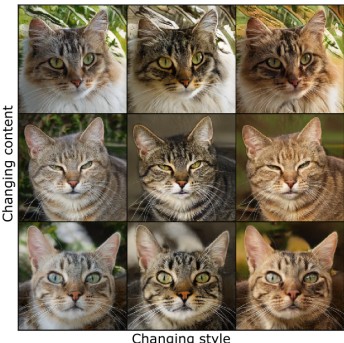

(c) $\widehat{d}_{\mathrm{C}} = 504, \widehat{d}_{\mathrm{S}} = 8$; FID: 6.08

Figure 17: Result of using small latent dimensions for AFHQ dataset.

