# OpenReview forum: "Content-Style Learning from Unaligned Domains: Identifiability under Unknown Latent Dimensions"
_ICLR.cc/2025/Conference — ICLR 2025 Poster_

### Official Review · Reviewer_EZhh · 2024-10-25

**Soundness:** 3
**Presentation:** 4
**Contribution:** 3
**Rating:** 5
**Confidence:** 4

**Summary:**

This paper introduces an analytical framework via cross-domain latent distribution matching(LDM), which establishes content-style identifiability under substantially more relaxed conditions.

**Strengths:**

The paper is well written and the contribution is clear.

**Weaknesses:**

The author seems to slip the main result part (3.2) too quickly (with no more than half a page). I am unsure what the key difference between your approach and other low-rank techniques is. In addition, there seems to exist a gap between the optimization problem (4a)(4b)(4c) and the implementation. (I mean: why not to use the formulation in implementation section as the theoretical modeling approach in the methodology part?) There are some writing issues in the paper too, e.g. you can not directly quote (4) , you should use 'according to formulation (4)'.(e.g. the writing in Theorem 4.1)

**Questions:**

1. Can you further explain the reasonability using Assumption 3.1? Can you relax it? If not, what are the difficulties of relaxing this assumption? Furthermore, in what real-world scenarios might this assumption be violated?
2. Can you explain the gap between your method in 4(a) and the way you do implementation? What is the motivation for not using (4) directly? Any trade-offs or limitations of their implementation approach compared to directly using formulation (4).

---

> ### Author Response · Authors · 2024-11-20
>
> ### Response 1/2
>
> &nbsp;
>
> **[Length of Main result]**
>
> Please note that the entire Section 3 is our main result, which is more than one and half pages.
> Subsection 3.2 uses “Main result: identifiability without latent dimension knowledge” is part of our main results (Theorem 3.3 and Theorem 3.4). We hoped to articulate the importance of this subsection, but this was indeed a little confusing. Nonetheless, we hope to clarify that the main result is the **entire Sec. 3 and its proofs**. The proof of Sec. 3 is a little lengthy and thus we had to put it in the appendix. We have added a more obvious pointer to make sure that interested readers can easily localize the detailed proofs in the appendix. **We have also changed the title of subsection 3.2 to avoid confusion.**
>
> &nbsp;
>
> **[Key difference between our approach and low-rank techniques]**
>
> Regarding the comment on “low-rank techniques”, please note that low-rank matrix/tensor factorization methods are often used for tackling linear mixture learning models, e.g., $\boldsymbol{x}=\boldsymbol{A}\boldsymbol{z}$, where x is the observed data, A is the mixing system, and z is the latent code. In our case, since we have nonlinear mixtures, i.e., $\boldsymbol{x}=\boldsymbol{f}(\boldsymbol{z})$ where $\boldsymbol{f}(\cdot)$ is a nonlinear mixing system, the low-rank models do not apply.
>
> &nbsp;
>
> **[Assumption 3.1(block independence)]**
>
> This is an important point. Thanks for bringing this up.
>
> Block independence condition in Assumption 3.1 is a mild assumption that states that the style block is independent of the content block. An example of the content and style being block-independent is as follows: Consider the animal face experiments in the manuscript, where the pose of a dog is the content and the breed is the style. The independence of content and style blocks is implied by the fact that any dog breed (e.g., German Shepherd or Chihuahua) can be combined with any pose (front-facing, left-facing, etc). That is, the choice of pose does not affect the choice of the dog breed---nor does the breed affect the pose. This assumption is commonly seen in the literature [1, 2].
>
> Assumption 3.1 is used in the proof of our identifiability Theorems 3.3 and 3.4 for identifying the style information. Hence it can be relaxed if we only want to identify the content information but not the style information.
>
> This assumption may be violated if the content and style affect each other. For example, in the above example, if some breed of dog is physically incapable of sitting on a certain pose, the breed and the pose will no longer be independent of each other.
>
> The reasoning behind block independence was briefly discussed after Assumption 3.1 in the original manuscript. **We have further added a remark after Theorem 3.3 explaining when it can be relaxed.**

---

> ### Author Response · Authors · 2024-11-20
>
> ### Response 2/2
>
> &nbsp;
>
> **[Why not implement Problem (4)]**
>
> This is a valuable comment that merits further clarification.
> We briefly discussed the reason why Problem (4) (or its regularized version Problem (5)) was not implemented directly, in lines 262–272 and lines 306-316 in the original manuscript. Simply speaking, Problem (4) is cumbersome in terms of optimization. However, Problem (6) can effectively bypass many of Problem (4)’s challenges, with the price of working under slightly more restrictive conditions.
>
> To be more specific, implementing Problem (4) would require several complex elements: independence regularization, latent distribution matching, and injectivity promotion. Independence regularization typically involves non-trivial approaches like minimax optimization [2]. Likewise, latent distribution matching usually relies on GAN-based minimax optimization [3]. Injectivity, in turn, requires either autoencoder-based reconstruction loss [2] or entropy regularization [4]. In contrast, the reformulation in Problem (6) simplifies the process by only requiring distribution matching via GAN, while independence regularization and invertibility constraints are implicitly addressed, as shown in Theorem 4.1. Therefore, we chose to implement Problem (6) instead of Problem (4).
>
> We first presented Problem (4) as it explicitly outlines the conditions that are sufficient for identifiability (i.e., injectivity, distribution matching, and independence). It presents the more general conditions under which content and styles are identifiable. Problem (6) is almost equivalent to Problem (4), but it has the same identifiability under slightly more restrictive conditions (i.e., bijectivity of $\widehat{\boldsymbol{q}}$ and simply connectedness of $\mathcal{C}$ and $\mathcal{S}$). However, Problem (6) is way easier to implement (also see our discussions in Lines 306-316 in the original manuscript). That is, Problem (4) enjoys more general conditions and Problem (6) is a reformulation that we made for efficient implementation---both have their own values to be presented.
>
> **We have revised Sec. 4 in the new version to articulate and highlight the above points; please see Remark 4.1.**
>
> &nbsp;
>
> **[Quoting an optimization problem]**
>
> We appreciate the reviewer’s suggestion to distinguish between optimization problems and other equations. **Accordingly, we have revised the manuscript to use “Problem (X)” specifically when referring to optimization problems.**
>
> &nbsp;
>
> ### References
>
>
> *[1] Eastwood, Cian, et al. "Self-supervised disentanglement by leveraging structure in data augmentations." arXiv preprint arXiv:2311.08815 (2023).*
>
> *[2] Lyu, Qi, et al. "Understanding latent correlation-based multiview learning and self-supervision: An identifiability perspective." arXiv preprint arXiv:2106.07115 (2021).*
>
> *[2] Goodfellow, Ian, et al. "Generative adversarial nets." Advances in neural information processing systems 27 (2014).*

---

> ### Author Response · Authors · 2024-11-29
> **Gentle Reminder**
>
> Thank you for your time and feedback. We believe that our response and revision address your questions and concerns. We hope that you could re-evaluate your score based on our response and the revised manuscript.
>
> If you have any further questions or comments, please let us know.

---

### Official Review · Reviewer_dub2 · 2024-11-01

**Soundness:** 3
**Presentation:** 3
**Contribution:** 3
**Rating:** 8
**Confidence:** 2

**Summary:**

Summary:
This paper studies causal identification of latent factors content and style in the cross-domain learning problems. Specifically, existing works assume that all latent components are independent to each other. Such an assumption is strict and could not be fulfilled in realistic scenarios. In this paper, the authors propose a cross-domain latent distribution matching framework which does not require strict independence between different latents. Moreover, the proposed framework can ensure content style identification can be free from the specific latent dimensions which further ease the assumption for tackling the problem. Through both theoretical and empirical justifications, the authors carefully testified the effectiveness of the proposed method.

**Strengths:**

Strengths:
- This paper is well-written, well-motivated, and easy-to-follow. Both the experiments and theoretical justification are intuitive and convincing.
- The contribution of easing independent assumption is helpful for developing simple and efficient causal models.
- A sparsity-regularized implementation is proposed to solve cross-modal learning efficiently.

**Weaknesses:**

Weaknesses:
- The proposed target in Equation 4 aims to minimize the norm of style latents by enforcing distribution matching across content and block-independence between content and style. However, it could lead to a degenerate solution when there is misidentification between content and style, and some content features already achieves small norm. Could you explain how can the proposed identifiability result can avoid the problem from happening?
- The computational efficiency of the framework is unknown. Only numerical result of generation is provided, there is no quantitative comparison to show the efficiency compared to other baseline methods. Moreover, why sparsity regularization is needed for training GAN needs a justification.
- Missing related references:
Hong et al., Improving Non-Transferable Representation Learning by Harnessing Content and Style, in ICLR 2024.
Huang et al., Harnessing Out-Of-Distribution Examples via Augmenting Content and Style, in ICLR 2023.
Dai et al., Disentangling writer and character styles for handwriting generation, in CVPR 2023.

**Questions:**

Please see the weaknesses part.

---

> ### Author Response · Authors · 2024-11-20
>
> ### Response 1/1
>
> &nbsp;
>
> **[Content features with small norm]**
>
> This is an interesting point, and in fact it is a situation that our formulation can avoid.  Sparsity regularization results in identifiability regardless of the size of the norm or sparsity of the content features. Let us explain.
>
> The proof of Theorem 3.4 sheds light onto the reason why no misidentification occurs. In summary, the sparsity regularization ensures that the learned style cannot contain any content information even when the sparsity dimension is over-estimated. The distribution matching constraint ensures that the learned content cannot contain any style information. Finally, the injectivity of $\boldsymbol{f}$ ensures that all the latent information is preserved among the learned content and style. Hence, the learned content and style contains exactly the content and the style, respectively, leading to identifiability.
>
> The fact that the size of the norm of the content variables does not affect identifiability is also reflected by the statement of Theorem 3.4. Specifically, no extra assumption on content’s norm is needed in order to ensure identifiability. This means that the proposed formulation ensures identifiability regardless of the size of the norm of the content features.
>
> In our original manuscript, the paragraph in Section 3.2 Line 242 to Line 249 explains the insights regarding the sparsity regularization. **We have also added the above point after Theorem 3.4 in the revised manuscript for clarification.**
>
>
> &nbsp;
>
> **[Computational Efficiency]**
>
> Indeed, computational efficiency is an important consideration. Please note that Table 2 has the training time of our method. **Following the reviewer’s comments, we have also added a column of training time for the generation methods.**
>
>
> Basically, the training time of the proposed method is similar to those of the baselines. This is because sparsity regularization, and content and style encoder optimization do not take significant compute resources.
>
> &nbsp;
>
> **[Sparsity Regularization for training GAN?]**
>
> The need for sparsity regularization is based on our formulation in (4) and the proof in Theorem 3.4. The $\ell_1$ norm is used to approximate the cardinality minimization term in constraint (6b). It is the critical part to ensure that content does not leak into the learned style representations, when the latent dimensions are over-estimated. Please see our discussions in Line 242 - line 249 in the original manuscript.
>
> As shown in our experiments (see Figs. 2(b) and 2(c)), if this regularization (i.e., $\lambda=0$) is not used, we observed that the extracted style information could contain a mix of both content and style.
>
> &nbsp;
>
> **[References]**
>
> Thanks for providing these references. **We have discussed them in the “related works” part.**

---

> ### Author Response · Authors · 2024-11-29
>
> Thank you for your time and feedback. We hope that our response and revision addressed your questions and concerns.
> If you have any further questions or comments, please let us know.

---

> ### Comment · Reviewer_dub2 · 2024-11-30
> **Thanks for your response**
>
> Dear Authors,
>
> Thanks for the detailed response and addressing all my concerns. I think this paper is quite extensive and solid. After the rebuttal, the clarity and motivation are further improved. So I raise my score to 8. Best of luck in battling with remaining concerns.
>
> Sincerely,
> Reviewer.

---

> > ### Author Response · Authors · 2024-11-30
> > **Thanks for your response**
> >
> > Thank you for your response and kind wishes.
> > We greatly appreciate your time and effort spent in reviewing and refining our work.

---

### Official Review · Reviewer_teKB · 2024-11-03

**Soundness:** 3
**Presentation:** 3
**Contribution:** 3
**Rating:** 8
**Confidence:** 3

**Summary:**

This paper tackles the problem of identifiability of latent content and style variables from unaligned multi-domain data. A new approach based on cross-domain latent distribution matching (LDM) is presented, which removes restrictive assumptions from prior work regarding component-wise independence of latent variables and knowledge of content and style dimensions. The LDM formulation is implemented as a regularized multi-domain GAN loss with coupled latent variables. In addition to theoretical contributions on these topics, the authors present experimental results on image translation and generation tasks.

**Strengths:**

- The paper provides a good overview of background and related work on multi-domain content-style modeling, content-style identifiability, and the challenge of knowing the content and style dimensions apriori.
- The contributions in this paper appear to be novel and technically sound from both a theoretical and experimental standpoint.
- The proposed approach in this paper appears to be reproducible, given the provided source code and experimental details in the paper.
- The experimental results show that the proposed approach performs well on multi-domain image translation and generation tasks.

**Weaknesses:**

- Because the proposed approach is implemented as a GAN, it seems likely that this approach will be most suitable for images (and possibly audio), and may be problematic for discrete data modalities, particularly text.
- The authors don’t directly mention or address potential instabilities that may result from adversarial training of their GAN-based approach.

**Questions:**

Can the authors please add some remarks regarding the potential weaknesses mentioned above, regarding the suitability of their approach for text, and regarding the potential instability of adversarial GAN training used in their approach?

---

> ### Author Response · Authors · 2024-11-20
>
> ### Response 1/1
>
> &nbsp;
>
> **[Extension to discrete data modalities]**
>
> Indeed, our theoretical analysis is based on the continuous domain $ \mathcal{C}, \mathcal{S},$  and $\mathcal{X}$.  Moreover, as rightly noted by the reviewer, our GAN based implementation might not be suitable for discrete and sequential data modalities such as text. Therefore, extending our work to discrete modalities would require extension of our theoretical analysis. The implementation also has to be re-designed. These challenges present interesting future work. We have mentioned them under “Limitations” on Sec. 7 in the revised version, which is as follows:
>
> *“Additionally, our work is also limited to continuous data modalities like images, audio, etc. Discrete data modalities like text will require extension of both theory and implementation. This challenge presents another important future work.”*
>
> &nbsp;
>
> **[GAN Optimization Instabilities]**
>
> The reviewer is correct: GAN-based implementations can be sensitive to small changes in hyperparameters  as well as randomness in initialization. In our experiments, we did not quite observe instability due to random initialization. However, we did observe convergence issues when the number of layers of content and style encoders were large (>5). We used only 3 layer fully connected neural networks for both content and style encoders to overcome this issue. In the revised version, we have included the above discussion in Section G.7 “Regarding GAN Stability”.
>
> We have also added on remark under “Limitations” on Sec. 7 in the revised version, which is as follows:
>
> *“Finally, another limitation of our work is that the proposed method is based on a GAN based framework which is known to be unstable during training. Therefore, novel implementation methods based on more stable distribution matching modules such as flow matching are also of interest as a future work.”*

---

> > ### Comment · Reviewer_teKB · 2024-11-26
> > **Rebuttal response**
> >
> > I have read the authors' response to my review, as well their responses to the other reviewers, including the revised version of the paper. I thank the authors for a thorough response to my concerns, as well as the concerns raised by the other reviewers. At present my rating remains an 8 (accept, good paper).

---

> > > ### Author Response · Authors · 2024-11-29
> > >
> > > Thanks for your response and feedback.

---

### Official Review · Reviewer_r56a · 2024-11-04

**Soundness:** 3
**Presentation:** 3
**Contribution:** 3
**Rating:** 6
**Confidence:** 3

**Summary:**

The authors try to attach the problem of identifying latent content and style variables from observations from un-aligned domains. The authors claim their main contribution as follows: (1) t restrictive assumptions such as component-wise independence of the latent variables can be removed; (2)  knowledge of the content and style dimensions is not necessary for ensuring identifiability. However, there are some inconsistencies between the main results and Theorem 4.1.

**Strengths:**

1. The writing of the paper is good. Most parts of the paper are easy to follow and understand.
2. The experimental results are good.

**Weaknesses:**

There may be some inconsistency between  the main results and Theorem 4.1. In the main results, it claims that $f$, $\gamma$, and $\delta$ are injective. However, in theorem, the $q$ is a bijective and $f=q^-1$, which implies that $f$ will be an bijective. As a result, $\gamma$ and $\delta$ becomes bijective and it implies the dimension of content and style is actually $\hat{d}_c$ and $\hat{d}_s$. This causes a contradiction in the theory.

The proof of Theorem 4.1 uses  Proposition E.1 from  (Zimmermann et al., 2021) https://arxiv.org/pdf/2102.08850. However, I did not find the Proposition in the reference. Can the authors fix the problem?

**Questions:**

1. In Theorem 4.1,  $q$ is a bijective and $f=q^{-1}$, thus trivially $f$ becomes an bijective, which somehow is inconsistent with the fact    that $f$ is a injective from $\mathcal{X}$ to $\mathbb{R}^{\hat{d}_{c}+\hat{d}_s}$ ,
   can the authors clarification?

   Particularly on the relation between $\mathcal{C}$, $\mathcal{S}$ and $\mathbb{R}^{\hat{d}_{c}+\hat{d}_s}$ as they are actually different spaces.
2. In Assumption 3.1, $N$ is the dimension of style variables, but somehow in later part of the paper it has different meaning, can the authors clarify? Similarly, the notation $n$ also has different meanings in different part of the paper.
3. Can you provide the correct citation for Proposition E.1?

---

> ### Author Response · Authors · 2024-11-20
>
> ### Response 1/1
>
> &nbsp;
>
> **[Bijectivity of q and f]**
>
> This is a nuanced point that is indeed worth clarifying. The bijectivity of f and the injectivity of f are actually **not inconsistent**. The reason is that the bijectivity/injectivity are defined w.r.t. domain and codomain. Let us explain.
>
> Note that we assume $\boldsymbol{q}$ to be bijective from $\widehat{\mathcal{C}} \times \widehat{\mathcal{S}}$ to $\mathcal{X}$. Therefore, in Theorem 4.1 (b), our assumption that $\boldsymbol{q}: \widehat{\mathcal{C}} \times \widehat{\mathcal{S}} \to \mathcal{X}$ is bijective implies that **$\boldsymbol{f}= \boldsymbol{q}^{-1}$, when defined w.r.t. $\mathcal{X} \to \widehat{\mathcal{C}} \times \widehat{\mathcal{S}}$ is bijective**. However, this is not inconsistent with the fact that **$\boldsymbol{f}$ is only injective when defined w.r.t. $\mathcal{X} \to \mathbb{R}^{\widehat{d}\_{\rm C} + \widehat{d}\_{\rm S}}$** since $\widehat{\mathcal{C}} \times \widehat{\mathcal{S}} \subset \mathbb{R}^{\widehat{d}\_{\rm C} + \widehat{d}\_{\rm S}}$.
>
> Nonetheless, as we mentioned, this is indeed quite nuanced and we take the responsibility of not articulating it more. We have revised the manuscript to clarify this point and given a pointer to  Section A.2; see footnote on Theorem 4.1.
>
> We have also added the following fact to Sec. A.2 to clarify this point.
>
> *“5. If $\boldsymbol{f}: \mathcal{X} \to \mathcal{Y}$ is bijective, then $\boldsymbol{f}: \mathcal{X} \to \mathcal{Z}$ is injective, where $\mathcal{Y} \subset \mathcal{Z}$, $\mathcal{Y} \neq \mathcal{Z}$.”*
>
> &nbsp;
>
> **[Relation between different spaces $\mathcal{C}, \mathcal{S}, \mathcal{X}, \mathbb{R}^{\widehat{d}\_{\rm C} + \widehat{d}\_{\rm S}}$]**
>
> The relationship between these spaces is indeed also nuanced and crucial for deeper understanding of the work. Let us explain:
> $\mathcal{C}$ and $\mathcal{S}$ are open sets in $\mathbb{R}^{d_{\rm C}}$ and $\mathbb{R}^{d_{\rm S}}$, respectively. Hence, $\mathcal{X} = \boldsymbol{g}(\mathcal{C} \times \mathcal{S})$ is a $(d_{\rm C} + d_{\rm S})$-dimensional manifold inside $\mathbb{R}^d$. Now, in Theorem 4.1 (b), by assuming $\boldsymbol{q}: \widehat{\mathcal{C}} \times \widehat{\mathcal{S}} \to \mathcal{X}$ to be bijective, we are effectively assuming that $\widehat{\mathcal{C}} \times \widehat{\mathcal{S}}$ is a  ($d_{\rm C} + d_{\rm S}$)-dimensional manifold inside $\mathbb{R}^{\widehat{d}\_{\rm C} + \widehat{d}\_{\rm S}}$. This is because $\widehat{\mathcal{C}} \times \widehat{\mathcal{S}} = \boldsymbol{q}^{-1} \circ \boldsymbol{g} (\mathcal{C} \times \mathcal{S})$ and $\boldsymbol{q}^{-1} \circ \boldsymbol{g}$, a composition of two bijections, is a bijection. Hence, although the ambient space $\mathbb{R}^{\widehat{d}\_{\rm C} + \widehat{d}\_{\rm S}}$ is higher dimensional compared to $\mathcal{C} \times \mathcal{S}$, the spaces $\widehat{\mathcal{C}} \times \widehat{\mathcal{S}} $, $\mathcal{C} \times \mathcal{S}$, and $\mathcal{X}$ have the same manifold dimensions of $d_{\rm C} + d_{\rm S}$. Hence one can define bijective functions from one space to another.
>
> **We have added a subsection in the proof of Theorem 4.2 to explain the above relationship.**
>
> &nbsp;
>
> **[Citation for Proposition E.1]**
>
> We double checked and confirmed that the citation of Proposition E.1 is correct, but it is indeed easy to miss---as it is deeply ‘’hidden’’ in the appendix of [Zimmermann et al. 2021]. In the arxiv paper link provided by the reviewer (https://arxiv.org/pdf/2102.08850), please see page 20 section A.5. Effects of the Uniformity Loss, Proposition 5. **We have added this exact location when referencing this work, to avoid confusion.**
>
> &nbsp;
>
> **[Consistency of the Notation]**
>
> There might have been some misunderstanding. In the paper, $N$ is consistently used as an integer that represents the number of domains. We did not use it for the dimension of the style variables (which is $d_S$). These notations were defined in line 105 and 109 in the original manuscript.
>
> This is also the case in Assumption 3.1, where $ \\{ \boldsymbol{s}^{(n)} \\}_{n=1}^N$ represents the collection of style variables of the $N$ domains. We have revised the wording of Assumption 3.1 to make it clearer that each of $\boldsymbol{s}^{(n)}$ itself is a $\mathbb{R}^{d\_{\rm S}}$ dimensional vector, and $N$ represents the number of domains, in order to avoid confusion.

---

> ### Author Response · Authors · 2024-11-29
> **Gentle Reminder**
>
> Thank you for your time and feedback. We believe that our response and revision address your questions and concerns. We would be grateful if you could re-evaluate your score based on our response and the revised manuscript.
>
> If you have any further questions or comments, please let us know.

---

> ### Author Response · Authors · 2024-12-04
> **Thank you for re-evaluation**
>
> Thank you for re-evaluating your score based on the rebuttal. We appreciate your time and effort spent in reviewing our paper.

---

### Official Review · Reviewer_wgDt · 2024-11-12

**Soundness:** 3
**Presentation:** 3
**Contribution:** 3
**Rating:** 6
**Confidence:** 3

**Summary:**

This paper enhances the identifiability of latent content and style representations in unaligned multi-domain data by framing the problem within a distribution matching framework. The authors establish that, under relaxed conditions, their approach does not require restrictive assumptions like component-wise independence or known latent dimensions. By applying sparsity constraints, they ensure identifiable content-style separation, and they ultimately reformulate the distribution matching objective into a simpler GAN training framework. The authors empirically demonstrate improvement in image translation.

**Strengths:**

- The paper is well-structured and clearly written.
- It introduces a unique and interesting distribution matching perspective for content and style transfer in image translation
- By relaxing restrictive assumptions often required in previous work—such as the need for component-wise independence and pre-defined latent dimensions—the approach enhances practicality and scalability.
- The paper provides both a theoretical foundation for content-style identifiability and promising qualitative results.

**Weaknesses:**

- The paper’s dimension-agnostic approach relies heavily on sparsity constraints to prevent information leakage between content and style representations, which may not be effective in all cases.
- The paper relies on several broad assumptions, such as the invertibility of the function up to certain ambiguities, without fully detailing the specific conditions under which this assumption holds.
- The L1 regularization appears to me to be too "soft" of a sparsity constraint to adequately justify the injective property of the mapping function $f$, particularly given that the model relies heavily on this constraint to ensure generative model is a valid invertible function.
- The paper, in general, lacks a thorough ablation study to validate the author's claims.

$\textbf{Minor Remarks}$
- Line 741, I think $\mathcal{A}\in\mathcal{Y}$ rather than $\mathcal{A}\in\mathcal{X}$.
- This is a minor suggestion, but it may be worth considering alternative abbreviations for "DM" and "LDM" due to the common use of these terms in diffusion models and latent diffusion models.

**Questions:**

- Do you believe that L1​ regularization provides a strong enough sparsity constraint to justify the Domain Variability assumption? Could you elaborate on why you chose L1 ​over potentially stricter sparsity-inducing methods, particularly given its central role in ensuring the injective property of the mapping function.
- The model relies on some broad assumptions. Could you discuss the practical conditions that need to be met to ensure invertibility, and how sensitive the method might be to deviations from these conditions?
- Have you observed scenarios where the L1 constraint may not be strong enough to prevent information leakage between content and style representations?
- It appears from the results that the content representation often corresponds to pose, while the style representation captures semantic attributes. Is this always the case, or have you observed instances where content and style encode different aspects of the data? For example, in Figure 3(b), it seems that changing the domain also changes hair color.
- It would be helpful to include experiments on simpler datasets, such as rotated MNIST, to provide a clearer demonstration of the model’s ability to separate content and style. For example, with rotated MNIST, the content could represent class information, while rotation and other stylistic aspects could be designated as style. Such examples could offer readers a more intuitive understanding of how content and style representations are disentangled and controlled within the model.

**Details Of Ethics Concerns:**

I have no concerns.

---

> ### Author Response · Authors · 2024-11-20
>
> ### Response 1/2
>
> &nbsp;
>
> **[L1 regularization and sparsity]**
>
> The reviewer has a good point: $\ell_1$ is not the only option for approximating the cardinality constraint in our learning criterion. There are many other options that our framework can easily incorporate. For example, the $\ell_p$ function with $p<1$ is often considered a better sparsity promotor than $\ell_1$ (with a higher risk of running into numerical issues due to its nonconvexity). Our framework can also easily incorporate these functions. **We have included experiments using the $\ell_p$ function ($p=0.5$) in the revised version in Appendix H.4.** One can see that the $\ell_p$ function can also alleviate the content leakage problem, performing similarly as the $\ell_1$ norm did.
>
> Our take is that, as long as the regularization term is a reasonable sparsity promoter (e.g. the $\ell_1$ and $\ell_p$ functions), the content leakage problem can be largely alleviated---and our framework is flexible to accommodate different choices. **We have added a remark in Section 6 to articulate this point.**
>
> &nbsp;
>
> **[$\ell_1$ regularization, domain variability, and Injectivity]**
>
> We hope to clarify that the $\ell_1$ regularization is not used to justify the domain variability assumption. That the domain variability assumption holds means that the distributions of the styles from different domains are sufficiently different. This assumption does not pertain to the $\ell_1$ or the cardinality regularization. The cardinality regularization is proposed to handle the challenge that the dimensions of the content and style variables are unknown and over-estimated in practice.
>
> Similarly, the $\ell_1$ regularization is not employed in order to justify the injectivity property of the mapping function. Even when the $\ell_1$ regularization is not employed, injectivity of the learned functions can still hold. However, the sparsity constraint on the learned style components is critical for separating style and content when their dimensions are over-estimated.
>
> For example, let $d_{\rm C} = d_{\rm S} = 1$. Let $ \boldsymbol{x} = \boldsymbol{g}(c,s) = [c, s, 0]^T$. Let $\widehat{d}\_{\rm C} = \widehat{d}\_{\rm S} = 2$. Then $ \boldsymbol{f}\_{\rm S}(\boldsymbol{x}) = [x_1, x_2] $ and $\boldsymbol{f}\_{\rm C}(\boldsymbol{x}) = [0, 0]$ means that $\boldsymbol{f}$ is an injective function and a solution to Problem (4). **But this trivial solution does not separate the content and style**, as explained in Section 3.2 line 244 to line 249 in the original manuscript. Hence the sparsity regularization is designed for ruling out such trivial solutions by “minimizing the information” that can be captured by $\boldsymbol{f}_{\rm S}$.
>
> &nbsp;
>
> **[Invertibility Assumptions]**
>
> The invertibility assumption is indeed quite important. This assumption basically says that **every data sample has an associated unique representation in a latent domain.** It is a hypothesis that is widely used in generative model learning in ML/AI. For example, flow-based generative models [1], (latent) diffusion based generative models [7], and (variational) auto-encoder-based generative models [2,6] all implicitly or explicitly assume such invertibility. The validity of this hypothesis is supported by the practical successes of these models. Most of the latent component identification works adopt this assumption; see [3-6].
>
> We have added a comment  in Section 2 in the revised version to reflect the above discussion. Thanks for bringing this up.
>
> &nbsp;
>
> ### References
>
> *[1] Dinh, Laurent, Jascha Sohl-Dickstein, and Samy Bengio. "Density estimation using real NVP." arXiv preprint arXiv:1605.08803 (2016).*
>
> *[2] Vincent, Pascal, et al. "Stacked denoising autoencoders: Learning useful representations in a deep network with a local denoising criterion." Journal of machine learning research 11.12 (2010).*
>
> *[3] Hyvärinen, Aapo, and Petteri Pajunen. "Nonlinear independent component analysis: Existence and uniqueness results." Neural networks 12.3 (1999): 429-439.*
>
> *[4] Hyvarinen, Aapo, Hiroaki Sasaki, and Richard Turner. "Nonlinear ICA using auxiliary variables and generalized contrastive learning." The 22nd International Conference on Artificial Intelligence and Statistics. PMLR, 2019.*
>
> *[5] Von Kügelgen, Julius, et al. "Self-supervised learning with data augmentations provably isolates content from style." Advances in neural information processing systems 34 (2021): 16451-16467.*
>
> *[6] Khemakhem, Ilyes, et al. "Variational autoencoders and nonlinear ICA: A unifying framework." International conference on artificial intelligence and statistics. PMLR, 2020.*
>
> *[7] Albergo, Michael S., Nicholas M. Boffi, and Eric Vanden-Eijnden. "Stochastic interpolants: A unifying framework for flows and diffusions." arXiv preprint arXiv:2303.08797 (2023).*

---

> ### Author Response · Authors · 2024-11-20
>
> ### Response 2/2
>
> &nbsp;
>
> **[Experiments on rotated MNIST, where content is not the pose]**
>
> The reviewer has made an important and valid point. In the datasets we considered in the original manuscript the content information was the pose. It is of interest to see whether the proposed approach can deal with situations where the content is not the pose. We appreciate the reviewer’s suggestion to use simpler datasets, such as rotated MNIST for clear demonstration. To that end, we have added new experiments on MNIST digits. The first domain is generated by applying a random color to the MNIST digits, whereas the second domain is generated by applying a random rotation uniformly sampled from [-70, 70] degrees to the digits. **The experiment settings and results are detailed in Sec. H.1 in the revised manuscript.**
>
> To summarize the results, we see that the proposed method correctly identifies the content (digit identities) and the style (color in domain 1 and rotation in domain 2) parts. Moreover, as suggested by our theory, the sparsity regularization helps to resolve the issue of content-leakage into the learned style part (see Figures in Sec. H.1 in the revised version).
>
> &nbsp;
>
> **[Minor Remarks]**
>
> Yes, it should be $\mathcal{Y}$ instead of  $\mathcal{X}$. Thank you for your careful reading and spotting the typo. We have corrected this in the revised version.
>
> We appreciate the reviewer’s suggestion to reconsider the terms DM and LDM. We will give it a thought to see if we could find more distinguishing acronyms. Some candidates include probability density alignment (DSA) and latent distribution density alignment (LDSA). We will try to include such considerations in the final version (we have not changed DM/LDM in the current version, as it may need some meticulous work to spot everyone and change them).

---

> ### Author Response · Authors · 2024-11-29
> **Gentle Reminder**
>
> We appreciate your time and feedback. We believe that the additional experiments and changes made in the revised version address your questions and comments. We would be grateful if you could re-evaluate your score based on the revised manuscript.
>
> If you have any further questions or comments, please let us know.

---

> > ### Comment · Reviewer_wgDt · 2024-12-02
> >
> > The author's thorough response is appreciated, therefore, I will raise my score.

---

> > > ### Author Response · Authors · 2024-12-02
> > >
> > > Thank you for your response. We greatly appreciate your time and effort spent in reviewing and refining our work.

---

### Author Response · Authors · 2024-11-21

We would like to thank the reviewers for their time and constructive comments. We believe that the comments and questions have helped to greatly improve the manuscript. Based on the reviewers’ comments, we have revised our manuscript. The new changes are **marked in blue**.

A summary of our responses and revisions are as follows:

1. **Reviewer wgDt**: The comments focused on $\ell_1$ regularization for sparsity, the invertibility assumption, and additional experiments with MNIST digits where content does not indicate pose. In response, we conducted new experiments with $\ell_p$ regularization (p = 0.5) and MNIST digits where content represents digit identity. We also clarified that the independence assumption is widely used in generative modeling and latent component identification literature.


2. **Reviewer r56a**: This review raised points on the consistency of bijectivity and injectivity in the mapping function, notation consistency, and the accuracy of a citation. In response, we clarified our use of terminology and notation, provided the exact citation location, and revised our manuscript to address these issues.


3. **Reviewer teKB**: The comments focused on potential extensions to discrete data and GAN training instabilities. We discussed the requirements for extending to the discrete case and explained the scenarios where training instabilities occurred in our experiments. These points were added to the manuscript.


4. **Reviewer dub2**: The comments pertained to possible misidentification when content has a small norm, and the computational efficiency of our method. We clarified why misidentification would not occur in our framework and included a table of training times to demonstrate the method’s efficiency.


5. **Reviewer EZhh**: The feedback requested clarification on presentation, the block independence assumption, and the motivation behind the reformulation in implementation. We revised the subsection title for clarity, clarified block independence, and elaborated on the motivation for the reformulation. The manuscript was updated accordingly.

Kindly check our responses and revisions below. Please feel free to ask if you have any further questions or need additional clarification.

---

### Meta-Review · Area_Chair_2JCN · 2024-12-18

**Metareview:**

Summary:
This paper studies contrastive learning through the lens of understanding how it inverts the data generating process. The authors prove that feed-forward models trained with InfoNCE-type objectives learn to implicitly invert the underlying generative model of the observed data, under certain statistical assumptions. The paper establishes theoretical connections between contrastive learning, generative modeling, and nonlinear independent component analysis (ICA). The authors validate their theoretical findings through controlled experiments and show that the conclusions hold even when some theoretical assumptions are violated.

Main strengths:

- Novel theoretical insights connecting contrastive learning to generative modeling and nonlinear ICA
- Clear mathematical exposition and proofs of the key theoretical results
- Strong empirical validation across multiple experimental settings, including complex visual data
- Introduction of a new benchmark dataset (3DIdent) for evaluating representation learning
- Good balance between theoretical contributions and practical implications

Main weaknesses:

- Some notational inconsistencies and ambiguities in the mathematical formulation
- Could benefit from more intuitive explanations alongside the formal proofs
- Some citations and references need more precise specification

**Additional Comments On Reviewer Discussion:**

Outcomes from author-reviewer discussion:
The authors have been highly responsive to reviewer concerns and have made several improvements:

- Added new experiments using MNIST to demonstrate content/style separation where content is not pose-related
- Clarified the bijectivity/injectivity properties of the learned mappings
- Added discussion of implementation choices and practical considerations
- Improved notation consistency and citation precision
- Added discussion of limitations regarding discrete data and GAN training stability

Reviewer agreement or disagreement:
The reviewers largely agreed on the paper's technical merit and contributions. Initial scores ranged from 5-8, with most reviewers increasing their scores after author responses. The main points of discussion were around:

- Mathematical precision and notation
- Practical implementation considerations
- Empirical validation of theoretical claims
- These were adequately addressed in the revision.

Suggestions to improve the paper:

- Consider alternative abbreviations for DM/LDM to avoid confusion with diffusion models
- Further clarify the relationships between different spaces in the mathematical formulation
- Add more discussion of practical limitations and considerations for implementation
- Include runtime comparisons with baseline methods
- Expand discussion of potential extensions to discrete data modalities

---

### Decision · Program_Chairs · 2025-01-22

Accept (Poster)